# LABEL-NOISE ROBUST DIFFUSION MODELS

**Byeonghu Na**[1]**, Yeongmin Kim**[1]**, HeeSun Bae**[1]**, Jung Hyun Lee**[2]**, Se Jung Kwon**[2]**,
Wanmo Kang**[1] **& Il-Chul Moon**[1,3]
[1]KAIST, [2]NAVER Cloud, [3]summary.ai
{wp03052,alsdudrla10,cat2507,wanmo.kang,icmoon}@kaist.ac.kr,
{junghyun123.lee,sejung.kwon}@navercorp.com

## ABSTRACT

Conditional diffusion models have shown remarkable performance in various generative tasks, but training them requires large-scale datasets that often contain noise in conditional inputs, a.k.a. noisy labels. This noise leads to condition mismatch and quality degradation of generated data. This paper proposes Transition-aware weighted Denoising Score Matching (TDSM) for training conditional diffusion models with noisy labels, which is the first study in the line of diffusion models. The TDSM objective contains a weighted sum of score networks, incorporating instance-wise and time-dependent label transition probabilities. We introduce a transition-aware weight estimator, which leverages a time-dependent noisy-label classifier distinctively customized to the diffusion process. Through experiments across various datasets and noisy label settings, TDSM improves the quality of generated samples aligned with given conditions. Furthermore, our method improves generation performance even on prevalent benchmark datasets, which implies the potential noisy labels and their risk of generative model learning. Finally, we show the improved performance of TDSM on top of conventional noisy label corrections, which empirically proving its contribution as a part of label-noise robust generative models. Our code is available at: `https://github.com/byeonghu-na/tdsm`.

## 1 INTRODUCTION

Diffusion models have gained significant interest in various fields, such as image (Song et al., 2020; Dhariwal & Nichol, 2021; Karras et al., 2022; Kim et al., 2023) and video generation (Ho et al., 2022; Voleti et al., 2022), for their high-quality sample generation. Moreover, they have shown impressive results in conditional generation problems (Kong et al., 2021; Meng et al., 2022a;b; Rombach et al., 2022; Saharia et al., 2022). However, training diffusion models requires large-scale datasets, which often contain data instances with noisy labels due to expensive labeling costs and a lack of experts (Xiao et al., 2015; Song et al., 2022). These noisy labels cause problems when learning conditional generative models, as shown in Figure 1 of Kaneko et al. (2019) and Figure 1b. Although the problem of learning with noisy labels has been extensively studied in supervised learning (Patrini et al., 2017; Han et al., 2018; Li et al., 2021; Nordstrom et al., 2022; Bae et al., 2024), there are only a few studies on generative models (Thekumparampil et al., 2018; Kaneko et al., 2019), and as far as we know, there has been no theoretical discussion and remedies for diffusion models.

This paper proposes a method for training conditional diffusion models with noisy labels. To achieve this, we show that the noisy-label conditional score can be interpreted as a linear combination of clean-label conditional scores based on instance-wise and time-dependent label transition information. Using this relationship, we propose an objective of Transition-aware weighted Denoising Score Matching (TDSM). To train a score network with the proposed objective, we introduce an estimator of the transition-aware weights, and we explain the practical implementation that is not trivial because of multiple score network evaluations from class transitions. We conduct experiments on various datasets and label noise settings, and our diffusion models, which are trained by the TDSM objective, generate samples that match the given condition well (see Figure 1c), and our models outperform baseline models on conditional metrics as well as most unconditional metrics. Furthermore, we empirically demonstrate that TDSM is a key component of label-noise robust diffusion models by showing improved performance when combined with conventional noisy label corrections.

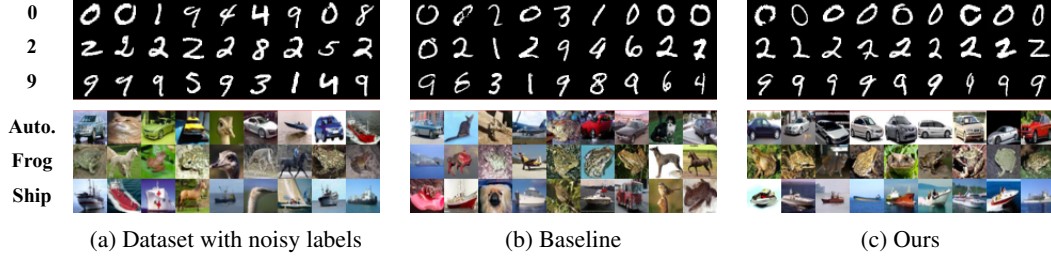

| (a) Dataset with noisy labels | (b) Baseline | (c) Ours |

Figure 1: (a) Examples of noisy labeled datasets of MNIST (top) and CIFAR-10 (bottom), and (b-c) the randomly generated images of baseline and our models, trained with the noisy labeled datasets.

In summary, our contributions are as follows:

- We propose Transition-aware weighted Denoising Score Matching for training conditional diffusion models with noisy labels, which is the first time in the diffusion model.
- We prove that there is a linear relationship between noisy-label and clean-label conditional scores based on the instance-wise and time-dependent label transition probability.
- We suggest an estimator structure with a time-dependent noisy-label classifier and a transition matrix to evaluate pairwise class transitions.
- We empirically investigate the negative impact of noisy labeled datasets on diffusion models, and the proposed method achieves superior performance in the presence of noisy labels.

## 2 PROBLEM FORMULATION

This section formulates the problem of learning diffusion models from noisy labels. We define the data space as $\mathcal{X} \in \mathbb{R}^d$ and the label space as $\mathcal{Y} = \{1, ..., c\}$, where $d$ is the data dimension and $c$ is the number of classes. A data instance is denoted by $\mathbf{x} \in \mathcal{X}$, and its corresponding clean label is denoted by $y \in \mathcal{Y}$. We assume that only a noisy labeled training dataset $\tilde{D} = \{(\mathbf{x}^{(i)}, \tilde{y}^{(i)})\}_{i=1}^n$ is available, sampled from a noisy-label data distribution $\tilde{p}_{\text{data}}(\mathbf{X}, \tilde{Y})$, where $\tilde{Y} \in \mathcal{Y}$ represents the noisy label corresponding to $\mathbf{X}$, and $n$ is the number of data instances.

**Label Noise** We focus on the class-conditional label noise setting, where the noisy label $\tilde{Y}$ is assumed to be independent of the instance $\mathbf{X}$ given the clean label $Y$ (Natarajan et al., 2013; Patrini et al., 2017). This setting is commonly used in the label noise literature (Kaneko et al., 2019; Li et al., 2021; Zhang et al., 2021b). From a generative perspective, it can be expressed as follows:

$$p(\mathbf{x}|\tilde{Y} = \tilde{y}) = \sum_{y=1}^c p(Y = y|\tilde{Y} = \tilde{y})p(\mathbf{x}|Y = y, \tilde{Y} = \tilde{y}) = \sum_{y=1}^c p(Y = y|\tilde{Y} = \tilde{y})p(\mathbf{x}|Y = y), \quad (1)$$

where the second equation is derived by the assumption from the class-conditional label noise setting. Eq. (1) means that each noisy-label conditional distribution is a mixture of clean-label conditional distributions. We define a reverse transition matrix as $\boldsymbol{S} = [S_{i,j}] \in [0, 1]^{c \times c}$ where $S_{i,j} = p(Y = j|\tilde{Y} = i)$.[1] By the definition, we have $\sum_{j=1}^c S_{i,j} = 1$. We provide a detailed review of learning with noisy labels in Appendix B.3.

**Diffusion Model** Now, we briefly introduce the diffusion model. We describe it through stochastic differential equations. Specifically, Eqs. (2) and (3) represent the forward diffusion process and its corresponding reverse process, respectively (Anderson, 1982; Song et al., 2020; Kim et al., 2022a).

$$d\mathbf{x}_t = \mathbf{f}(\mathbf{x}_t, t)dt + g(t)d\mathbf{w}_t, \quad (2)$$

$$d\mathbf{x}_t = [\mathbf{f}(\mathbf{x}_t, t) - g^2(t)\nabla_{\mathbf{x}_t} \log p_t(\mathbf{x}_t)]d\bar{t} + g(t)d\bar{\mathbf{w}}_t, \quad (3)$$

where $\mathbf{f}$ is the drift function; $g$ is the volatility function; $p_t(\mathbf{x}_t)$ is the probability density of $\mathbf{x}_t$; $\mathbf{w}_t$ and $\bar{\mathbf{w}}_t$ denote the forward and reverse standard Brownian motion, respectively; and $t \in [0, T]$. The

---

[1]We called $\boldsymbol{S}$ the *reverse* transition matrix because in most of the literature on learning with noisy labels, the transition matrix usually contains the information of $p(\tilde{Y} = \tilde{y}|Y = y)$. We omit *reverse* if it is not confusing.

forward process gradually perturbs the data instance $\mathbf{x} = \mathbf{x}_0$ to $\mathbf{x}_T$, and the reverse process gradually denoises from $\mathbf{x}_T$ to $\mathbf{x}_0$. We can sample data instances through the reverse process.

In spite that the generation task will require the reverse diffusion process, this process is intractable because a data score $\nabla_{\mathbf{x}_t} \log p_t(\mathbf{x}_t)$ is generally not accessible. Therefore, the diffusion model aims to train the score network to approximate $\nabla_{\mathbf{x}_t} \log p_t(\mathbf{x}_t)$ through the score matching objective function with L2 loss (Song et al., 2020). Specifically, the conditional diffusion model trains a score network $\mathbf{s}_{\boldsymbol{\theta}}$, where $\boldsymbol{\theta}$ is trainable parameters, to estimate the gradient of conditional log-likelihood, $\nabla_{\mathbf{x}_t} \log p_t(\mathbf{x}_t|Y)$ (Chao et al., 2022). Denoising score matching (DSM) (Vincent, 2011; Song et al., 2020) is commonly used to train a score network:

$$\mathcal{L}_{\text{DSM}}(\boldsymbol{\theta}; p_{\text{data}}(\mathbf{X}, Y)) \tag{4}$$
$$:= \mathbb{E}_t\Big\{\lambda(t)\mathbb{E}_{\mathbf{x},y \sim p_{\text{data}}(\mathbf{X},Y)}\mathbb{E}_{\mathbf{x}_t \sim p_{t|0}(\mathbf{X}_t|\mathbf{x},y)}\Big[\big|\big|\mathbf{s}_{\boldsymbol{\theta}}(\mathbf{x}_t, y, t) - \nabla_{\mathbf{x}_t} \log p_{t|0}(\mathbf{x}_t|\mathbf{x}, Y = y)\big|\big|_2^2\Big]\Big\},$$

where $p_{\text{data}}$ is a clean-label data distribution, $p_{t|0}(\mathbf{X}_t|\mathbf{X}_0 = \mathbf{x}, Y = y)$ is a perturbation kernel, $t$ is sampled over $[0, T]$, and $\lambda(t)$ is a temporal weight function.[2] The perturbation kernel $p_{t|0}(\mathbf{X}_t|\mathbf{x}, y)$ is generally Gaussian distributed and independent of the label, so it can be computed in closed form. In this paper, we mainly focus on directly learning a conditional score network, and we discuss the application of guidance methods in Appendix C. More discussion is in Appendices B.1 and B.2.

**Label-Noise Robust Generation of Diffusion Models**  Our goal is, given a training dataset with noisy labels, to train a conditional diffusion model that can generate samples conditioned on a specified clean label. This task poses a significant challenge due to the presence of noisy labels. When we train a diffusion model using Eq. (4) with a noisy labeled training dataset, the model eventually learns the noisy-label conditional score, $\nabla_{\mathbf{x}_t} \log p_t(\mathbf{x}_t|\tilde{Y})$. As a result, the generative distribution from the model follows the mixture of the conditional distributions (see Eq. 1), leading to the generation of samples from different classes. Furthermore, this problem cannot be solved by directly applying methods from generative adversarial networks (GANs) (Kaneko et al., 2019; Thekumparampil et al., 2018), as we are dealing with a time-dependent vector that has a probabilistic interpretation, namely the data score, and the optimization problem has a different structure.

## 3 LABEL-NOISE ROBUST DIFFUSION MODELS

This section presents our approach to training diffusion models with noisy labels. We begin by formulating an objective function that leverages the linear relationship between the clean-label and noisy-label conditional scores. Our proposed objective function involves a weighted sum of conditional score networks, where the weights are determined by time-dependent label transition information at the instance level. To estimate these weights, we introduce an estimator that utilizes a transition matrix and a time-dependent noisy-label classifier. Lastly, we present some useful techniques for practical implementations.

### 3.1 LINEAR RELATIONSHIP BETWEEN CLEAN- AND NOISY-LABEL CONDITIONAL SCORES

As discussed in Section 2, if the score network is optimized by the original DSM objective, Eq. (4), with a noisy labeled dataset, assuming $Y = \tilde{Y}$; then the score network is expected to converge on the noisy-label conditional score. See Appendix A.1 for further discussion.

**Remark.** *Let $\boldsymbol{\theta}_{DSM}^* := \arg\min_{\boldsymbol{\theta}} \mathcal{L}_{DSM}(\boldsymbol{\theta}; \tilde{p}_{data}(\mathbf{X}, \tilde{Y}))$ be the optimal parameters obtained by minimizing the DSM objective. Then, $\mathbf{s}_{\boldsymbol{\theta}_{DSM}^*}(\mathbf{x}_t, y, t) = \nabla_{\mathbf{x}_t} \log p_t(\mathbf{x}_t|\tilde{Y} = y)$ for all $\mathbf{x}_t, y, t$.*

Therefore, to train the score network in the alignment of the clean-label conditional score, we modify the objective function to adjust the gradient signal from the score matching. We start this adjustment by establishing the relationship between clean-label and noisy-label conditional scores.

**Theorem 1.** *Under a class-conditional label noise setting, for all $\mathbf{x}_t, \tilde{y}, t$,*

$$\nabla_{\mathbf{x}_t} \log p_t(\mathbf{x}_t|\tilde{Y} = \tilde{y}) = \sum_{y=1}^c w(\mathbf{x}_t, \tilde{y}, y, t)\nabla_{\mathbf{x}_t} \log p_t(\mathbf{x}_t|Y = y), \tag{5}$$

*where $w(\mathbf{x}_t, \tilde{y}, y, t) := p(Y = y|\tilde{Y} = \tilde{y})\frac{p_t(\mathbf{x}_t|Y=y)}{p_t(\mathbf{x}_t|\tilde{Y}=\tilde{y})}$.*

---

[2] We add the distribution $p_{\text{data}}(\mathbf{X}, Y)$ in the notation of $\mathcal{L}$ to emphasize the input distribution since we are considering the distribution change due to label noise.

Figure 2: Contour maps of $w(\mathbf{x}_t, \tilde{Y} = 1, Y = 1, t)$ in the 2-D Gaussian mixture model at different diffusion timesteps. The label transition probability is set to $p(Y|\tilde{Y} = 1) = (0.8, 0.2)$. The dots represent samples from each clean label (orange for class 1, green for class 2), and the dashed lines represent contours with annotated values.

The proof of Theorem 1 is provided in Appendix A.2. Since $w(\mathbf{x}_t, \tilde{y}, y, t) \geq 0$ and $\sum_{y=1}^{c} w(\mathbf{x}_t, \tilde{y}, y, t) = 1$, Theorem 1 implies that the noisy-label conditional score can be expressed as a convex combination of the clean-label conditional scores with coefficients $w(\cdot, \cdot, y, \cdot)$.

We call $w(\mathbf{x}_t, \tilde{y}, y, t)$ the *transition-aware weight function*. This function represents instance-wise and time-dependent (reverse) label transitions by Proposition 2.[3] See Appendix A.3 for the proof.

**Proposition 2.** *Under a class-conditional label noise setting, $w(\mathbf{x}_t, \tilde{y}, y, t) = p_t(Y = y|\tilde{Y} = \tilde{y}, \mathbf{x}_t)$.*

This probability has been used for evaluating a confidence score $p(Y = y|\tilde{Y} = y, \mathbf{x}_t)$ in some supervised classification frameworks for instance-dependent label noise learning (Berthon et al., 2021) whereas this paper is its first application to the deep generative model community. There are similar approaches to overcome noisy labels in GANs under the class-conditional label noise setting (Kaneko et al., 2019; Thekumparampil et al., 2018), but these approaches can simply utilize $p(Y = y|\tilde{Y} = \tilde{y})$ without either instance-based estimation or diffusion time-dependent estimation. However, training diffusion models with noisy labels poses a significant challenge because we need instance-dependent label noise information, even under the class-conditional label noise, and this distinction is the contribution of Theorem 1.

To visualize our transition-aware weight function, we create an analogy of a 2-D Gaussian mixture model. The detailed setup is explained in Appendix E.4. Figure 2 plots contour maps of $w(\mathbf{x}_t, \tilde{Y} = 1, Y = 1, t)$ with different diffusion timesteps. Note that $\sum_{y=1}^{2} w(\mathbf{x}_t, \tilde{Y} = 1, Y = y, t) = 1$. In the small timesteps, $w(\mathbf{x}_t, \tilde{Y} = 1, Y, t)$ tends to become hard labels with respect to the clean labels. This means that the noisy-label conditional score, computed by the weighted sum of the scores, becomes equal to the conditional score of the clean label of $\mathbf{x}_t$. On the other hand, in the large timesteps, $w(\mathbf{x}_t, \tilde{Y} = 1, Y, t)$ vectors for most $\mathbf{x}_t$ are similar to the prior probability of label transition, $p(Y|\tilde{Y} = 1) = (0.8, 0.2)$. Consequently, the noisy-label conditional scores for most $\mathbf{x}_t$ are influenced by the clean conditional scores of the other class; and this mis-matched score is being learned. This mis-matched learning leads to both improper generation with intended conditions and lower generation performance in its quality.

### 3.2 TRANSITION-AWARE WEIGHTED DENOISING SCORE MATCHING

From the result of Theorem 1, we propose an objective of transition-aware weighted denoising score matching (TDSM), which is designed to minimize the distance between the transition-aware weighted sum of conditional score network outputs and the perturbed data score:

$$\mathcal{L}_{\text{TDSM}}(\boldsymbol{\theta}; \tilde{p}_{\text{data}}(\mathbf{X}, \tilde{Y})) \tag{6}$$

$$:= \mathbb{E}_t \left\{ \lambda(t) \mathbb{E}_{\mathbf{x}, \tilde{y} \sim \tilde{p}_{\text{data}}} \mathbb{E}_{\mathbf{x}_t \sim p_{t|0}} \left[ \left\| \sum_{y=1}^{c} w(\mathbf{x}_t, \tilde{y}, y, t) \mathbf{s}_{\boldsymbol{\theta}}(\mathbf{x}_t, y, t) - \nabla_{\mathbf{x}_t} \log p_t(\mathbf{x}_t|\mathbf{x}, \tilde{Y} = \tilde{y}) \right\|_2^2 \right] \right\}.$$

The intuition behind the TDSM objective is as follows. According to Remark, the denoising score matching objective optimizes the noisy-label conditional score. However, Theorem 1 shows that this score can be expressed as a convex combination of the clean-label conditional scores using the transition-aware weight function. Therefore, by using the transition-aware weighted sum of the conditional score model outputs as the target of L2 loss, our score network will eventually converge to the clean-label conditional score. This intuition is theoretically guaranteed by Theorem 3.

---

[3] It should be noted that $p_t(Y = y|\tilde{Y} = \tilde{y}, \mathbf{x}_t) \neq p_t(Y = y|\tilde{Y} = \tilde{y})$.

**Theorem 3.** *Let $\boldsymbol{\theta}^*_{TDSM} := \arg\min_{\boldsymbol{\theta}} \mathcal{L}_{TDSM}(\boldsymbol{\theta}; \tilde{p}_{data}(\mathbf{X}, \tilde{Y}))$ be the optimal parameters obtained by minimizing the TDSM objective. Then, under a class-conditional label noise setting with an invertible transition matrix, $\mathbf{s}_{\boldsymbol{\theta}^*_{TDSM}}(\mathbf{x}_t, y, t) = \nabla_{\mathbf{x}_t} \log p_t(\mathbf{x}_t | Y = y)$ for all $\mathbf{x}_t, y, t$.*

See Appendix A.4 for the proof. We discuss the other forms of the objective function, such as noise estimation (Ho et al., 2020) and data reconstruction (Kingma et al., 2021), in Appendix A.6.

It should be noted that the invertibility assumption of the transition matrix has been widely used in the noisy label community (Li et al., 2021; Zhu et al., 2021). Here, we assume that the given noisy label is sufficiently likely to be a clean label, and without this assumption, there is no empirical evidence to claim a certain class to be a clean label. In other words, the transition matrix would be a diagonally dominant matrix, which is invertible. Additionally, a possible approach to ensure the invertibility of the transition matrix is to mix the transition matrix with the identity matrix (Patrini et al., 2017).

The alternative approach to consider is the $\boldsymbol{S}$-weighted DSM objective, denoted as $\mathcal{L}_{\mathrm{SDSM}}$. In this objective, the weights in the TDSM objective are replaced by time- and instance-independent weights $S_{\tilde{y},y} = p(Y = y | \tilde{Y} = \tilde{y})$ from the transition matrix $\boldsymbol{S}$, which is inspired by the GAN-based methods:

$$\mathcal{L}_{\mathrm{SDSM}}(\boldsymbol{\theta}; \tilde{p}_{\mathrm{data}}(\mathbf{X}, \tilde{Y})) := \mathbb{E}_t \left\{ \lambda(t) \mathbb{E}_{\mathbf{x}, \tilde{y}} \mathbb{E}_{\mathbf{x}_t} \left[ \left\| \sum_{y=1}^c S_{\tilde{y},y} \mathbf{s}_{\boldsymbol{\theta}}(\mathbf{x}_t, y, t) - \nabla_{\mathbf{x}_t} \log p_t(\mathbf{x}_t | \mathbf{x}, \tilde{Y} = \tilde{y}) \right\|_2^2 \right] \right\}. \quad (7)$$

However, Proposition 4 proves that the score network trained by the $\boldsymbol{S}$-weighted DSM objective cannot converge to a clean-label conditional score. The proof is provided in Appendix A.5.

**Proposition 4.** *Let $\boldsymbol{\theta}^*_{SDSM} := \arg\min_{\boldsymbol{\theta}} \mathcal{L}_{SDSM}(\boldsymbol{\theta}; \tilde{p}_{data}(\mathbf{X}, \tilde{Y}))$ be the optimal parameters obtained by minimizing the $\boldsymbol{S}$-weighted DSM objective. Then, under a class-conditional label noise setting with an invertible transition matrix, $\mathbf{s}_{\boldsymbol{\theta}^*_{SDSM}}(\mathbf{x}_t, y, t)$ differs from $\nabla_{\mathbf{x}_t} \log p_t(\mathbf{x}_t | Y = y)$.*

Figure 8 in Appendix F.1 visualizes the optimal score networks of DSM variants in a 2-D toy case.

### 3.3 ESTIMATION OF TRANSITION-AWARE WEIGHTS

To implement the TDSM objective, we need to estimate the transition-aware weight function $w(\mathbf{x}, \tilde{y}, y, t)$. First, we consider the case that we have the transition matrix $\boldsymbol{S}$, so we can evaluate $p(Y = y | \tilde{Y} = \tilde{y})$, which has been a common goal in the community of noisy labels (Li et al., 2021; Zhang et al., 2021b). Then, we can compute the transition-aware weight function $w$ using the noisy label prediction probability $p_t(\tilde{Y} | \mathbf{x}_t)$ and the transition matrix $\boldsymbol{S}$. The below introduces the estimator $\hat{w}$ of $w$ using the noisy labeled dataset (see Appendix A.7 for detailed derivations):

$$w(\mathbf{x}_t, \tilde{y}, y, t) = \frac{S_{\tilde{y},y} p(\tilde{Y} = \tilde{y})}{p_t(\tilde{Y} = \tilde{y} | \mathbf{x}_t)} \sum_{i=1}^c \frac{S_{y,i}^{-1} p_t(\tilde{Y} = i | \mathbf{x}_t)}{p(\tilde{Y} = i)}, \quad \hat{w}(\mathbf{x}_t, \tilde{y}, y, t) := \frac{S_{\tilde{y},y} n_{\tilde{y}}}{\tilde{\mathrm{h}}_{\boldsymbol{\phi}}(\mathbf{x}_t, t)_{\tilde{y}}} \sum_{i=1}^c \frac{S_{y,i}^{-1} \tilde{\mathrm{h}}_{\boldsymbol{\phi}}(\mathbf{x}_t, t)_i}{n_i}, \quad (8)$$

where $\tilde{\mathbf{h}}_{\boldsymbol{\phi}}(\mathbf{x}_t, t)$ is the time-dependent noisy-label classifier; $\boldsymbol{\phi}$ is the parameters of the classifier; and $n_i$ is the number of data instances for $i$-th class. We obtain the estimator of the noisy label prediction probability $p_t(\tilde{Y} | \mathbf{x}_t)$ as $\tilde{\mathbf{h}}_{\boldsymbol{\phi}}(\mathbf{x}_t, t)$. Additionally, we estimate the noisy label prior $p(\tilde{Y})$ using the statistics of the dataset, i.e., $n_i / n$. Note that the transition matrix $\boldsymbol{S}$ is fixed and independent of $\mathbf{x}_t$, so the inverse matrix operation is required only once during the entire training process.

In practice, the reverse transition matrix, $\boldsymbol{S}$, may not be known. However, there exist methods in the learning with the noisy label community that can estimate the transition matrix $\boldsymbol{T}$, where $T_{i,j} = p(\tilde{Y} = j | Y = i)$, from the noisy labeled dataset (Li et al., 2021; Zhang et al., 2021b). Using the estimated transition matrix and the noisy label prior, we can obtain the reverse transition matrix by applying Bayes' theorem. (See Appendix A.8.) We applied the transition matrix estimation method from VolMinNet (Li et al., 2021) to our time-dependent noisy-label classifier, $\tilde{\mathbf{h}}_{\boldsymbol{\phi}}$, and we analyzed the effect of this estimation in Section 4.5.

### 3.4 PRACTICAL IMPLEMENTATION

**Training Procedure**   Figure 3 illustrates the overall training procedure with the TDSM objective. First, we prepare the time-dependent noisy-label classifier and the transition matrix. The noisy-label classifier $\tilde{\mathbf{h}}_{\boldsymbol{\phi}^*}$ is obtained by training with a combination of cross-entropy losses over different time steps (Song et al., 2020; Chao et al., 2022) on the noisy labeled dataset. We estimate the transition matrix $\boldsymbol{S}$ using prior knowledge or the existing methods discussed in Section 3.3. During the training

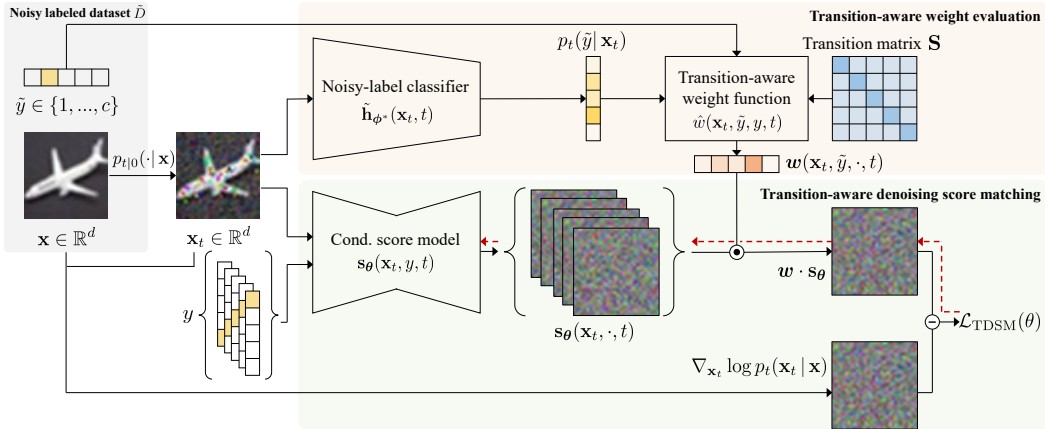

Figure 3: The training procedure of the proposed approach. The solid black arrows indicate the forward propagation, and the dashed red arrows represent the gradient signal flow. The filled circle operation denotes the dot product operation, and the dashed operation represents the L2 loss. The noisy-label classifier $\tilde{\mathbf{h}}_{\phi^*}$ can be obtained by the cross-entropy loss on the noisy labeled dataset $\tilde{D}$.

iterations of the score network, we evaluate the transition-aware weights using Eq. (8). Next, we obtain the score network outputs for all classes. Finally, we compute the TDSM objective value using the weights and score outputs via Eq. (6), and optimize the model using the gradient descent method.

**Reduction on Time and Memory Usage**
The TDSM objective involves network evaluations for all classes, which can cause memory shortages during backpropagation. To mitigate this problem, we only perform backpropagation for the network output corresponding to the given noisy label (lines 7-8 of Algorithm 1). We choose this output corresponding to the given label as it is the dominant term of the transition-aware weight function for most instances.

Furthermore, we observed that the transition-aware weights $w(\mathbf{x}_t, \tilde{y}, y, t)$ have negligible values for most $(\tilde{y}, y)$ pairs. Specifically, on average, only 1.3 elements of the weight vector $\boldsymbol{w}(\mathbf{x}_t, \tilde{y}, \cdot, t)$ have values greater than 0.01 in our experiments on the CIFAR datasets. To reduce the training time of forward propagation, we skip network evaluations for instances, where the corresponding transition-aware weight function value was below a threshold $\tau$ (line 9 of Algorithm 1). We set $\tau$ to 0.01 for all experiments, and Appendix F.9 provides the ablation study of $\tau$. We discuss the training time further in Appendix F.4. Algorithm 1 specifies a training procedure for our method.

---

**Algorithm 1:** Training algorithm with TDSM

**Input:** Noisy labeled dataset $\tilde{D}$, transition matrix $\boldsymbol{S}$, noisy-label classifier $\tilde{\mathbf{h}}_{\phi^*}$, perturbation kernel $p_{t|0}$, temporal weights $\lambda$, skip threshold $\tau$
**Output:** Conditional score model $\mathbf{s}_{\boldsymbol{\theta}}$

1  **while** *not converged* **do**
2      Sample $(\mathbf{x}, \tilde{y})$ from $\tilde{D}$, and time $t$ from $[0, T]$
3      Sample $\mathbf{x}_t$ from the transition kernel $p_{t|0}$
4      Evaluate $w(y) = \hat{w}(\mathbf{x}_t, \tilde{y}, y, t)$ using Eq. (8)
5      $\mathbf{s} = 0$
6      **for** $y \in \{1, ..., c\}$ **do**
7          **if** $y = \tilde{y}$ **then**
8              $\mathbf{s} \leftarrow \mathbf{s} + w(y)\mathbf{s}_{\boldsymbol{\theta}}(\mathbf{x}_t, y, t)$
9          **else if** $w(\mathbf{x}_t, \tilde{y}, y, t) > \tau$ **then**
10             $\mathbf{s} \leftarrow \mathbf{s} + w(y)\mathbf{s}_{\boldsymbol{\theta}}(\mathbf{x}_t, y, t).\text{detach}()$
11         **end**
12     **end**
13     $l \leftarrow \lambda(t)||\mathbf{s} - \nabla_{\mathbf{x}_t} \log p_t(\mathbf{x}_t|\mathbf{x}, \tilde{Y} = \tilde{y})||_2^2$
14     Update $\boldsymbol{\theta}$ by $l$ using the gradient descent method
15 **end**

---

## 4 EXPERIMENTS

We evaluate our method on three benchmark datasets commonly used for both image generation and label noise learning: MNIST (LeCun et al., 2010), CIFAR-10, and CIFAR-100 (Krizhevsky, 2009). We also perform experiments on a real-world noisy labeled dataset, Clothing-1M (Xiao et al., 2015). For the benchmark datasets, we create two types of label noise: symmetric and asymmetric noise (Kaneko et al., 2019). For symmetric noise, we randomly flip the ground-truth label to another

Table 1: Experimental results on the MNIST, CIFAR-10, and CIFAR-100 datasets with various noise settings. The percentages in headers represent the noise rate. 'un' and 'cond' indicate whether a metric is unconditional or conditional. **Bold** numbers indicate better performance.

| | | | | Symmetric | | | | Asymmetric | | | | Clean |
| | | | | 20% | | 40% | | 20% | | 40% | | 0% |
| | | Metric | | DSM | TDSM | DSM | TDSM | DSM | TDSM | DSM | TDSM | DSM |
|---|---|---|---|---|---|---|---|---|---|---|---|---|
| MNIST | un | Density | (↑) | 81.11 | **84.83** | 81.93 | **84.55** | 84.23 | **85.27** | 84.47 | **84.71** | 86.20 |
| | | Coverage | (↑) | 81.23 | **82.16** | **81.65** | 81.31 | 82.30 | **82.45** | 81.97 | **82.27** | 82.90 |
| | cond | CAS | (↑) | 94.31 | **98.22** | 72.52 | **96.49** | 95.25 | **98.22** | 89.29 | **96.54** | 98.55 |
| | | CW-Density | (↑) | 69.78 | **82.99** | 55.70 | **80.09** | 78.58 | **83.74** | 73.54 | **81.65** | 85.79 |
| | | CW-Coverage | (↑) | 76.77 | **80.93** | 70.45 | **79.21** | 79.97 | **81.35** | 77.50 | **80.57** | 82.09 |
| CIFAR-10 | un | FID | (↓) | **2.00** | 2.06 | **2.07** | 2.43 | 2.02 | **1.95** | 2.23 | **2.06** | 1.92 |
| | | IS | (↑) | 9.91 | **9.97** | 9.83 | **9.96** | **10.06** | 10.04 | **10.09** | 10.02 | 10.03 |
| | | Density | (↑) | 100.03 | **106.13** | 100.94 | **111.63** | 100.66 | **104.15** | 101.25 | **105.19** | 103.08 |
| | | Coverage | (↑) | 81.13 | **81.89** | 80.93 | **82.03** | 81.36 | **81.81** | 81.10 | **81.90** | 81.90 |
| | cond | CW-FID | (↓) | 16.21 | **12.16** | 30.45 | **15.92** | 11.97 | **10.89** | 15.18 | **12.54** | 10.23 |
| | | CAS | (↑) | 66.80 | **70.92** | 47.21 | **62.28** | 72.66 | **74.28** | 68.98 | **71.51** | 77.74 |
| | | CW-Density | (↑) | 88.45 | **99.52** | 73.02 | **97.80** | 96.10 | **101.77** | 92.13 | **99.21** | 102.63 |
| | | CW-Coverage | (↑) | 77.80 | **80.29** | 71.63 | **78.65** | 79.95 | **80.99** | 78.12 | **79.98** | 81.57 |
| CIFAR-100 | un | FID | (↓) | **2.96** | 4.26 | **3.36** | 6.85 | 2.76 | **2.64** | **2.73** | 2.81 | 2.51 |
| | | IS | (↑) | 12.28 | **12.29** | 11.86 | **12.07** | 12.49 | **12.79** | 12.51 | **12.57** | 12.80 |
| | | Density | (↑) | 83.01 | **85.66** | 81.70 | **88.45** | 87.36 | **88.41** | 87.06 | 87.01 | 87.98 |
| | | Coverage | (↑) | **75.02** | 74.90 | **73.92** | 72.12 | 77.04 | **77.46** | 76.56 | 76.27 | 77.63 |
| | cond | CW-FID | (↓) | 79.91 | **78.71** | 100.04 | **93.24** | 75.39 | **69.83** | 89.13 | **73.13** | 66.97 |
| | | CAS | (↑) | 25.49 | **28.54** | 15.41 | **21.17** | 33.31 | **37.33** | 23.50 | **34.47** | 39.50 |
| | | CW-Density | (↑) | 66.47 | **70.62** | 49.77 | **60.60** | 72.14 | **78.92** | 60.27 | **74.30** | 82.58 |
| | | CW-Coverage | (↑) | 70.11 | **70.77** | 60.64 | **63.89** | 71.08 | **74.01** | 64.19 | **71.48** | 75.78 |

class, while for asymmetric noise, we flip the ground-truth label to a predefined similar class. We refer to flipping probability as noise rate. Throughout the experiments, we mainly use EDM (Karras et al., 2022) as the backbone of the diffusion models. We provide the results of other backbones in Appendix F.6. Additional experimental settings are provided in Appendices E.1 and E.2.

We evaluate conditional generative models from multiple perspectives using four unconditional metrics[4], including Fréchet Inception Distance (FID) (Heusel et al., 2017), Inception Score (IS) (Salimans et al., 2016), Density, and Coverage (Naeem et al., 2020), and four conditional metrics, namely CW-FID, CW-Density, CW-Coverage (Chao et al., 2022), and Classification Accuracy Score (CAS) (Ravuri & Vinyals, 2019). The Class-Wise (CW) metric evaluates the metric separately for each class and averages them. A detailed description of the metrics is in Appendix E.3.

## 4.1 ANALYSIS ON BENCHMARK DATASET WITH SYNTHETIC LABEL NOISE

Table 1 presents the performance of the baseline models trained with the original DSM objective and our models trained with the TDSM objective on the benchmark datasets with various noise settings. First, the performance of the baseline models decreases on both unconditional and conditional metrics when trained on noisy datasets compared to the clean datasets. This indicates that the label noise in the diffusion model training degrades the sample quality and causes a class mismatch problem.

Second, our models outperform the baseline models in all cases with respect to the conditional metrics. As the noise increases, the performance differences become more significant. We compare the CW metrics evaluated separately for each class in Appendix F.7, which shows that our models perform better than the baselines in most classes. Furthermore, for the unconditional metrics, our models beat the baseline models in most cases. Additionally, Figure 4 provides conditionally generated images from fixed $x_T$ for each model. The images generated by our model have better quality with an accurate class representation of the intended class than those generated by the baseline model.

We also compare the generation performances with the label-noise robust GAN model (Kaneko et al., 2019) in Table 8 of Appendix F.5. The diffusion models generate images much better than the GAN models even in the noisy label dataset, and our models boost the performances by increasing the robustness to label noise.

---

[4]The unconditional metric is calculated from the generated images only, not accounting their labels.

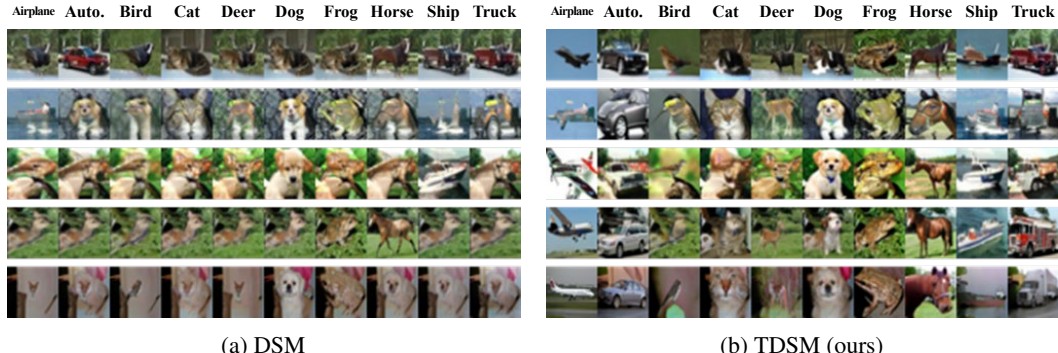

| (a) DSM | (b) TDSM (ours) |

Figure 4: Generated images from (a) baseline and (b) our models, trained on the CIFAR-10 datasets under 40% symmetric noise. Each row contains the samples generated by each class for a fixed $\mathbf{x}_T$.

Table 2: Experimental results on the clean MNIST, CIFAR-10, and CIFAR-100 dataset. The results of conditional metrics are in Table 12.

| Metric | | MNIST | | CIFAR-10 | | CIFAR-100 | |
|---|---|---|---|---|---|---|---|
| | | DSM | TDSM | DSM | TDSM | DSM | TDSM |
| FID | ($\downarrow$) | - | - | 1.92 | **1.91** | **2.51** | 2.67 |
| IS | ($\uparrow$) | - | - | 10.03 | **10.10** | 12.80 | **12.85** |
| Density | ($\uparrow$) | 86.20 | **88.08** | 103.08 | **104.35** | 87.98 | **90.04** |
| Coverage | ($\uparrow$) | 82.90 | **83.69** | 81.90 | **82.07** | 77.63 | **78.28** |

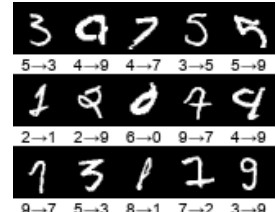

Figure 5: Noisy labels of MNIST, captured by $\hat{w}$. Marks below images denote 'label → prediction'.

## 4.2 ANALYSIS ON BENCHMARK DATASET WITH ANNOTATED LABEL

We evaluate the performance of our model on the benchmark datasets with original annotated labels to explore potentially noisy or ambiguous labels in these datasets. We apply our proposed method using the transition matrix with 5% symmetric noise. Table 2 shows that our label-noise robust models consistently outperform the baseline models. This indicates that existing benchmark datasets may suffer from noisy labels. This phenomenon is well-known in the noisy-label supervised tasks (Bae et al., 2022). To further investigate this observation, we compute the transition-aware weights $\hat{w}(\mathbf{x}_{t_\epsilon}, \bar{y}, y, t_\epsilon) \approx p_{t_\epsilon}(Y = y|\tilde{Y} = \bar{y}, \mathbf{x}_{t_\epsilon})$ at small time $t_\epsilon = 0.01$ for each MNIST train image and label pair $(\mathbf{x}, \bar{y})$. Then, we identify 15 images with the smallest values of $\hat{w}(\mathbf{x}_{t_\epsilon}, \bar{y}, \bar{y}, t_\epsilon)$ and check the predicted label from ours, which is $\arg\max_y \hat{w}(\mathbf{x}_{t_\epsilon}, \bar{y}, y, t)$. Figure 5 shows that the benchmark dataset also contains examples with noisy or ambiguous labels, and this could be the reason why our models improve the performance even when trained on the clean labeled datasets.

## 4.3 ANALYSIS ON REAL-WORLD DATASET

We apply our model to the Clothing-1M dataset, which contains of 1 million clothing images for 14 classes, to evaluate its performance under real-world label noise. This dataset is obtained by crawling images from websites, which introduces label noise, and its label accuracy is 61.54% (Xiao et al., 2015). We also have 25K images containing both noisy and clean labels. We followed the experimental setup from Kaneko et al. (2019). Specifically, we estimate the transition matrix using the statistics of the 25K clean training dataset,

Table 3: Experimental results on the Clothing-1M dataset.

| Metric | DSM | TDSM |
|---|---|---|
| FID ($\downarrow$) | 6.67 | **4.94** |
| CAS ($\uparrow$) | 46.52 | **47.79** |

and we resize the images to a resolution of 64×64. Table 3 shows that our model improves both FID and CAS metrics compared to the baseline model. We did not use other metrics, such as CW-FID, due to the limited amount of clean labeled data for each class (Kaneko et al., 2019). We attach the generated images of the Clothing-1M dataset in Appendix F.16. These results indicate that our model performs well even with real-world label noise.

Table 4: Experimental results of combining with the noisy label corrector, DISC (Li et al., 2023), on the CIFAR-100 under 40% noise.

| Metric | | Symmetric | | Asymmetric | |
|---|---|---|---|---|---|
| | | DSM | TDSM | DSM | TDSM |
| un — FID | (↓) | **2.54** | 2.84 | 4.00 | **3.41** |
| un — IS | (↑) | 12.80 | **12.94** | 12.51 | **12.83** |
| un — Density | (↑) | 87.28 | **90.20** | 83.65 | **88.10** |
| un — Coverage | (↑) | 77.44 | **77.63** | 75.94 | **77.57** |
| cond — CW-FID | (↓) | 67.52 | **67.33** | 78.93 | **76.62** |
| cond — CAS | (↑) | 42.15 | **42.39** | 39.60 | **39.72** |
| cond — CW-Density | (↑) | 82.04 | **85.44** | 76.04 | **81.69** |
| cond — CW-Coverage | (↑) | 75.20 | **75.61** | 70.39 | **71.62** |

Table 5: Ablation study of weight functions on the CIFAR-10 dataset under 40% symmetric noise.

| Metric | | DSM | $S$-DSM | TDSM | |
|---|---|---|---|---|---|
| | | | | $w_S$ | $w_{\hat{S}}$ |
| un — FID | (↓) | 2.07 | 3.20 | 2.43 | 2.32 |
| un — IS | (↑) | 9.83 | 9.92 | 9.96 | 9.97 |
| un — Density | (↑) | 100.94 | 118.85 | 111.63 | 110.67 |
| un — Coverage | (↑) | 80.93 | 81.27 | 82.03 | 82.23 |
| cond — CW-FID | (↓) | 30.45 | 16.26 | 15.92 | 15.83 |
| cond — CAS | (↑) | 47.21 | 63.46 | 62.28 | 61.37 |
| cond — CW-Density | (↑) | 73.02 | 107.24 | 97.80 | 97.59 |
| cond — CW-Coverage | (↑) | 71.63 | 78.32 | 78.65 | 78.68 |

## 4.4 COMBINING WITH THE EXISTING NOISY LABEL CORRECTOR

The existing classifiers to mitigate the noisy label can be considered as finding the true label after noise filtering. By pipelining this noisy label corrector and our TDSM approach, we can find a better noise filtering in terms of generation performance. From this perspective, our approach is orthogonal to existing supervised learning methods for noisy labels, offering an integration of both paradigms. To validate this premise, we conducted experiments by 1) obtaining the corrected labels from the existing classifier learning methods with noisy labels, VolMinNet (Li et al., 2021) and DISC (Li et al., 2023), and 2) training the diffusion model with the corrected labels. The results are summarized in Table 4, and Tables 10 and 11 of Appendix F.8 for the full set of results. The experimental results indicate that applying our TDSM objective consistently improves the performance even with the corrected labels. Therefore, we believe that our approach tackles the noisy label problem from a diffusion model learning perspective, providing an orthogonal direction compared to conventional noisy label methods.

## 4.5 ABLATION STUDY OF TRANSITION-AWARE WEIGHTS

**Estimating the Transition Matrix $S$** So far, we have assumed that the transition matrix $S$ is given. To demonstrate the practical usefulness of our model, we combine our model with an existing method of classifier learning with noisy label, VolMinNet (Li et al., 2021), to estimate the noisy-label classifier and the transition matrix, simultaneously. (See Appendix E.5 for a detailed procedure.) Subsequently, we obtain the estimated transition matrix $\hat{S}$ and we evaluate the weight function from $\hat{S}$, denoted as $w_{\hat{S}}$. As shown in the $w_{\hat{S}}$ column of Table 5, we observe that the model outperforms the baseline model, even when utilizing the learned transition matrix.

**Instance- and Time-Dependent Transition-Aware Weights** To investigate the effect of instance- and time-dependent transition-aware weights, we train a model with the $S$-weighted DSM objective, where the weights contain instance- and time-independent label transition information. As shown in the $S$-DSM column of Table 5, we observe that the model is still robust to noisy labels compared to the baseline model as it still reflects the label transition information. However, the model performs slightly worse on some metrics than the one trained with instance- and time-dependent weight. Interestingly, the $S$-weighted model performs well on density-related metrics. However, since density metrics are insensitive to mode dropping (Naeem et al., 2020), it is possible that the model ignores some areas that are relatively more affected by label noise due to instance-independent weights. In this case, the performance of other metrics will suffer, but the density metrics will be unaffected.

## 5 CONCLUSION

This paper addresses the problem of noisy labels in conditional diffusion models and proposes a new objective called Transition-aware weighted Denoising Score Matching. This objective utilizes a transition-aware weight function that leverages an instance-wise and time-dependent label transition information specifically tailored to the diffusion model. Experimental results on various datasets and noisy label settings demonstrate that the proposed methods outperform baseline models in both conditional and unconditional performance.

ACKNOWLEDGMENTS

This work was supported by the National Research Foundation of Korea (NRF) grant funded by the Korea government (MSIT) (NRF-2019R1A5A1028324).

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

CONTENTS

# A  PROOFS AND DERIVATIONS

## A.1  DISCUSSION OF REMARK

We provide a detailed derivation for the statement in the Remark.

**Remark.** *Let $\boldsymbol{\theta}^*_{DSM} := \arg\min_{\boldsymbol{\theta}} \mathcal{L}_{DSM}(\boldsymbol{\theta}; \tilde{p}_{data}(\mathbf{X}, \tilde{Y}))$ be the optimal parameters obtained by minimizing the DSM objective. Then, $\mathbf{s}_{\boldsymbol{\theta}^*_{DSM}}(\mathbf{x}_t, y, t) = \nabla_{\mathbf{x}_t} \log p_t(\mathbf{x}_t | \tilde{Y} = y)$ for all $\mathbf{x}_t, y, t$.*

Although it is a direct consequence of previous papers (Vincent, 2011; Song & Ermon, 2019), we start by presenting the theoretical derivation from them. We denote the explicit score matching objective as $\mathcal{L}_{\text{ESM}}(\boldsymbol{\theta}; p_{\text{data}}(\mathbf{X}, Y))$, i.e.,

$$\mathcal{L}_{\text{ESM}}(\boldsymbol{\theta}; p_{\text{data}}(\mathbf{X}, Y))$$
$$:= \mathbb{E}_t\left\{\lambda(t)\mathbb{E}_{y\sim p_{\text{data}}(Y)}\mathbb{E}_{\mathbf{x}_t\sim p_t(\mathbf{X}_t|y)}\left[||s_{\boldsymbol{\theta}}(\mathbf{x}_t, y, t) - \nabla_{\mathbf{x}_t}\log p_t(\mathbf{x}_t|Y=y)||_2^2\right]\right\}. \quad (9)$$

Then, by the previous work (Vincent, 2011; Song & Ermon, 2019), the denoising score matching objective is equivalent to the explicit score matching objective, i.e., for all $p_{\text{data}}(\mathbf{X}, Y)$,

$$\mathcal{L}_{\text{ESM}}(\boldsymbol{\theta}; p_{\text{data}}(\mathbf{X}, Y)) = \mathcal{L}_{\text{DSM}}(\boldsymbol{\theta}; p_{\text{data}}(\mathbf{X}, Y)) + C, \quad (10)$$

where $C$ is a constant that does not depend on $\boldsymbol{\theta}$. Therefore, the optimal points of the two objective functions are the same.

By assuming $Y = \tilde{Y}$, we can apply $\mathcal{L}_{\text{ESM}}$ and $\mathcal{L}_{\text{DSM}}$ to a noisy labeled dataset. Let $\boldsymbol{\theta}^*_{\text{ESM}} := \arg\min_{\boldsymbol{\theta}} \mathcal{L}_{\text{ESM}}(\boldsymbol{\theta}; \tilde{p}_{\text{data}}(\mathbf{X}, \tilde{Y}))$. Then, $s_{\boldsymbol{\theta}^*_{\text{ESM}}} = \nabla_{\mathbf{x}_t}\log p_t(\mathbf{x}_t|\tilde{Y} = y)$. Thus, $s_{\boldsymbol{\theta}^*_{\text{ESM}}} = s_{\boldsymbol{\theta}^*_{\text{DSM}}} = \nabla_{\mathbf{x}_t}\log p_t(\mathbf{x}_t|\tilde{Y} = y)$.

## A.2  PROOF OF THEOREM 1

**Theorem 1.** *Under a class-conditional label noise setting, for all $\mathbf{x}_t, \tilde{y}, t$,*

$$\nabla_{\mathbf{x}_t}\log p_t(\mathbf{x}_t|\tilde{Y} = \tilde{y}) = \sum_{y=1}^c w(\mathbf{x}_t, \tilde{y}, y, t)\nabla_{\mathbf{x}_t}\log p_t(\mathbf{x}_t|Y = y), \quad (5)$$

*where $w(\mathbf{x}_t, \tilde{y}, y, t) := p(Y = y|\tilde{Y} = \tilde{y})\frac{p_t(\mathbf{x}_t|Y=y)}{p_t(\mathbf{x}_t|\tilde{Y}=\tilde{y})}$.*

*Proof.* First, for all $t$, the perturbed distribution $p_t$ satisfies:

$$p_t(\mathbf{x}_t|\tilde{Y} = \tilde{y}) = \sum_{y=1}^c p(Y = y|\tilde{Y} = \tilde{y})p_t(\mathbf{x}_t|Y = y) \quad \forall\mathbf{x}_t, \tilde{y}, \quad (11)$$

under the class-conditional label noise setting (see Eq. 1 in the main paper). It means that the noise label transition does not depend on the timesteps. This is because the label is determined by the data distribution $p_{\text{data}}$. Then, Eq. (5) can be derived as follows.

$$\nabla_{\mathbf{x}_t}\log p_t(\mathbf{x}_t|\tilde{Y} = \tilde{y}) = \frac{\nabla_{\mathbf{x}_t}p_t(\mathbf{x}_t|\tilde{Y} = \tilde{y})}{p_t(\mathbf{x}_t|\tilde{Y} = \tilde{y})} \quad (12)$$

$$= \frac{\sum_{y=1}^c p(Y = y|\tilde{Y} = \tilde{y})\nabla_{\mathbf{x}_t}p_t(\mathbf{x}_t|Y = y)}{p_t(\mathbf{x}_t|\tilde{Y} = \tilde{y})} \quad (\because \text{Eq. (11)}) \quad (13)$$

$$= \sum_{y=1}^c \frac{p(Y = y|\tilde{Y} = \tilde{y})p_t(\mathbf{x}_t|Y = y)}{p_t(\mathbf{x}_t|\tilde{Y} = \tilde{y})} \frac{\nabla_{\mathbf{x}_t}p_t(\mathbf{x}_t|Y = y)}{p_t(\mathbf{x}_t|Y = y)} \quad (14)$$

$$= \sum_{y=1}^c \frac{p(Y = y|\tilde{Y} = \tilde{y})p_t(\mathbf{x}_t|Y = y)}{p_t(\mathbf{x}_t|\tilde{Y} = \tilde{y})}\nabla_{\mathbf{x}_t}\log p_t(\mathbf{x}_t|Y = y) \quad (15)$$

$$= \sum_{y=1}^c w(\mathbf{x}_t, \tilde{y}, y, t)\nabla_{\mathbf{x}_t}\log p_t(\mathbf{x}_t|Y = y) \quad (16)$$

$\square$

### A.3  PROOF OF PROPOSITION 2

**Proposition 2.** *Under a class-conditional label noise setting, $w(\mathbf{x}_t, \tilde{y}, y, t) = p_t(Y = y | \tilde{Y} = \tilde{y}, \mathbf{x}_t)$.*

*Proof.* We derive the property that the transition-aware weight function represents the instance-wise label transition under the class-conditional label noise setting, i.e.,

$$w(\mathbf{x}_t, \tilde{y}, y, t) = p_t(Y = y | \tilde{Y} = \tilde{y}, \mathbf{x}_t). \tag{17}$$

The derivation is based on Bayes' theorem and the class-conditional label noise assumption.

$$
\begin{aligned}
w(\mathbf{x}_t, \tilde{y}, y, t) &= p(Y = y | \tilde{Y} = \tilde{y}) \frac{p_t(\mathbf{x}_t | Y = y)}{p_t(\mathbf{x}_t | \tilde{Y} = \tilde{y})} \\
&= p(Y = y | \tilde{Y} = \tilde{y}) \frac{p(\tilde{Y} = \tilde{y})}{p(Y = y)} \frac{p_t(Y = y | \mathbf{x}_t) p_t(\mathbf{x}_t)}{p_t(\tilde{Y} = \tilde{y} | \mathbf{x}_t) p_t(\mathbf{x}_t)} \\
&= \frac{p(\tilde{Y} = \tilde{y} | Y = y) p(Y = y | \mathbf{x}_t)}{p_t(\tilde{Y} = \tilde{y} | \mathbf{x}_t)} \\
&= \frac{p(\tilde{Y} = \tilde{y} | Y = y, \mathbf{x}_t) p(Y = y | \mathbf{x}_t)}{p_t(\tilde{Y} = \tilde{y} | \mathbf{x}_t)} \quad (\because \text{ conditional indep. of } \tilde{Y} \text{ and } \mathbf{X}_t \text{ given } Y) \\
&= \frac{p(\tilde{Y} = \tilde{y}, Y = y | \mathbf{x}_t)}{p_t(\tilde{Y} = \tilde{y} | \mathbf{x}_t)} \\
&= p_t(Y = y | \tilde{Y} = \tilde{y}, \mathbf{x}_t).
\end{aligned} \tag{18}
$$

$\square$

### A.4  PROOF OF THEOREM 3

**Theorem 3.** *Let $\boldsymbol{\theta}_{TDSM}^* := \arg\min_{\boldsymbol{\theta}} \mathcal{L}_{TDSM}(\boldsymbol{\theta}; \tilde{p}_{data}(\mathbf{X}, \tilde{Y}))$ be the optimal parameters obtained by minimizing the TDSM objective. Then, under a class-conditional label noise setting with an invertible transition matrix, $\mathbf{s}_{\boldsymbol{\theta}_{TDSM}^*}(\mathbf{x}_t, y, t) = \nabla_{\mathbf{x}_t} \log p_t(\mathbf{x}_t | Y = y)$ for all $\mathbf{x}_t, y, t$.*

*Proof.* First, by Remark and Theorem 1, we have the following equation for $\theta_{\text{TDSM}}^*$:

$$\sum_{y=1}^{c} w(\mathbf{x}_t, \tilde{y}, y, t) \mathbf{s}_{\boldsymbol{\theta}_{\text{TDSM}}^*}(\mathbf{x}_t, y, t) = \nabla_{\mathbf{x}_t} \log p_t(\mathbf{x}_t | \tilde{Y} = \tilde{y}) = \sum_{y=1}^{c} w(\mathbf{x}_t, \tilde{y}, y, t) \nabla_{\mathbf{x}_t} \log p_t(\mathbf{x}_t | Y = y), \tag{19}$$

for all $\mathbf{x}_t, \tilde{y}, t$. Let $\boldsymbol{W}^{(t)}(\mathbf{x}_t) = w(\mathbf{x}_t, \cdot, \cdot, t)$ is the $c \times c$ matrix, where $W_{\tilde{y}, y}^{(t)}(\mathbf{x}_t) = w(\mathbf{x}_t, \tilde{y}, y, t)$ for $\tilde{y}, y = 1, ..., c$. Then, Eq. (19) can be written in matrix form:

$$\boldsymbol{W}^{(t)}(\mathbf{x}_t) \Big[ \mathbf{s}_{\boldsymbol{\theta}_{\text{TDSM}}^*}(\mathbf{x}_t, y, t) \Big]_{y=1..c} = \boldsymbol{W}^{(t)}(\mathbf{x}_t) \Big[ \nabla_{\mathbf{x}_t} \log p_t(\mathbf{x}_t | Y = y) \Big]_{y=1,...,c}, \tag{20}$$

for all $\mathbf{x}_t, t$. Then, it is enough to show that $\boldsymbol{W}^{(t)}(\mathbf{x}_t)$ is invertible. This is because, by multiplying the inverse of $\boldsymbol{W}^{(t)}(\mathbf{x}_t)$ for both sides in Eq. (20), we obtain $\mathbf{s}_{\boldsymbol{\theta}_{\text{TDSM}}^*}(\mathbf{x}_t, y, t) = \nabla_{\mathbf{x}_t} \log p_t(\mathbf{x}_t | Y = y)$.

We can express $\boldsymbol{W}^{(t)}(\mathbf{x}_t) = \tilde{\boldsymbol{L}}^{(t)}(\mathbf{x}_t) \boldsymbol{S} \boldsymbol{L}^{(t)}(\mathbf{x}_t)$ where $\tilde{\boldsymbol{L}}^{(t)}(\mathbf{x}_t) := \text{diag}\big(1/p_t(\mathbf{x}_t | \tilde{Y} = \tilde{y})\big)$ and $\boldsymbol{L}^{(t)}(\mathbf{x}_t) := \text{diag}\big(p_t(\mathbf{x}_t | Y = y)\big)$. Then, $\tilde{\boldsymbol{L}}(\mathbf{x}_t)$ and $\boldsymbol{L}(\mathbf{x}_t)$ is invertible since the support of $p_t$ is the whole space. Also, $\boldsymbol{S}$ is invertible due to the assumption of the statement. Thus, $\boldsymbol{W}^{(t)}(\mathbf{x}_t)$ is invertible. $\square$

### A.5  PROOF OF PROPOSITION 4

**Proposition 4.** *Let $\boldsymbol{\theta}_{SDSM}^* := \arg\min_{\boldsymbol{\theta}} \mathcal{L}_{SDSM}(\boldsymbol{\theta}; \tilde{p}_{data}(\mathbf{X}, \tilde{Y}))$ be the optimal parameters obtained by minimizing the $\boldsymbol{S}$-weighted DSM objective. Then, under a class-conditional label noise setting with an invertible transition matrix, $\mathbf{s}_{\boldsymbol{\theta}_{SDSM}^*}(\mathbf{x}_t, y, t)$ differs from $\nabla_{\mathbf{x}_t} \log p_t(\mathbf{x}_t | Y = y)$.*

*Proof.* Similar to the proof of Theorem 3 in Appendix A.4, we have the following equation for $\theta^*_{\text{S-DSM}}$:

$$\sum_{y=1}^{c} p(Y=y|\tilde{Y}=\tilde{y})\mathbf{s}_{\theta^*_{\text{S-DSM}}}(\mathbf{x}_t,y,t) = \nabla_{\mathbf{x}_t}\log p_t(\mathbf{x}_t|\tilde{Y}=\tilde{y}) \tag{21}$$

$$= \sum_{y=1}^{c} w(\mathbf{x}_t,\tilde{y},y,t)\nabla_{\mathbf{x}_t}\log p_t(\mathbf{x}_t|Y=y), \tag{22}$$

for all $\mathbf{x}_t, \tilde{y}, t$. With the same notation in Appendix A.4, Eq. (21) can be written in matrix form:

$$\boldsymbol{S}\Big[\mathbf{s}_{\theta^*_{\text{S-DSM}}}(\mathbf{x}_t,y,t)\Big]_{y=1..c} = \boldsymbol{W}^{(t)}(\mathbf{x}_t)\Big[\nabla_{\mathbf{x}_t}\log p_t(\mathbf{x}_t|Y=y)\Big]_{y=1,\ldots,c}, \tag{23}$$

$$\Rightarrow \Big[\mathbf{s}_{\theta^*_{\text{S-DSM}}}(\mathbf{x}_t,y,t)\Big]_{y=1..c} = \boldsymbol{S}^{-1}\boldsymbol{W}^{(t)}(\mathbf{x}_t)\Big[\nabla_{\mathbf{x}_t}\log p_t(\mathbf{x}_t|Y=y)\Big]_{y=1,\ldots,c}, \tag{24}$$

for all $\mathbf{x}_t, t$. Since $w(\mathbf{x}_t,\tilde{y},y,t) \neq p(Y=y|\tilde{Y}=\tilde{y})$ for label noise setting, $\boldsymbol{S}^{-1}\boldsymbol{W}^{(t)}(\mathbf{x}_t) \neq \boldsymbol{I}$. Thus, $\mathbf{s}_{\theta^*_{\text{S-DSM}}}(\mathbf{x}_t,y,t)$ differs from the clean-label conditional score, $\nabla_{\mathbf{x}_t}\log p_t(\mathbf{x}_t|Y=y)$. □

### A.6 DERIVATION OF THE TDSM WITH OTHER FORMATS OF OBJECTIVE FUNCTION

Assuming a Gaussian perturbation kernel, noise or data samples from the perturbation kernel can be expressed as a linear transformation on the score vector. Since our proposed TDSM is also formed as convex combination of score networks, it can be applied by replacing a single network output with a weighted sum of network outputs.

We provide an example for the derivation of the TDSM with EDM framework we used primarily in our experiments. In EDM (Karras et al., 2022), the perturbed data sample $\mathbf{x}_t$ from $p_t$ is obtained by adding i.i.d. Gaussian noise $\mathbf{n}$ of standard deviation $t$ to the data instance $\mathbf{x}$. They introduce a denoiser network $\mathbf{D}$ that is trained by minimizing the data reconstruction loss:

$$\mathcal{L}_{\text{DSM-RC}}(\boldsymbol{\theta}; p_{\text{data}}(\mathbf{X},Y)) := \mathbb{E}_t\Big\{\lambda(t)\mathbb{E}_{\mathbf{x},y\sim p_{\text{data}}(\mathbf{X},Y)}\mathbb{E}_{\mathbf{n}\sim\mathcal{N}(\mathbf{0}|t^2\mathbf{I})}\Big[\big|\big|\mathbf{D}(\mathbf{x}+\mathbf{n},y,t)-\mathbf{x}\big|\big|_2^2\Big]\Big\}, \tag{25}$$

The optimal denoiser network $\mathbf{D}^*_{\text{DSM}}$ trained by Eq. (25) satisfy:

$$\nabla_{\mathbf{x}_t}\log p_t(\mathbf{x}_t|Y=y) = (\mathbf{D}^*_{\text{DSM}}(\mathbf{x}_t,y,t)-\mathbf{x}_t)/t^2. \tag{26}$$

Then, we can formulate the TDSM objective function for applying Eq. (25) to a noisy labled dataset, called TDSM-RC objective.

$$\mathcal{L}_{\text{TDSM-RC}}(\boldsymbol{\theta}; \tilde{p}_{\text{data}}(\mathbf{X},\tilde{Y}))$$

$$:= \mathbb{E}_t\Big\{\lambda(t)\mathbb{E}_{\mathbf{x},\tilde{y}\sim\tilde{p}_{\text{data}}(\mathbf{X},\tilde{Y})}\mathbb{E}_{\mathbf{n}\sim\mathcal{N}(\mathbf{0}|t^2\mathbf{I})}\Big[\big|\big|\sum_{y=1}^{c} w(\mathbf{x}+\mathbf{n},\tilde{y},y,t)\mathbf{D}(\mathbf{x}+\mathbf{n},y,t)-\mathbf{x}\big|\big|_2^2\Big]\Big\}. \tag{27}$$

By Theorem 1 and Eq. (26), the optimal denoiser network $\mathbf{D}^*_{\text{TDSM}}$ trained by Eq. (27) on a noisy-label data distribution satisfy:

$$\sum_{y=1}^{c} w(\mathbf{x}_t,\tilde{y},y,t)\nabla_{\mathbf{x}_t}\log p_t(\mathbf{x}_t|Y=y) = \nabla_{\mathbf{x}_t}\log p_t(\mathbf{x}_t|\tilde{Y}=\tilde{y})$$

$$= \Big(\sum_{y=1}^{c} w(\mathbf{x}_t,\tilde{y},y,t)\mathbf{D}^*_{\text{TDSM}}(\mathbf{x}_t,y,t)-\mathbf{x}_t\Big)/t^2. \tag{28}$$

Because $\sum_{y=1}^{c} w(\mathbf{x}_t,\tilde{y},y,t)=1$ for all $\mathbf{x}_t,\tilde{y},t$, we can further derive as follows:

$$\sum_{y=1}^{c} w(\mathbf{x}_t,\tilde{y},y,t)\nabla_{\mathbf{x}_t}\log p_t(\mathbf{x}_t|Y=y) = \sum_{y=1}^{c} w(\mathbf{x}_t,\tilde{y},y,t)\Big(\frac{\mathbf{D}^*_{\text{TDSM}}(\mathbf{x}_t,y,t)-\mathbf{x}_t}{t^2}\Big). \tag{29}$$

By following the same derivation as in the proof of Theorem 3, the followings are satisfied:

$$\nabla_{\mathbf{x}_t}\log p_t(\mathbf{x}_t|Y=y) = (\mathbf{D}^*_{\text{TDSM}}(\mathbf{x}_t,y,t)-\mathbf{x}_t)/t^2. \tag{30}$$

Therefore, we can verify that the denoiser network which is trained by the TDSM-RC objective function on the noisy-label data distribution converges to the clean denoiser network.

## A.7 DERIVATION OF THE EQ. 8

We can compute the transition-aware weight function $w$ using the noisy label prediction probability $p_t(\tilde{Y}|\mathbf{x}_t)$ and the transition matrix $\boldsymbol{S}$. To achieve this, we use Bayes' theorem on both the numerator and the denominator:

$$\frac{p_t(\mathbf{x}_t|Y=y)}{p_t(\mathbf{x}_t|\tilde{Y}=\tilde{y})} = \frac{p_t(Y=y|\mathbf{x}_t)p_t(\mathbf{x}_t)p(\tilde{Y}=\tilde{y})}{p_t(\tilde{Y}=\tilde{y}|\mathbf{x}_t)p_t(\mathbf{x}_t)p(Y=y)} = \frac{p(\tilde{Y}=\tilde{y})}{p_t(\tilde{Y}=\tilde{y}|\mathbf{x}_t)}\frac{p_t(Y=y|\mathbf{x}_t)}{p(Y=y)}. \tag{31}$$

Furthermore, because of the class-conditional label noise assumption, we know that:

$$\frac{p_t(\tilde{Y}=i|\mathbf{x}_t)}{p(\tilde{Y}=i)} = \sum_{j=1}^{c} p(Y=j|\tilde{Y}=i)\frac{p_t(Y=j|\mathbf{x}_t)}{p(Y=j)} \text{ for } i=1,...,c. \tag{32}$$

This equation can be written in matrix form using the invertible $\boldsymbol{S}$ as follows:

$$\left[\frac{p_t(\tilde{Y}=i|\mathbf{x}_t)}{p(\tilde{Y}=i)}\right]_{i=1..c} = \boldsymbol{S}\left[\frac{p_t(Y=j|\mathbf{x}_t)}{p(Y=j)}\right]_{j=1..c} \iff \left[\frac{p_t(Y=j|\mathbf{x}_t)}{p(Y=j)}\right]_{j=1..c} = \boldsymbol{S}^{-1}\left[\frac{p_t(\tilde{Y}=i|\mathbf{x}_t)}{p(\tilde{Y}=i)}\right]_{i=1..c}. \tag{33}$$

Then, we can compute the transition-aware weight function $w$ using the noisy label prediction probability $p_t(\tilde{Y}|\mathbf{x}_t)$ and the transition matrix $\boldsymbol{S}$:

$$w(\mathbf{x}_t,\tilde{y},y,t) = \frac{S_{\tilde{y},y}p(\tilde{Y}=\tilde{y})}{p_t(\tilde{Y}=\tilde{y}|\mathbf{x}_t)}\sum_{i=1}^{c}\frac{S_{y,i}^{-1}p_t(\tilde{Y}=i|\mathbf{x}_t)}{p(\tilde{Y}=i)}. \tag{34}$$

## A.8 COMPUTING THE REVERSE TRANSITION MATRIX $\boldsymbol{S}$ FROM THE TRANSITION MATRIX $\boldsymbol{T}$

We would like to clarify that we have defined the transition matrix $\boldsymbol{T}$ where $T_{i,j} = p(\tilde{Y}=j|Y=i)$ and the reverse transition matrix $\boldsymbol{S}$ where $S_{i,j} = p(Y=j|\tilde{y}=i)$. The elements of the reverse transition matrix can be computed as follows:

$$S_{i,j} = p(Y=j|\tilde{Y}=i) = p(\tilde{Y}=i|Y=j)p(Y=j)/p(\tilde{Y}=i) = T_{j,i}p(Y=j)/p(\tilde{Y}=i). \tag{35}$$

If we have the transition matrix $\boldsymbol{T}$ and the noisy label prior $p(\tilde{Y})$, we can compute the clean label prior $p(Y)$ using the following relationship.

$$\text{diag}\left(\left[\frac{1}{p(\tilde{Y}=i)}\right]_{i=1..c}\right)\boldsymbol{T}^{T}\left[p(Y=j)\right]_{j=1..c} = \mathbf{1}_c \tag{36}$$

This equation is based on Eq. (35) and $\sum_{j=1}^{c} S_{i,j} = 1$. Thus, we can compute the reverse transition matrix $\boldsymbol{S}$ from the transition matrix $\boldsymbol{T}$.

# B RELATED WORKS

## B.1 DIFFUSION MODELS

The goal of generative models is to approximate the data distribution $p_{\text{data}}(\mathbf{x}_0)$ to the model distribution $p_{\boldsymbol{\theta}}(\mathbf{x}_0)$. Specifically, likelihood-based generative models define an explicit specification of the distribution, and the objectives of these models are to maximize the likelihood. These models typically define the easy-to-sample prior distribution and generate samples from this prior distribution.

Denoising Diffusion Probabilistic Model (DDPM) defines a fixed noise process, characterized as a Markov chain, that perturbs the data instance by adding Gaussian noise (Sohl-Dickstein et al., 2015; Ho et al., 2020):

$$q(\mathbf{x}_{1:T}|\mathbf{x}_0) := \prod_{t=1}^{T} q(\mathbf{x}_t|\mathbf{x}_{t-1}) \text{ where } q(\mathbf{x}_t|\mathbf{x}_{t-1}) := \mathcal{N}(\mathbf{x}_t; \sqrt{1-\beta_t}\mathbf{x}_{t-1}, \beta_t\mathbf{I}), \tag{37}$$

where $\mathbf{x}_{1:T}$ are latent variables of the same dimensionality as $\mathbf{x}_0$ and $\{\beta_t\}$ be the varaince schedule parameters. Then, the goal of DDPM is to approximate this fixed noise process by a trainable Markov chain with Gaussian transitions, which is called denoising process:

$$p_{\boldsymbol{\theta}}(\mathbf{x}_{0:T}) := p_T(\mathbf{x}_T)\prod_{t=1}^{T} p_{t-1|t}^{\boldsymbol{\theta}}(\mathbf{x}_{t-1}|\mathbf{x}_t) \text{ where } p_{t-1|t}^{\boldsymbol{\theta}}(\mathbf{x}_{t-1}|\mathbf{x}_t) := \mathcal{N}(\mathbf{x}_{t-1}; \boldsymbol{\mu}_{\boldsymbol{\theta}}(\mathbf{x}_t, t), \boldsymbol{\Sigma}_{\boldsymbol{\theta}}(\mathbf{x}_t, t)),$$

(38)

where $p_T(\mathbf{x}_T)$ is the prior distribution, e.g., the standard Gaussian distribution. The training objective of DDPM is to minimize the upper bound of the negative log-likelihood:

$$\mathbb{E}[-\log p_{\boldsymbol{\theta}}(\mathbf{x}_0)] \leq \mathbb{E}_q\bigg[-\log p_{0|1}^{\boldsymbol{\theta}}(\mathbf{x}_0|\mathbf{x}_1) + \sum_{t=2}^{T} D_{\mathrm{KL}}\big(q(\mathbf{x}_{t-1}|\mathbf{x}_t, \mathbf{x}_0) \,||\, p_{t-1|t}^{\boldsymbol{\theta}}(\mathbf{x}_{t-1}|\mathbf{x}_t)\big)$$

$$+ D_{\mathrm{KL}}\big(q(\mathbf{x}_T|\mathbf{x}_0) \,||\, p_T(\mathbf{x}_T)\big)\bigg] \quad (39)$$

For the continuous time spaces, this DDPM process can be described through Stochastic Differential Equations (SDEs) (Song et al., 2021; 2020). Specifically, Eq. (2) represents the forward diffusion process, which is defined by the drift function $\mathbf{f}$ and the volatility function $g$.

$$\mathrm{d}\mathbf{x}_t = \mathbf{f}(\mathbf{x}_t, t)\mathrm{d}t + g(t)\mathrm{d}\mathbf{w}_t, \quad (40)$$

where $\mathbf{w}_t$ denotes the standard Brownian motion and $t \in [0, T]$. The forward process gradually perturbs the data instance $\mathbf{x} = \mathbf{x}_0$ to $\mathbf{x}_T$.

The SDE community also elucidated the existence of a corresponding reverse diffusion process (Anderson, 1982):

$$\mathrm{d}\mathbf{x}_t = [\mathbf{f}(\mathbf{x}_t, t) - g^2(t)\nabla_{\mathbf{x}_t}\log p_t(\mathbf{x}_t)]\mathrm{d}\bar{t} + g(t)\mathrm{d}\bar{\mathbf{w}}_t, \quad (41)$$

where $p_t(\mathbf{x}_t)$ is the probability density of $\mathbf{x}_t$, defined by the forward process, and $\bar{\mathbf{w}}_t$ is the reverse-time standard Brownian motion. The reverse process gradually denoises from $\mathbf{x}_T$ to $\mathbf{x}_0$. We can sample data instances through the reverse process.

In spite that the generation task will require the reverse diffusion process, this process is intractable because a data score $\nabla_{\mathbf{x}_t}\log p_t(\mathbf{x}_t)$ is generally not accessible. Therefore, the diffusion model aims to train the score network to approximate $\nabla_{\mathbf{x}_t}\log p_t(\mathbf{x}_t)$ through the score matching objective function with L2 loss (Song et al., 2020). Like the training objective of DDPM, it is known that the score matching objective is known to be an upper bound on the negative log-likelihood for specific temporal weight functions. This score matching objective is equivalently formulated as the noise prediction (Ho et al., 2020; Dhariwal & Nichol, 2021) or the data reconstruction (Kingma et al., 2021; Karras et al., 2022) objectives.

### B.2 CONDITIONAL DIFFUSION MODELS

Conditional diffusion models have also been developed in order to generate samples that match a desired condition $y$. For the conditional models, we need to approximate the conditional score $\nabla_{\mathbf{x}_t}\log p_t(\mathbf{x}_t|y)$, so we additionally take a condition input to a score network (Dhariwal & Nichol, 2021; Karras et al., 2022). Additionally, there are some approaches to guide conditional generation. One approach decomposes the conditional score using Bayes' theorem, and estimates the gradient of the log posterior probability of label using an auxiliary classifier:

$$\nabla_{\mathbf{x}_t}\log p_t(\mathbf{x}_t|Y = y) = \nabla_{\mathbf{x}_t}\log p_t(\mathbf{x}_t) + \nabla_{\mathbf{x}_t}\log p_t(Y = y|\mathbf{x}_t). \quad (42)$$

This is known as the classifier guidance method (Dhariwal & Nichol, 2021; Chao et al., 2022; Kim et al., 2023). To amplify the effect of the condition, a positive scaling factor can be multiplied to the classifier gradient term.

On the other hand, the classifier-free guidance method (Ho & Salimans, 2021) is also proposed, which is a way to achieve the same effect as the classifier guidance without using a classifier. This method utilizes the score network, which produces both unconditional and conditional scores by

introducing an auxiliary class for the unconditional score. Then, they use a linear combination of the unconditional score $\mathbf{s}_{\boldsymbol{\theta}}(\mathbf{x}_t, t)$ and the conditional score $\mathbf{s}_{\boldsymbol{\theta}}(\mathbf{x}_t, y, t)$ with $\alpha > 0$ for sampling.

$$\mathbf{s}_{\mathrm{CFG}}(\mathbf{x}_t, y, t) \coloneqq (1+\alpha)\mathbf{s}_{\boldsymbol{\theta}}(\mathbf{x}_t, y, t) - \alpha\mathbf{s}_{\boldsymbol{\theta}}(\mathbf{x}_t, t). \tag{43}$$

Conditional diffusion models have been used in various applications by using conditions as labels (Kong et al., 2021; Meng et al., 2022a), text (Nichol et al., 2022; Rombach et al., 2022), or latent representations (Kim et al., 2022b; Preechakul et al., 2022).

### B.3 LEARNING WITH NOISY LABELS

Deep neural networks are known to be vulnerable to overfitting, even when trained on a dataset with random labels (Zhang et al., 2021a). Consequently, the presence of noisy labels in a dataset poses a significant challenge. To mitigate this problem, numerous approaches to classifier learning from noisy labels have been introduced, including sample selection (Cheng et al., 2021; Han et al., 2018; Jiang et al., 2018), label correction (Wang et al., 2021; Zheng et al., 2021; 2020), robust loss (Ghosh et al., 2017; Wang et al., 2019; Zhang & Sabuncu, 2018), and transition matrix estimation (Patrini et al., 2017; Xia et al., 2019; Zhang et al., 2021b).

Specifically, transition matrix estimation methods model the relationship between clean and noisy labels when estimating the transition matrix, which represents the probability that a clean label of a sample is corrupted to another label. Patrini et al. (2017) proposed two loss correction structures: forward and backward correction. They also provided the theoretical analysis of the statistical consistency. To estimate the transition matrix, they use the anchor points that belong to a specific class with probability one. Xia et al. (2019) argued that anchor points are not available in the datasets. To address this problem, they proposed $T$-revision, which revises the learned transition matrix and validates from a noisy validation dataset. Yao et al. (2020) proposed the dual-$T$ estimator by factorizing the product of two easily estimated transition matrices. Zhang et al. (2021b) pointed out the overconfident predictions of neural networks and proposed the new regularization terms which is based on the total variation distance. Li et al. (2021) proposed another regularization term which minimizes the volume of the simplex formed by the transition matrix. Zhu et al. (2021) suggested the new transition matrix estimator based on the clusterability, which implies that nearby instances would have the same label. Cheng et al. (2022) estimated the transition matrix using a forward-backward cycle-consistency regularization.

Mixture proportion estimation has been used to solve weakly supervised learning problems, so it is an area that is closely related to the transition matrix estimation (Scott, 2015).[5] For example, Scott et al. (2013) proposed a consistent estimator from the results of mixture proportion estimation to train the classifier under asymmetric label noise. There have been a number of studies based on mixture fraction estimation, each with assumptions such as anchor set (Liu & Tao, 2015; Scott, 2015), separability (Ramaswamy et al., 2016), linear independence (Yu et al., 2018), and tight posterior bound (Zhu et al., 2023). Our estimation of transition-aware weights differs from the traditional mixture proportion methods in that our weights depend on the diffusion timestep, which requires training a time-dependent noisy label classifier. Additionally, it is important to note that our weight function represents the weights related to the data score relationship, and these values do not directly indicate the proportion of the distribution. Therefore, conventional methods for estimating mixture proportions based on the samples may not be directly applicable.

### B.4 LEARNING GENERATIVE MODELS WITH UNCURATED LABEL DISTRIBUTION

The labeled datasets required for training conditional generative models may suffer from the uncurated label distribution, such as noisy labels and imbalanced labels. Some previous studies have pointed out these problems.

There are a few studies for generative model learning from noisy labels. Thekumparampil et al. (2018); Kaneko et al. (2019) proposed algorithms to overcome the difficulties of training GANs with noisy labels by matching the noisy-label conditional probability with the transition matrix. Specifically, the structure of a generator is similar to the traditional GAN, but they modify the

---

[5]This phenomenon can also be found in Eq. (1) in the main paper, where each noisy-label conditional distribution can be expressed by a mixture of clean-label conditional distributions.

Table 6: Comparison between the label-noise robust GAN (rGAN) (Thekumparampil et al., 2018; Kaneko et al., 2019) and diffusion models (ours).

|  | Thekumparampil et al. (2018); Kaneko et al. (2019) | Ours |
|---|---|---|
| Base model | GAN | Diffusion model |
| Objective | Noisy-label conditional distribution matching | Noisy-label conditional score matching |
| Label transition information | $p(\tilde{y}|y)$ | $p_t(y|\tilde{y}, x_t)$ |
| – Time-dependent label transition | ✗ | ✓ |
| – Instance-wise label transition | ✗ | ✓ |

discriminator structure by incorporating the label transition matrix. When the discriminator is fed the samples generated by the generator and the corresponding labels, the labels are corrupted by the label transition matrix to become noisy labels. Since the given dataset is also a noisy labeled dataset, training the discriminator in this way makes it a match between the noisy labeled datasets. Then, the generator can have a clean label conditional distribution. However, the diffusion model cannot completely mitigate the impact of noisy labels with the transition matrix alone because it targets the gradient of the log-likelihood, which is explained in Section 3. Therefore, we propose to adapt the transition matrix framework to fit the diffusion models. We summarize the differences between the GAN-based models and our diffusion model in Table 6.

In addition to the problem of noisy labels, various efforts have been made to tackle the issue of uncurated label distribution in generative models. For example, (Rangwani et al., 2021; 2022; Qin et al., 2023; Kim et al., 2024) addressed the problem of class imbalance or long-tailed distribution when training generative models. Rangwani et al. (2021) highlighted the problem of long-tailed distributions in conditional GAN training, and they proposed the class balancing regularizer with the theoretical evidence. Rangwani et al. (2022) identified class-specific mode collapse in conditional GANs trained on long-tailed distributions, and they introduced a group spectral regularization to reduce the spectral norms of grouped parameters, as motivated by their analysis. More recently, Qin et al. (2023) aimed to alleviate the class imbalance problem in diffusion models. They proposed a new class-balancing diffusion objective based on theoretical analysis. Also, Kim et al. (2024) proposed time-dependent importance reweighted denoising score matching to address the problem of dataset bias. Similar to these works, we identify challenges of diffusion models under noisy labeled datasets and propose a novel label-noise robust objective based on our theoretical analysis.

## C  GUIDANCE METHODS WITH TRANSITION-AWARE WEIGHTS

In addition to generating conditional samples directly from a conditional score network, there are two methods for guiding conditional generation in diffusion models: classifier guidance (Song et al., 2020; Dhariwal & Nichol, 2021) and classifier-free guidance (Ho & Salimans, 2021). However, these guidance methods also suffer from label noise, as shown in the below experiment. To address this issue, we propose strategies to apply our transition-aware weight function in these methods, which can mitigate the effects of noisy labels.

The classifier guidance method (Song et al., 2020; Dhariwal & Nichol, 2021) utilizes an unconditional diffusion model and a classifier to generate samples that satisfy a certain condition by estimating a conditional score using Eq. (44).

$$\nabla_{\mathbf{x}_t} \log p_t(\mathbf{x}_t|Y = y) = \nabla_{\mathbf{x}_t} \log p_t(\mathbf{x}_t) + \nabla_{\mathbf{x}_t} \log p_t(Y = y|\mathbf{x}_t). \tag{44}$$

However, when dealing with noisy labels, the classifier provides misleading guidance even though the unconditional diffusion model is not affected by noisy labels. In this case, our proposed transition-aware weight function can provide correct guidance using an unconditional diffusion model and a noisy-label classifier. We can use Theorem 1 and Eq. (44) to derive the relationship between the gradient of the log probability of clean and noisy labels:

$$\nabla_{\mathbf{x}_t} \log p_t(\mathbf{x}_t|\tilde{Y} = \tilde{y}) = \sum_{y=1}^{c} w(\mathbf{x}_t, \tilde{y}, y, t) \nabla_{\mathbf{x}_t} \log p_t(\mathbf{x}_t|Y = y), \tag{45}$$

$$\nabla_{\mathbf{x}_t} \log p_t(\mathbf{x}_t) + \nabla_{\mathbf{x}_t} \log p_t(\tilde{Y} = \tilde{y}|\mathbf{x}_t) = \sum_{y=1}^{c} w(\mathbf{x}_t, \tilde{y}, y, t)\{\nabla_{\mathbf{x}_t} \log p_t(\mathbf{x}_t) + \nabla_{\mathbf{x}_t} \log p_t(Y = y|\mathbf{x}_t)\}. \tag{46}$$

Because $\sum_{y=1}^{c} w(\mathbf{x}_t, \tilde{y}, y, t) = 1$ for all $\mathbf{x}_t, \tilde{y}, t$, we have:

$$\nabla_{\mathbf{x}_t} \log p_t(\tilde{Y} = \tilde{y}|\mathbf{x}_t) = \sum_{y=1}^{c} w(\mathbf{x}_t, \tilde{y}, y, t) \nabla_{\mathbf{x}_t} \log p_t(Y = y|\mathbf{x}_t) \text{ for } \tilde{y} = 1, ..., c, \quad (47)$$

$$\nabla_{\mathbf{x}_t} \log p_t(Y = y|\mathbf{x}_t) = \sum_{\tilde{y}=1}^{c} w^*(\mathbf{x}_t, \tilde{y}, y, t) \nabla_{\mathbf{x}_t} \log p_t(\tilde{Y} = \tilde{y}|\mathbf{x}_t) \text{ for } y = 1, ..., c, \quad (48)$$

where $w^*(\mathbf{x}_t, \tilde{y}, y, t)$ is the $(y, \tilde{y})$-th element of the matrix $(w(\mathbf{x}_t, \cdot, \cdot, t))^{-1}$, and $w^*(\mathbf{x}_t, \tilde{y}, y, t)$ is well-defined if the transition matrix $S$ is invertible. Using this derivation, we propose a transition-aware weighted classifier guidance method that can be sampled by a noisy-label classifier:

$$\nabla_{\mathbf{x}_t} \log p_t(\mathbf{x}_t|Y = y) = \nabla_{\mathbf{x}_t} \log p_t(\mathbf{x}_t) + \sum_{\tilde{y}=1}^{c} w^*(\mathbf{x}_t, \tilde{y}, y, t) \nabla_{\mathbf{x}_t} \log p_t(\tilde{Y} = \tilde{y}|\mathbf{x}_t). \quad (49)$$

We can evaluate the weighted sum of gradients by $\nabla_{\mathbf{x}_t} \left\{ \sum_{\tilde{y}=1}^{c} \bar{w}^*(\mathbf{x}_t, \tilde{y}, y, t) \log p_t(\tilde{Y} = \tilde{y}|\mathbf{x}_t) \right\}$, where $\bar{w}^*(\mathbf{x}_t, \tilde{y}, y, t)$ is a detached version of $w^*(\mathbf{x}_t, \tilde{y}, y, t)$. This method can be applied without additional training by utilizing the already trained noisy classifier.

On the other hand, the classifier-free guidance method (Ho & Salimans, 2021) uses unconditional and conditional scores, without a classifier, to achieve the same effect as the classifier guidance method. Therefore, when a training dataset contains noisy labels, we only need to change the training objective of the conditional score network from DSM to TDSM.

We experiment with applying guidance methods on the noisy labeled dataset. We report the performance by changing the hyperparameter that balances the effects of the unconditional score network and the guidance signal. This hyperparameter is known to trade off between unconditional and conditional performance (Chao et al., 2022). Figure 6 observes that our approach generates better conditioned images for the same image quality compared to the baseline.

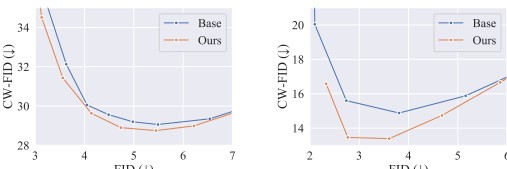

(a) Classifier guidance    (b) Classifier-free guidance

Figure 6: Trade-off between FID and CW-FID on CIFAR-10 dataset under 40% symmetric noise rate.

## D  AFFINE SCORE

Although our model tends to produce more samples that match the given condition, it still produces some mislabeled samples. To address this issue, we propose an affine combination method of the noisy-label conditional score and the clean-label conditional score by leveraging their linear relationship, which is proved by Theorem 1. For each condition $y \in \mathcal{Y}$ and non-negative $\lambda$, we can obtain an equation as follows.

$$(1 + \lambda)\nabla_{\mathbf{x}_t} \log p_t(\mathbf{x}_t|Y = y) - \lambda \nabla_{\mathbf{x}_t} \log p_t(\mathbf{x}_t|\tilde{Y} = y)$$

$$= (1 + \lambda)\nabla_{\mathbf{x}_t} \log p_t(\mathbf{x}_t|Y = y) - \lambda \sum_{j=1}^{c} w(\mathbf{x}_t, j, y, t) \nabla_{\mathbf{x}_t} \log p_t(\mathbf{x}_t|Y = j) \quad (50)$$

$$= \{1 + \lambda(1 - w(\mathbf{x}_t, y, y, t))\}\nabla_{\mathbf{x}_t} \log p_t(\mathbf{x}_t|Y = y) + \sum_{j \neq y}(-\lambda w(\mathbf{x}_t, j, y, t))\nabla_{\mathbf{x}_t} \log p_t(\mathbf{x}_t|Y = j).$$

It should be noted that, for non-negative $\lambda$, the followings are satisfied: $\{1 + \lambda(1 - w(\mathbf{x}_t, y, y, t))\} + \sum_{j \neq y}(-\lambda w(\mathbf{x}_t, j, y, t)) = 1$. Therefore, Eq. (50), called an affine score, is an affine combination of the clean-label conditional scores.

Specifically, this affine score gives more weight to the target label $y$, while exerting the opposite force on the non-target labels. Theoretically, $w$ satisfies $1 - w(\mathbf{x}_t, y, y, t) \leq 0$, and $-\lambda w(\mathbf{x}_t, j, y, t) \leq 0$ for $j \neq y$. We experimentally confirmed that the affine score with the proper $\lambda$ maintained the sample quality and produced almost no conditionally mismatched outliers.

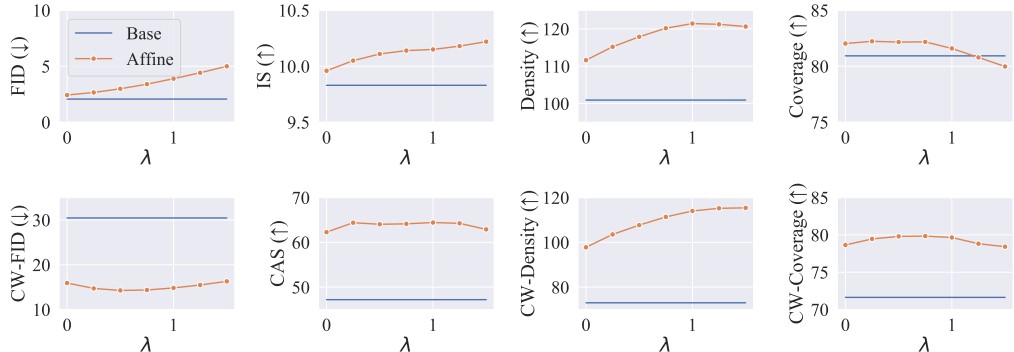

Figure 7: Ablation study of the hyperparameter $\lambda$ in the affine scores using the model trained on the CIFAR-10 datasets under 40% symmetric noise rate.

Figure 7 illustrates the performance of images generated from an affine score by varying the hyperparameter $\lambda$. The results indicate that using an affine score yields better samples across all performance aspects, except for FID. The generated images presented in this paper employ the affine score with $\lambda = 1$.

## E ADDITIONAL EXPERIMENTAL SETTINGS

### E.1 DATASETS

We describe the generation of noisy labeled datasets in more detail. The overall setup follows Kaneko et al. (2019). Specifically, for symmetric noise, labels are randomly perturbed with the probability of the noise rate. For asymmetric noise, we perturbed labels with the probability of the noise rate only for similar classes. In this case, for MNIST, labels are flipped by $2 \rightarrow 7$, $3 \rightarrow 8$, and $5 \leftrightarrow 6$, and for CIFAR-10, labels are flipped by `truck` $\rightarrow$ `automobile`, `bird` $\rightarrow$ `airplane`, `deer` $\rightarrow$ `horse`, `cat` $\leftrightarrow$ `dog`. For CIFAR-100, we have circularly flipped classes in the same superclass.

### E.2 MODEL CONFIGURATIONS

We utilized 8 NVIDIA Tesla P40 GPUs and employed CUDA 11.4 and PyTorch 1.12 versions in our experiments. Our model framework and code are based on EDM (Karras et al., 2022). [6] For all experiments, we used DDPM++ network architecture with a U-net backbone, which is originally proposed by Song et al. (2020) and modified by Karras et al. (2022). The experimental settings align with CIFAR-10 experiments in Karras et al. (2022). The score network was trained with a batch size of 512, and the training iterations were set to 400,000 for MNIST and CIFAR-10, 200,000 for CIFAR-100. For Clothing-1M, the score network is trained with a batch size 256 for 200,000 training iterations.

For the noisy-label classifier, we use shallow convolutional neural network backbones for MNIST, ResNet-18 (He et al., 2016) backbones for CIFAR-10, and ResNet-34 backbones for CIFAR-100 and Clothing-1M. When computing the noisy-label probability, we multiply the softmax output of the classifier and the transition matrix to ensure that the output belongs to the simplex formed by the transition matrix. The classifier was trained by the cross-entropy loss with Adam optimizer using a learning rate of 0.001 and a batch size of 1,024 for 200,000 training iterations. We use the EDM deterministic sampler (Karras et al., 2022) with 18 steps (NFE is 35), which is the default setting of CIFAR-10 in Karras et al. (2022).

---

[6]https://github.com/NVlabs/edm

### E.3 EVALUATION METRICS

We evaluate conditional generative models from multiple perspectives using four unconditional generation metrics, including Fréchet Inception Distance (FID) (Heusel et al., 2017), Inception Score (IS) (Salimans et al., 2016), Density, and Coverage (Naeem et al., 2020), and four conditional generation metrics, namely CW-FID, CW-Density, CW-Coverage (Chao et al., 2022), and Classification Accuracy Score (CAS) (Ravuri & Vinyals, 2019). We use DLSM (Chao et al., 2022) code [7] for evaluations.

The term *unconditional* metric refers to a metric that can measure samples regardless of their labels. We generate samples by performing conditional generation on each class. We use these samples to evaluate both unconditional and conditional metrics on each model. The unconditional metric is calculated from the generated images only, not accounting their labels. The unconditional metric is evaluated in the same way in other papers addressing the conditional generation (Kaneko et al., 2019; Chao et al., 2022).

FID measures the distance between real and generated images in the pre-trained feature space (Szegedy et al., 2016), indicating the fidelity and diversity of generated images. IS quantifies how well the generated images capture diverse classes and whether each image looks like a single class, reflecting image fidelity. The Density and Coverage are reliable versions (Naeem et al., 2020) of Precision and Recall (Sajjadi et al., 2018; Kynkäänniemi et al., 2019), respectively. These two metrics examine how well one image distribution covers the other in the feature space (Szegedy et al., 2016). Higher Density and Coverage values indicate better image quality and diversity, respectively. We do not use Precision and Recall metrics. This is because the nearest neighbor distribution produces an overestimated distribution around the outliers (Naeem et al., 2020), so these metrics are not appropriate in the presence of noisy labels.

To measure how well the generated images match the given conditions, we evaluate metrics on Class-Wise (CW) images. The CW-metric evaluates the metric separately for each class and averages them (Chao et al., 2022). Note that CW-FID, also called intra-FID, is a widely used metric in the field of conditional generative models (Miyato & Koyama, 2018; Kaneko et al., 2019). Previous label-noise robust GAN model (Kaneko et al., 2019) mainly used this metric, and they mentioned that this metric is a measure of the quality of a conditional generative distribution. Additionally, we evaluate CAS, which is the classification accuracy of the test dataset, where the classifier is trained on generated samples. This metric demonstrates that the generated images contain representative class information. The classifier, which is to evaluate CAS metric, is trained by the cross-entropy loss with Adam optimizer using a learning rate of 0.01 with batch size 128 for 200 epochs. The classifiers are shallow convolutional neural networks for MNIST, ResNet-50 for CIFAR-10 and CIFAR-100, and pre-trained ResNet-50 for Clothing-1M.

It should be noted that the MNIST dataset is not suitable for evaluation using these metrics except for CAS, so instead we use density and coverage metrics based on the features of a randomly initialized VGG-16 (Simonyan & Zisserman, 2015), as suggested by Naeem et al. (2020).

### E.4 EXPERIMENTAL SETTING FOR 2-D TOY CASE

We consider a 2-class Gaussian mixture model with $p(\mathbf{x}|y = 1) = \mathcal{N}(\mathbf{x}; (3,3)^T, \mathbf{I})$, $p(\mathbf{x}|y = 2) = \mathcal{N}(\mathbf{x}; (-3,-3)^T, \mathbf{I})$, and $p(y = 1) = p(y = 2) = 0.5$. Also, we set the transition matrix as $\boldsymbol{S} = \left( \begin{smallmatrix} 0.8 & 0.2 \\ 0.2 & 0.8 \end{smallmatrix} \right)$. In this example, we use the VE SDE for the diffusion process, e.g., the conditional distributions of class 1 on time $t$ are: $p_t(\mathbf{x}_t|y = 1) = \mathcal{N}(\mathbf{x}; (3,3)^T, (1+t^2)\mathbf{I})$. Then, we can evaluate $w(\mathbf{x}_t, \tilde{y}, y, t)$ from definition. Note that $w(\mathbf{x}_t, \tilde{y}, y = 1, t) + w(\mathbf{x}_t, \tilde{y}, y = 2, t) = 1$ for all $\mathbf{x}_t, \tilde{y}, t$.

### E.5 ESTIMATING THE TRANSITION MATRIX

We utilized the VolMinNet (Li et al., 2021) framework to obtain the noisy label classifier and the transition matrix in an end-to-end manner. The VolMinNet loss function consists of the sum of the cross-entropy loss for the noisy label using the transition matrix and the loss that minimizes the volume of the simplex formed by the transition matrix. We utilize this loss to train the classifier.

---

[7]https://github.com/chen-hao-chao/dlsm

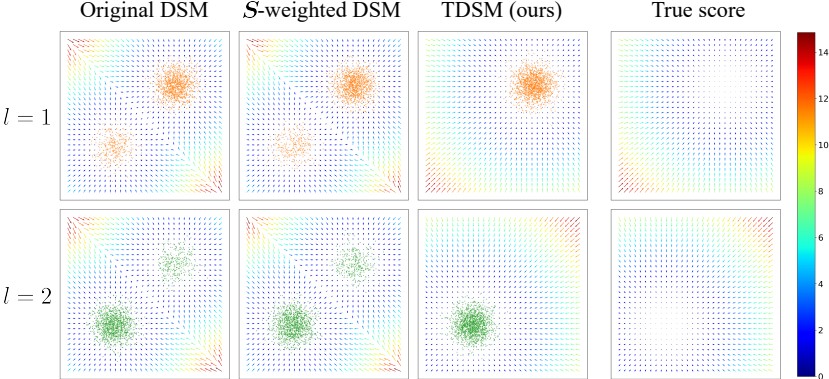

Figure 8: Comparison of the optimal score networks in a 2-dimensional toy case. The dots depict generated samples based on each score network. The arrows represent the score vectors, with the color representing the norm of the vectors.

In this case, we get the estimated transition matrix, and the noisy-label probability is obtained by applying the transition matrix to the classifier output.

To train the transition matrix using the VolMinNet loss, we make the elements of the transition matrix as trainable parameters. The initialization and the optimizer setting of the transition matrix is followed by Li et al. (Li et al., 2021). In particular, the transition matrix was trained for about 7,500 iterations, which is different from the number of iterations used for training the classifier. Therefore, we fix the transition matrix after the matrix training iterations while only the classifier was further trained. Since our noisy-label classifier needs to learn the noisy labels, it requires more iterations. We observed that continuously training of the transition matrix causes it to converge towards the identity matrix.

## F ADDITIONAL EXPERIMENT RESULTS

### F.1 ANALYSIS ON 2-D TOY CASE

Figure 8 visualizes the optimal score networks of DSM variants in a 2-dimensional toy case. We consider a 2-class Gaussian mixture model with $p(\mathbf{x}|y = 1) = \mathcal{N}(\mathbf{x}; (3, 3)^T, \mathbf{I})$, $p(\mathbf{x}|y = 2) = \mathcal{N}(\mathbf{x}; (-3, -3)^T, \mathbf{I})$, and we set the transition matrix as $\boldsymbol{S} = \left( \begin{smallmatrix} 0.8 & 0.2 \\ 0.2 & 0.8 \end{smallmatrix} \right)$. $\boldsymbol{S}$-weighted DSM means that the weight function is determined solely by the transition matrix $\boldsymbol{S}$, i.e., $w'(\mathbf{x}, \tilde{y}, y, t) = p(Y = y|\tilde{Y} = \tilde{y})$. This analysis reveals that the instance-independent transition-aware weights cannot produce a clean-label conditional score in the diffusion models, which is different from the GAN models.

### F.2 REPEATED EXPERIMENTS

we run repeated experiments on the CIFAR-10 dataset with 40% symmetric and asymmetric noise. Figure 9 shows the average values of each metric along the training iterations, with min-max values. We also perform t-tests for all cases, and in most cases we observe a statistically significant improvement over the baselines.

### F.3 LEARNING TRAJECTORY

Figure 10 presents the trajectory of the metrics during the training process. For the conditional metrics, we find that the baseline model does not reach the performance of our early-stage model, even with continued training. Furthermore, for all metrics except FID, our model consistently outperforms the baseline across all snapshots. This observation highlights the persistent efforts of our model in estimating the clean-label conditional distribution and the difficulty of existing models in overcoming label noise.

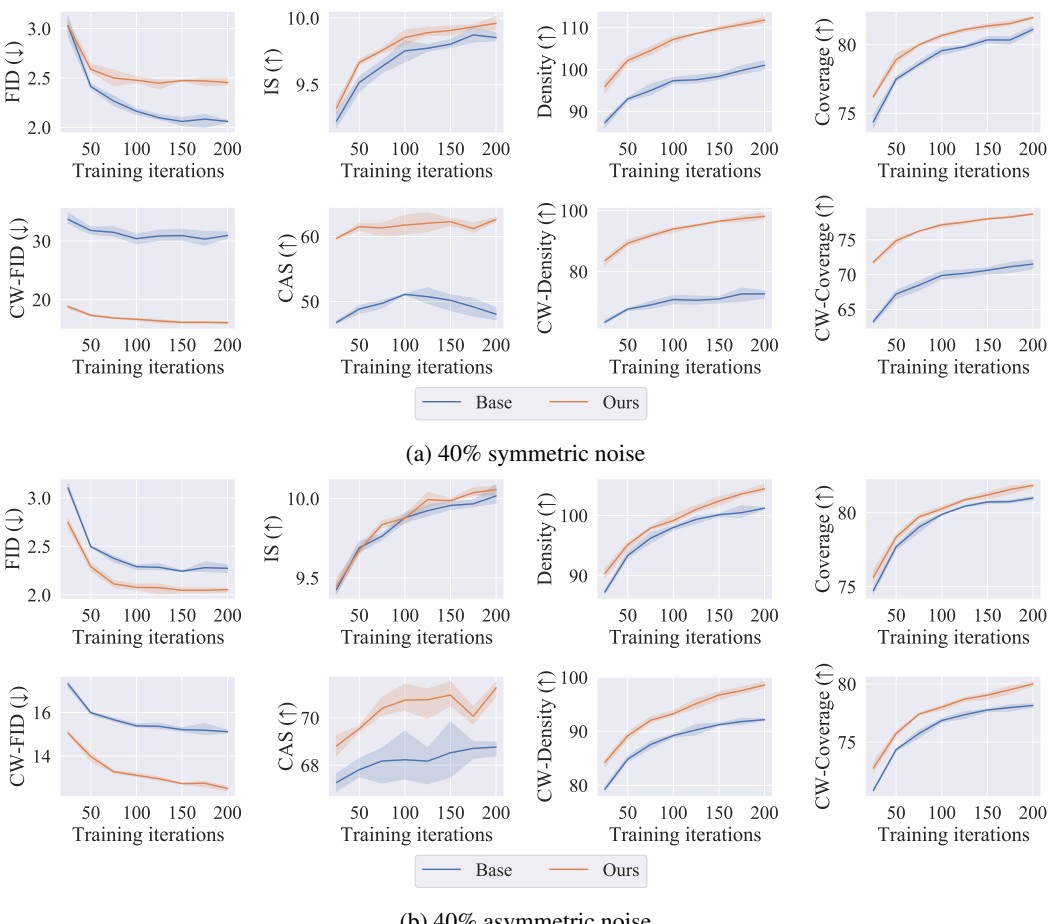

(a) 40% symmetric noise

(b) 40% asymmetric noise

Figure 9: The results of repeated experiments on the CIFAR-10 dataset for each training iteration (every 25 million images). We ran the experiment three times. The solid lines represent the mean of the metrics, and the shaded regions illustrate the minimum and maximum values.

## F.4 TRAINING AND INFERENCE TIME

First, the inference (or sampling) time remains unchanged compared to the baseline model. This is because we only change the training objective to ensure that the score network becomes the clean-label conditional score given a noisy labeled dataset. During inference, the existing sampler is used for sampling, which is unaffected by the introduced weights, thus preserving the inference time.

In the process of evaluating the training objective, it is necessary to compute the weighted sum of the score networks. However, we introduce the practical techniques for reducing on time and memory usage in Section 3.4. In our experiments, Table 7 shows that the training time for our models on CIFAR-10 and CIFAR-100 is only about 1.5 times, not the number of classes times, as long.

In addition, Figure 10 provides a comparison of performance based on training iterations. For the conditional metrics, our models outperform the best baseline model for all training iterations, and for the unconditional metric, we observe that most metrics are similar to or better than the best baseline model's performance around the halfway point of our model's training. Consequently, we can achieve better performance on noisy labeled datasets within the same training time.

## F.5 COMPARISON WITH GAN-BASED MODELS

We compare the generation performances with the label-noise robust GAN model (Kaneko et al., 2019) in Table 8. The diffusion models generate images much better than the GAN model even in

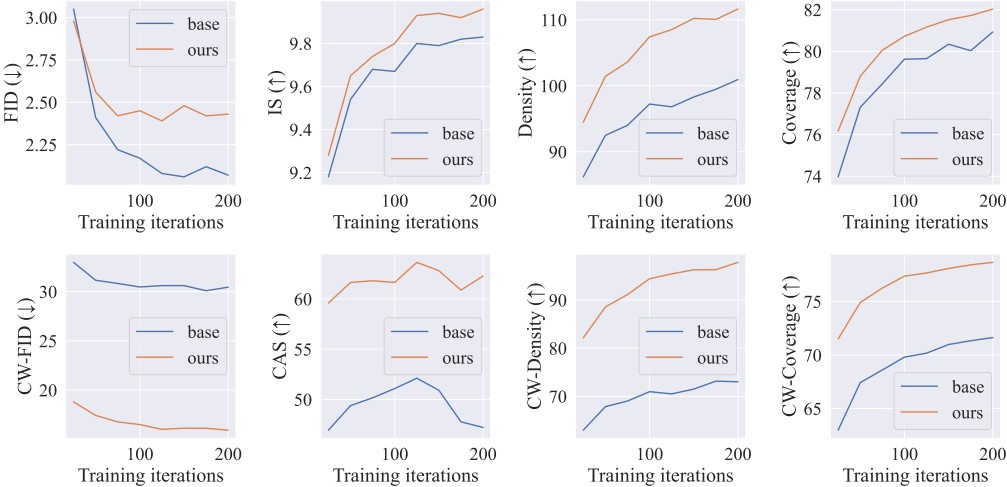

Figure 10: The experiment results on the CIFAR-10 datasets under 40% symmetric noise rate with respect to the training iterations (in every 25 millions of images).

Table 7: Training time on the CIFAR-10 and CIFAR-100 datasets. The values represent the training time (second) per 1K images.

| Dataset | Model | Symmetric | | Asymmetric | |
|---------|-------|-----------|------|------------|------|
| | | 20% | 40% | 20% | 40% |
| CIFAR-10 | DSM | 1.77 | | | |
| | TDSM | 2.21 | 2.43 | 1.85 | 1.90 |
| CIFAR-100 | DSM | 1.78 | | | |
| | TDSM | 2.35 | 3.06 | 2.01 | 2.27 |

the noisy label dataset, and our models boost the performances by increasing the robustness to label noise. Overall, the results demonstrate that our proposed approach is effective in handling label noise in diffusion models, leading to improved image generation performance under noisy label settings.

### F.6 EXPERIMENTAL RESULTS WITH OTHER DIFFUSION FRAMEWORK

Table 9 shows the experimental results with the DDPM++ (Song et al., 2020) framework. Consistent with previous results in Table 1, we observe that our model outperforms the baseline model on all conditional metrics and most unconditional metrics.

### F.7 PERFORMANCE DIFFERENCES FOR THE CLASS-WISE METRICS

Figure 11 shows the performance difference between our models and the baseline models for each class. As shown in the figure, our models outperform in most classes. In particular, we see an improvement in performance even for classes that are not affected by label noise for asymmetric noise cases.

### F.8 ADDITIONAL RESULTS FOR COMBINING WITH THE EXISTING NOISY LABEL CORRECTOR

We conducted experiments by 1) obtaining the corrected labels from the existing classifier learning methods with noisy labels, VolMinNet (Li et al., 2021) and DISC (Li et al., 2023), and 2) training the diffusion model with the corrected labels. We use the same transition matrix structure with a revised noise ratio for training the diffusion model. The results are summarized in Table 10 for VolMinNet and Table 11 for DISC. The experimental results indicate that applying our TDSM objective consistently

Table 8: Experimental results on the CIFAR-10 and CIFAR-100 datasets under 40% noise rate, compared with the GAN-based model. 'GAN' is the baseline GAN model and 'rGAN' is the label-noise robust GAN model.

| | | | Symmetric | | | | Asymmetric | | | |
|---|---|---|---|---|---|---|---|---|---|---|
| | Metric | | GAN | rGAN | DSM | TDSM | GAN | rGAN | DSM | TDSM |
| CIFAR-10 un | FID | ($\downarrow$) | 14.93 | 11.08 | **2.07** | 2.43 | 11.04 | 10.46 | 2.23 | **2.06** |
| | IS | ($\uparrow$) | 8.21 | 8.51 | 9.83 | **9.96** | 8.54 | 8.56 | **10.09** | 10.02 |
| | Density | ($\uparrow$) | 70.02 | 78.90 | 100.94 | **111.63** | 78.06 | 77.47 | 101.25 | **105.19** |
| | Coverage | ($\uparrow$) | 52.51 | 58.92 | 80.93 | **82.03** | 59.19 | 58.45 | 81.10 | **81.90** |
| CIFAR-10 cond | CW-FID | ($\downarrow$) | 53.26 | 27.03 | 30.45 | **15.92** | 29.10 | 26.29 | 15.18 | **12.54** |
| | CAS | ($\uparrow$) | 33.26 | 53.27 | 47.21 | **62.28** | 45.07 | 49.34 | 68.98 | **71.51** |
| | CW-Density | ($\uparrow$) | 49.46 | 71.63 | 73.02 | **97.80** | 68.40 | 70.71 | 92.13 | **99.21** |
| | CW-Coverage | ($\uparrow$) | 44.02 | 56.51 | 71.63 | **78.65** | 55.48 | 55.91 | 78.12 | **79.98** |
| CIFAR-100 un | FID | ($\downarrow$) | 19.46 | 15.09 | **3.36** | 6.85 | 13.81 | 13.96 | **2.73** | 2.81 |
| | IS | ($\uparrow$) | 8.05 | 9.11 | 11.86 | **12.07** | 9.12 | 9.43 | 12.51 | **12.57** |
| | Density | ($\uparrow$) | 69.85 | 75.16 | 81.70 | **88.45** | 79.13 | 78.16 | **87.06** | 87.01 |
| | Coverage | ($\uparrow$) | 45.14 | 50.42 | **73.92** | 72.12 | 53.25 | 52.40 | **76.56** | 76.27 |
| CIFAR-100 cond | CW-FID | ($\downarrow$) | 137.35 | 109.64 | 100.04 | **93.24** | 114.71 | 105.93 | 89.13 | **73.13** |
| | CAS | ($\uparrow$) | 9.89 | 19.84 | 15.41 | **21.17** | 13.84 | 18.08 | 23.50 | **34.47** |
| | CW-Density | ($\uparrow$) | 33.37 | 53.21 | 49.77 | **60.60** | 47.92 | 60.29 | 60.27 | **74.30** |
| | CW-Coverage | ($\uparrow$) | 33.50 | 42.31 | 60.64 | **63.89** | 41.57 | 45.09 | 64.19 | **71.48** |

Table 9: Experimental results for DDPM++ (Song et al., 2020) backbone on the CIFAR-10 dataset with various noise settings. The percentages in headers represent the noise rate. 'un' and 'cond' indicate whether a metric is related to unconditional or conditional generation, respectively. **Bold** numbers indicate better performance.

| | | | Symmetric | | | | Asymmetric | | | |
|---|---|---|---|---|---|---|---|---|---|---|
| | | | 20% | | 40% | | 20% | | 40% | |
| | Metric | | DSM | TDSM | DSM | TDSM | DSM | TDSM | DSM | TDSM |
| un | FID | ($\downarrow$) | **2.11** | 2.24 | **2.27** | 2.61 | 2.20 | **2.07** | 2.44 | **2.24** |
| | IS | ($\uparrow$) | 9.85 | **10.04** | 9.79 | **9.93** | 9.89 | **9.98** | 9.95 | 9.95 |
| | Density | ($\uparrow$) | 103.17 | **110.15** | 101.67 | **110.72** | 104.01 | **106.22** | 104.30 | **106.04** |
| | Coverage | ($\uparrow$) | 81.21 | **82.14** | 80.69 | **81.28** | 81.58 | **81.87** | 81.48 | **81.73** |
| cond | CW-FID | ($\downarrow$) | 16.11 | **11.95** | 29.31 | **15.87** | 12.06 | **10.99** | 15.32 | **12.92** |
| | CAS | ($\uparrow$) | 66.86 | **71.18** | 52.70 | **61.35** | 72.75 | **73.80** | 68.58 | **71.22** |
| | CW-Density | ($\uparrow$) | 91.83 | **103.94** | 74.57 | **98.02** | 99.30 | **103.30** | 95.12 | **100.30** |
| | CW-Coverage | ($\uparrow$) | 78.47 | **80.72** | 71.76 | **78.27** | 80.50 | **81.14** | 78.67 | **79.77** |

improves the performance even with the corrected labels. Therefore, we believe that our approach tackles the noisy label problem from a diffusion model learning perspective, providing an orthogonal direction compared to conventional noisy label methods.

## F.9 ABLATION STUDIES ON SKIP THRESHOLD

Figure 12 demonstrates the trade-off between CW-FID and the training time per 1K images depending on the skip threshold $\tau$, as explained in Section 3.4. Using a small threshold leads to a long training time due to more network evaluations, and using a high threshold leads to a decrease in performance due to inaccurate targets. From this result, we set $\tau$ as 0.01 for all experiments.

## F.10 ADDITIONAL RESULTS FOR THE BENCHMARK DATASET WITH ANNOTATED LABEL

The experiments for Table 2 in Section 4.2 applied our models to benchmark dataset with the conjecture that some annotated labels may be noisy. For the conditional metrics, we need to have true labels, but due to the conjecture, it may not appropriate to use these metrics with the annotated labels.

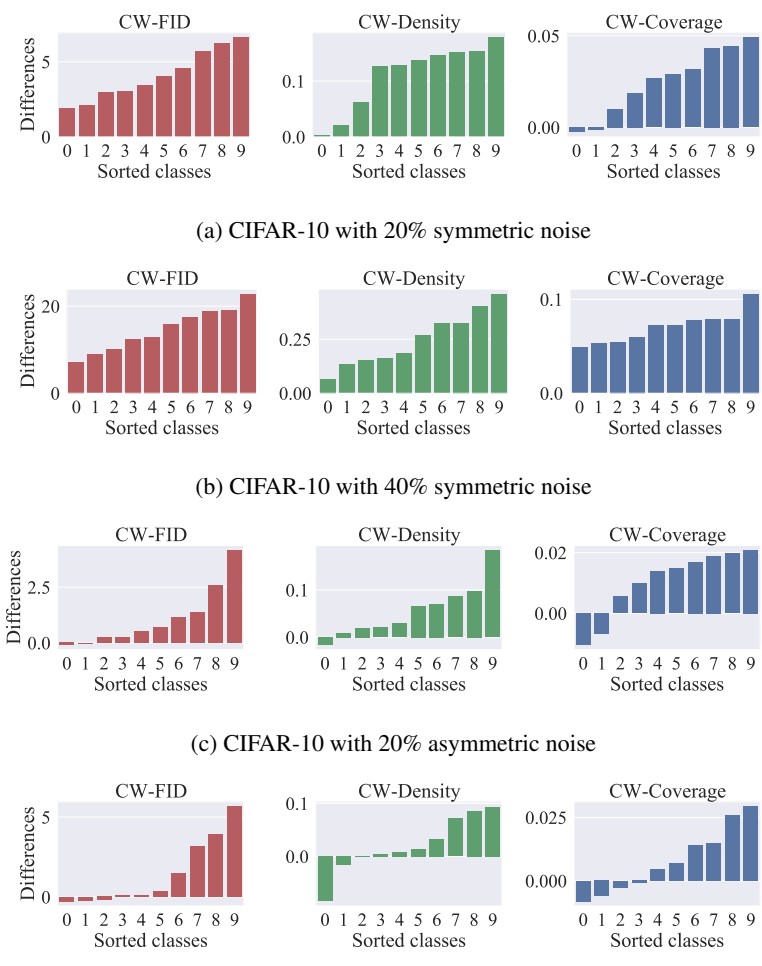

(a) CIFAR-10 with 20% symmetric noise

(b) CIFAR-10 with 40% symmetric noise

(c) CIFAR-10 with 20% asymmetric noise

(d) CIFAR-10 with 40% asymmetric noise

Figure 11: The sorted differences between the DSM (baseline) and TDSM (ours) on the CIFAR-10 dataset with various noise settings.

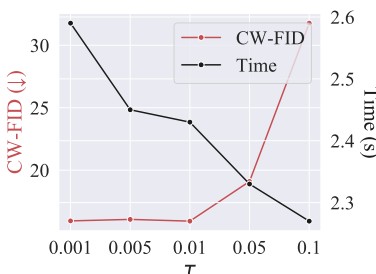

Figure 12: Ablation study of the skip threshold $\tau$.

Nevertheless, it is possible to evaluate the conditional metrics with the annotated labels. The results are shown in Table 12, and we find that our model further improves the intra-class sample quality.

To see the impact of the noise rate of TDSM on the clean dataset, we train TDSM models on the clean CIFAR-10 dataset assuming different symmetric noise rates. As shown in Table 13, the model performs best at a noise rate of 5%. In addition, most TDSM models show improvements compared to

Table 10: Experimental results of combining with the noisy label corrector, VolMinNet (Li et al., 2021), on the CIFAR-10 and CIFAR-100 datasets under various noise settings.

| | | Metric | | Symmetric | | | | Asymmetric | | | | Clean |
| | | | | 20% | | 40% | | 20% | | 40% | | 0% |
| | | | | DSM | TDSM | DSM | TDSM | DSM | TDSM | DSM | TDSM | DSM |
|---|---|---|---|---|---|---|---|---|---|---|---|---|
| CIFAR-10 | un | FID | (↓) | **1.94** | 2.04 | **1.97** | 2.18 | **1.92** | 2.03 | **1.95** | 1.98 | 1.92 |
| | | IS | (↑) | 9.90 | **10.02** | 9.90 | **10.09** | 9.98 | 9.98 | 9.96 | **9.99** | 10.03 |
| | | Density | (↑) | 101.32 | **104.83** | 103.26 | **110.09** | 102.15 | **103.78** | 102.33 | **102.98** | 103.08 |
| | | Coverage | (↑) | 81.25 | **82.04** | 81.45 | **82.30** | 81.65 | **81.79** | 81.67 | **81.98** | 81.90 |
| | cond | CW-FID | (↓) | 10.57 | **10.56** | 11.42 | **10.88** | 10.33 | **10.26** | 10.62 | **10.61** | 10.23 |
| | | CAS | (↑) | 75.85 | **75.96** | 73.32 | **73.38** | 75.63 | **76.54** | 76.27 | **76.53** | 77.74 |
| | | CW-Density | (↑) | 99.92 | **103.61** | 99.03 | **107.22** | 100.86 | **102.62** | 100.75 | **101.37** | 102.63 |
| | | CW-Coverage | (↑) | 80.65 | **81.48** | 80.25 | **81.46** | 81.09 | **81.31** | 80.94 | **81.10** | 81.57 |
| CIFAR-100 | un | FID | (↓) | **2.62** | 3.79 | **2.85** | 3.76 | 2.51 | **2.46** | 3.74 | **3.24** | 2.51 |
| | | IS | (↑) | 12.53 | **12.63** | 12.46 | **12.70** | 12.81 | **12.90** | 12.52 | **12.86** | 12.80 |
| | | Density | (↑) | 87.20 | **87.68** | 85.01 | **91.76** | 86.43 | **88.63** | 84.77 | **87.97** | 87.98 |
| | | Coverage | (↑) | 77.27 | **77.29** | 76.40 | **76.55** | 77.58 | **78.31** | 76.67 | **77.76** | 77.63 |
| | cond | CW-FID | (↓) | 68.69 | **67.96** | 72.17 | **71.18** | 67.43 | **66.96** | 75.05 | **73.06** | 66.97 |
| | | CAS | (↑) | 36.08 | **38.38** | 35.32 | **36.47** | 39.64 | **40.46** | 38.84 | **40.33** | 39.50 |
| | | CW-Density | (↑) | 79.73 | **80.11** | 74.58 | **82.39** | 80.72 | **82.74** | 77.50 | **80.65** | 82.58 |
| | | CW-Coverage | (↑) | 75.04 | **75.51** | 72.65 | **73.24** | 75.34 | **76.12** | 72.68 | **73.42** | 75.78 |

Table 11: Experimental results of combining with the noisy label corrector, DISC (Li et al., 2023), on the CIFAR-100 datasets under various noise settings.

| | | Metric | | Symmetric | | | | Asymmetric | | | | Clean |
| | | | | 20% | | 40% | | 20% | | 40% | | 0% |
| | | | | DSM | TDSM | DSM | TDSM | DSM | TDSM | DSM | TDSM | DSM |
|---|---|---|---|---|---|---|---|---|---|---|---|---|
| CIFAR-100 | un | FID | (↓) | **2.47** | 2.65 | **2.54** | 2.84 | 2.51 | **2.45** | 4.00 | **3.41** | 2.51 |
| | | IS | (↑) | 12.69 | **12.83** | 12.80 | **12.94** | 12.75 | **12.96** | 12.51 | **12.83** | 12.80 |
| | | Density | (↑) | 87.89 | **89.91** | 87.28 | **90.20** | 87.79 | **90.52** | 83.65 | **88.10** | 87.98 |
| | | Coverage | (↑) | 77.61 | **78.08** | 77.44 | **77.63** | 77.62 | **78.42** | 75.94 | **77.57** | 77.63 |
| | cond | CW-FID | (↓) | 67.10 | **66.82** | 67.52 | **67.33** | 67.32 | **66.68** | 78.93 | **76.62** | 66.97 |
| | | CAS | (↑) | 39.19 | **41.83** | 42.15 | **42.39** | 41.40 | **41.90** | 39.60 | **39.72** | 39.50 |
| | | CW-Density | (↑) | 82.62 | **84.94** | 82.04 | **85.44** | 82.58 | **85.52** | 76.04 | **81.69** | 82.58 |
| | | CW-Coverage | (↑) | 75.70 | **76.54** | 75.20 | **75.61** | 75.44 | **76.00** | 70.39 | **71.62** | 75.78 |

DSM. This suggests that the consideration of label transitions over timesteps contributes to improved performance even on clean datasets. However, it is important to note that additional research is needed to account for the actual noise labels present in the benchmark dataset because our experiments assume symmetric noise.

## F.11 Experimental results on severe noise rate

To verify the robustness of the proposed model under extreme noise, we perform the experiments on the CIFAR-10 dataset with severe symmetric noise rates of 60% and 80%. The results are presented in Table 14. These results show that TDSM consistently improves performance even at severe noise rates.

## F.12 Further analysis of the transition-aware weighted function

In this subsection, we clarify the meaning of the transition-aware weighted function $w$. Label recovery is an intuitive way to overcome the problem of noisy labels in a dataset. Our method does not specifically focus on robustness to noisy labels through label recovery; instead, it focuses on robustness to noisy labels from a diffusion model training perspective. Consequently, our approach can be synergistically combined with existing label recovery methods to further improve performance.

Table 12: Full experimental results on the clean MNIST, CIFAR-10, and CIFAR-100 dataset.

| | Metric | | MNIST | | CIFAR-10 | | CIFAR-100 | |
|---|---|---|---|---|---|---|---|---|
| | | | Base | Ours | Base | Ours | Base | Ours |
| un | FID | ($\downarrow$) | - | - | 1.92 | **1.91** | **2.51** | 2.67 |
| | IS | ($\uparrow$) | - | - | 10.03 | **10.10** | 12.80 | **12.85** |
| | Density | ($\uparrow$) | 86.20 | **88.08** | 103.08 | **104.35** | 87.98 | **90.04** |
| | Coverage | ($\uparrow$) | 82.90 | **83.69** | 81.90 | **82.07** | 77.63 | **78.28** |
| cond | CW-FID | ($\downarrow$) | - | - | 10.23 | **10.18** | 66.97 | **66.68** |
| | CAS | ($\uparrow$) | **98.55** | 98.50 | **77.74** | 77.07 | **39.50** | 39.10 |
| | CW-Density | ($\uparrow$) | 85.79 | **87.96** | 102.63 | **103.69** | 82.58 | **84.96** |
| | CW-Coverage | ($\uparrow$) | 82.09 | **82.98** | 81.57 | **81.88** | 75.78 | **76.53** |

Table 13: Experimental results on the clean CIFAR-10 dataset varying the noise rate parameters in TDSM. The percentage in the header indicates the assumed noise rate.

| Metric | | DSM | TDSM | | | | |
|---|---|---|---|---|---|---|---|
| | | | 1% | 2.5% | 5% | 7.5% | 10% |
| FID | ($\downarrow$) | 1.92 | 1.94 | 1.93 | **1.91** | 2.02 | 2.00 |
| IS | ($\uparrow$) | 10.03 | 10.08 | 10.04 | **10.10** | 10.04 | 10.06 |
| Density | ($\uparrow$) | 103.08 | 103.83 | 104.07 | 104.35 | **105.44** | 105.15 |
| Coverage | ($\uparrow$) | 81.90 | 81.79 | 81.90 | **82.07** | 82.04 | 81.95 |

It is important to analyze how our method overcomes the noisy label robustness in diffusion models. Our analysis indicates that our proposed TDSM objective provides the diffusion model with information about the diffused label noise in the dataset. This information is represented by the transition-aware weight function $w$, depending upon the diffusion time. Therefore, we estimate the instance- and time-dependent label transition probability with the transition-aware weight function.

We want to emphasize that the transition-aware weight function $w$ plays a different role from a typical classifier. This probability represents the relationship between the noisy and clean label conditional scores (which are also time-dependent) and is an important element of the proposed training objective of a diffusion model. Therefore, we need to estimate this label transition probabilities over diffusion timesteps.

Specifically, our transition-aware weight function can be reformulated as follows:

$$w(\mathbf{x}_t, \tilde{y}, y, t) = \frac{p_t(Y = y|\mathbf{x}_t)p(\tilde{Y} = \tilde{y}|Y = y)}{p_t(\tilde{Y} = \tilde{y}|\mathbf{x}_t)}. \tag{51}$$

As seen in the expression, the weight function is composed of the clean label classifier $p_t(Y = y|\mathbf{x}_t)$, label transition prior $p(\tilde{Y} = \tilde{y}|Y = y)$, and the noisy label classifier $p_t(\tilde{Y} = \tilde{y}|\mathbf{x}_t)$. Given the triplet of perturbed data instance and its corresponding noisy labels and timestep, i.e., $(\mathbf{x}_t, \tilde{y}, t)$, each component of $w$ has the following characteristics with respect to $y$: 1) $p(Y = y|\mathbf{x}_t)$ is maximized when $y$ is the clean label of the clean data $\mathbf{x}_0$ of $\mathbf{x}_t$; 2) $p(\tilde{Y} = \tilde{y}|Y = y)$ is maximized when $y$ is the noisy label $\tilde{y}$ in general. The reason for 2) is that in general, a given noisy label is sufficiently likely to be a clean label. These two trade-offs imply that the $w$ function does not behave like a clean label classifier.

Furthermore, for large enough $t$, the distribution of $\mathbf{x}_t$ converges to a label-independent prior distribution by the design of the diffusion process, so that $p_t(Y|\mathbf{x}_t)$ converges to a uniform distribution. Therefore, for sufficiently large $t$, the $w$ function converges to a transition matrix $\boldsymbol{S}$. This phenomenon can also be demonstrated in Figure 2 of the 2-D Gaussian mixture model example. In summary, the transition-aware weight function needed to overcome noisy labels in a diffusion model is not represented clean label recovery information only, and this function has time-varying information.

Table 14: Experimental results on the CIFAR-10 datasets with symmetric noise containing severe noise rates. The percentages in headers represent the noise rate. 'un' and 'cond' indicate whether a metric is unconditional or conditional. **Bold** numbers indicate better performance.

|  | Metric | | 20% | | 40% | | 60% | | 80% | |
|---|---|---|---|---|---|---|---|---|---|---|
|  |  |  | DSM | TDSM | DSM | TDSM | DSM | TDSM | DSM | TDSM |
| un | FID | ($\downarrow$) | **2.00** | 2.06 | **2.07** | 2.43 | **2.17** | 2.55 | **2.22** | 2.44 |
|  | IS | ($\uparrow$) | 9.91 | **9.97** | 9.83 | **9.96** | 9.75 | **10.09** | 9.73 | **9.79** |
|  | Density | ($\uparrow$) | 100.03 | **106.13** | 100.94 | **111.63** | 102.17 | **110.78** | 102.09 | **103.69** |
|  | Coverage | ($\uparrow$) | 81.13 | **81.89** | 80.93 | **82.03** | 81.00 | **81.65** | 80.81 | **81.15** |
| cond | CW-FID | ($\downarrow$) | 16.21 | **12.16** | 30.45 | **15.92** | 48.94 | **23.36** | 72.51 | **53.57** |
|  | CAS | ($\uparrow$) | 66.80 | **70.92** | 47.21 | **62.28** | 29.72 | **52.38** | 9.93 | **24.68** |
|  | CW-Density | ($\uparrow$) | 88.45 | **99.52** | 73.02 | **97.80** | 58.42 | **86.58** | 43.97 | **54.30** |
|  | CW-Coverage | ($\uparrow$) | 77.80 | **80.29** | 71.63 | **78.65** | 61.88 | **75.08** | 44.53 | **56.47** |

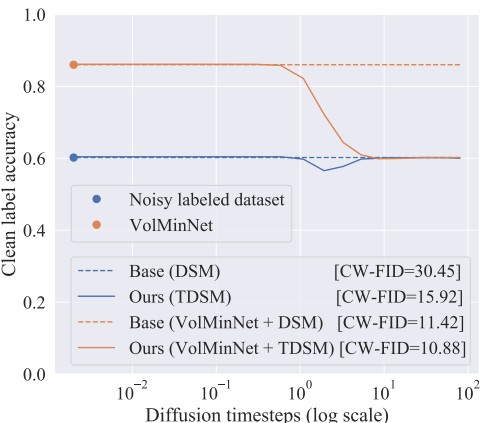

Figure 13: The clean label accuracy of the transition-aware weight function over diffusion timesteps on the CIFAR-10 training datasets under 40% symmetric noise.

It is possible to check the accuracy of the transition-aware weight function on the training dataset since the function provides the predicted probability of a clean label given a noisy label and an instance. Figure 13 shows the clean label accuracy of the transition-aware weight function over diffusion timesteps. Since the noisy labels in the dataset are used as the input conditions of the baseline model for the entire diffusion timesteps, we plot the percentages of clean labels in the datasets as dots at $t = 0$ and plot a dashed horizontal line for comparison.

Focusing on 'Ours (VolMinNet + TDSM)', which provides the best generation performance, we found that the behavior of the $w$ function varies with diffusion timesteps, as mentioned above. In particular, as $t$ increases, the clean label accuracy converges to the clean rate of the given dataset, which is 0.6. This is because the $w$ function converges to the label transition prior $p(\tilde{Y} = \tilde{y}|Y = y)$. 'Base (VolMinNet + DSM)', which does not take this into account, learns the diffusion model only with corrected labels, leading to the performance difference. Therefore, while existing noisy label methods that focus on clean label recovery certainly contribute significantly to generative performance, considering the TDSM objective in diffusion model training enables additional performance improvement independent of clean label recovery.

### F.13 ANALYSIS OF THE NOISY LABEL CLASSIFIER

To investigate the effect of the noisy label classifier, we train the diffusion models with different training levels of noisy label classifiers. To analyze the noisy label classifiers, we measured 1) the noisy label classification performance of the noisy label classifier over diffusion timesteps; and 2) the

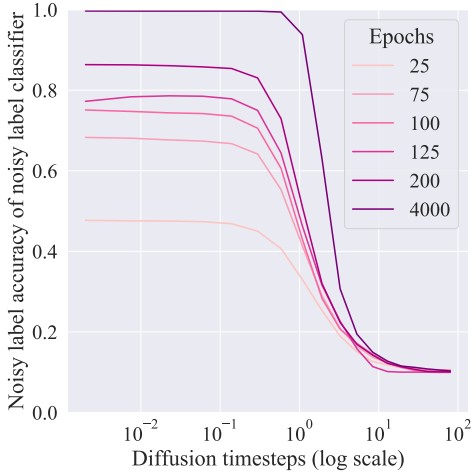 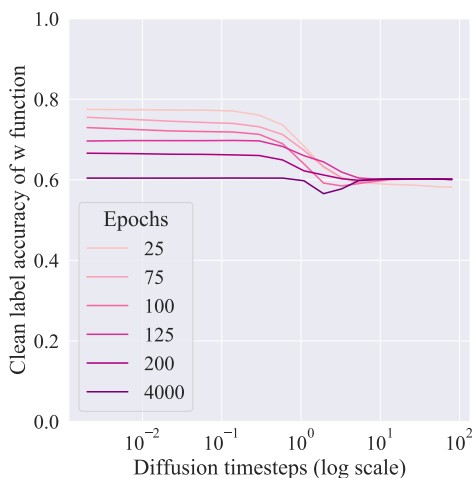

(a) Noisy label accuracy of noisy label classifier

(b) Clean label accuracy of $w$ function

Figure 14: Classification performance of the noisy label classifier with varying training levels over diffusion timesteps on the CIFAR-10 dataset with 40% symmetric noise. The numbers in the legend represent the training epochs of the noisy label classifier.

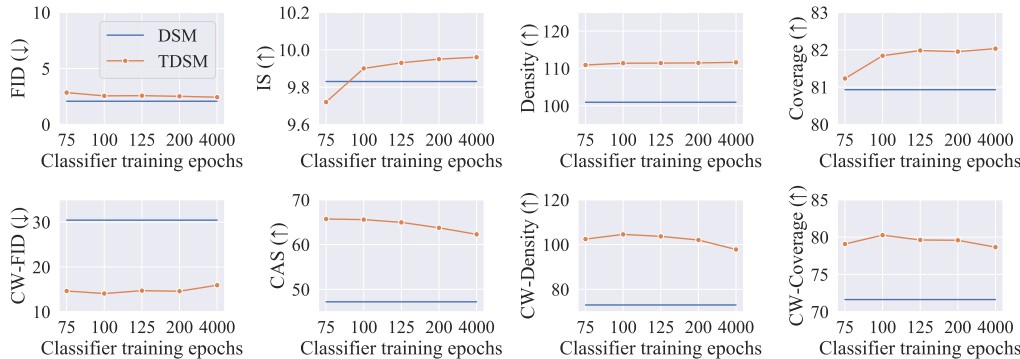

Figure 15: Generation performance of TDSM models on the CIFAR-10 dataset with 40% symmetric noise with respect to the training epochs of the noisy label classifier. DSM is independent of the classifier, but is shown as a horizontal line for comparison.

clean label classification performance of the $w$ function, evaluated with the classifier, over diffusion timesteps, in Figure 14. We also compare the generation performance for the diffusion model trained with each classifier in Figure 15 and Table 15.

The performance of the noisy label classifier in predicting noisy labels increased with the number of epochs trained on the classifier, but its ability to predict clean labels in clean data space decreased. Meanwhile, in terms of the generation performance of the diffusion model, the less trained classifier performed better with respect to the conditional metrics, and the more trained classifier performed better with respect to the unconditional metrics. Note that all TDSM models outperform the baseline, DSM, on most metrics.

We interpret this result as follows. 1) Due to the iterative sampling procedure of the diffusion model, the noisy label classifier needs to learn enough of all the perturbed samples of the diffusion timesteps to improve the quality of the generated samples. 2) At this time, excessive training of the noisy label classifier leads to an overconfidence problem, particularly in the data space at $t = 0$. This overconfidence reduces the information of the clean label and affects the information of the condition.

Table 15: Generation performance of TDSM models with different training levels of noisy label classifiers on the CIFAR-10 dataset with 40% symmetric noise. The epochs represent the training epochs of the noisy label classifier.

| | Metric | | DSM | TDSM | | | | |
|---|---|---|---|---|---|---|---|---|
| | | | | 75 epochs | 100 epochs | 125 epochs | 200 epochs | 4000 epochs |
| un | FID | (↓) | **2.07** | 2.84 | 2.55 | 2.57 | 2.51 | 2.43 |
| | IS | (↑) | 9.83 | 9.72 | 9.90 | 9.93 | 9.95 | **9.96** |
| | Density | (↑) | 100.94 | 110.92 | 111.40 | 111.43 | 111.48 | **111.63** |
| | Coverage | (↑) | 80.93 | 81.23 | 81.84 | 81.98 | 81.95 | **82.03** |
| cond | CW-FID | (↓) | 30.45 | 14.63 | **14.06** | 14.70 | 14.58 | 15.92 |
| | CAS | (↑) | 47.21 | **65.70** | 65.57 | 64.95 | 63.74 | 62.28 |
| | CW-Density | (↑) | 73.02 | 102.43 | **104.54** | 103.65 | 102.03 | 97.80 |
| | CW-Coverage | (↑) | 71.63 | 79.07 | **80.26** | 79.61 | 79.57 | 78.65 |

Table 16: Generation performance of TDSM models for different assumed noise rates on the CIFAR-10 dataset with 40% symmetric noise. The percentage in parentheses refers to the noise rate assumed by the TDSM models.

| | Metric | | DSM | TDSM (40%) | TDSM (43%) |
|---|---|---|---|---|---|
| un | FID | (↓) | **2.07** | 2.43 | 2.53 |
| | IS | (↑) | 9.83 | **9.96** | **9.96** |
| | Density | (↑) | 100.94 | 111.63 | **113.21** |
| | Coverage | (↑) | 80.93 | 82.03 | **82.18** |
| cond | CW-FID | (↓) | 30.45 | 15.92 | **15.76** |
| | CAS | (↑) | 47.21 | **62.28** | 62.10 |
| | CW-Density | (↑) | 73.02 | 97.80 | **100.06** |
| | CW-Coverage | (↑) | 71.63 | 78.65 | **79.11** |

Therefore, it is possible to improve the generation performance by training the noisy label classifier more effectively, e.g., by adjusting the temporal weight function $\lambda(t)$, and further work is needed.

### F.14   ANALYSIS OF THE POTENTIAL LABEL NOISE IN THE BENCHMARK DATASET

In Section 4.2, we discuss the presence of noise in the annotated labels of existing benchmark datasets. This could potentially affect the results in our experiments on benchamrk dataset with synthetic label noise. To assess this influence, we apply TDSM to the CIFAR-10 dataset with 40% symmetric noise, assuming a 43% noise rate. In this case, we assume that the 60% of unaffected data in our noisy label generation process has the 5% potential noise predicted by the previous experiment, resulting in an additional 3% of total data.

Table 16 shows the results of the TDSM model assuming a 43% noise rate on the CIFAR-10 dataset with 40% symmetric noise. Interestingly, we find that assuming potential label noise yields additional performance improvements for most metrics. This result further supports our conjecture that the benchmark dataset contains examples with noisy or ambiguous labels.

### F.15   ADDITIONAL GENERATED IMAGES

Figures 16 to 21 provides the uncurated generated images of the baseline and our models in Table 1. As with the quantitative results, our models demonstrate the ability to generate images that closely match the given conditions.

### F.16   GENERATED IMAGES ON CLOTHING-1M

Figure 22 contains the uncurated generated images of the baseline and our models, trained on the Clothing-1M dataset. Although our model generates images that exhibit some alignment with

the specified conditions, the disparity is not significant. This observation can be attributed to the underlying assumption of class-conditional label noise. Consequently, it is evident that further research is necessary in the area of generative model learning from instance-dependent noisy labels, similar to the research in supervised learning dealing with such noisy labels.

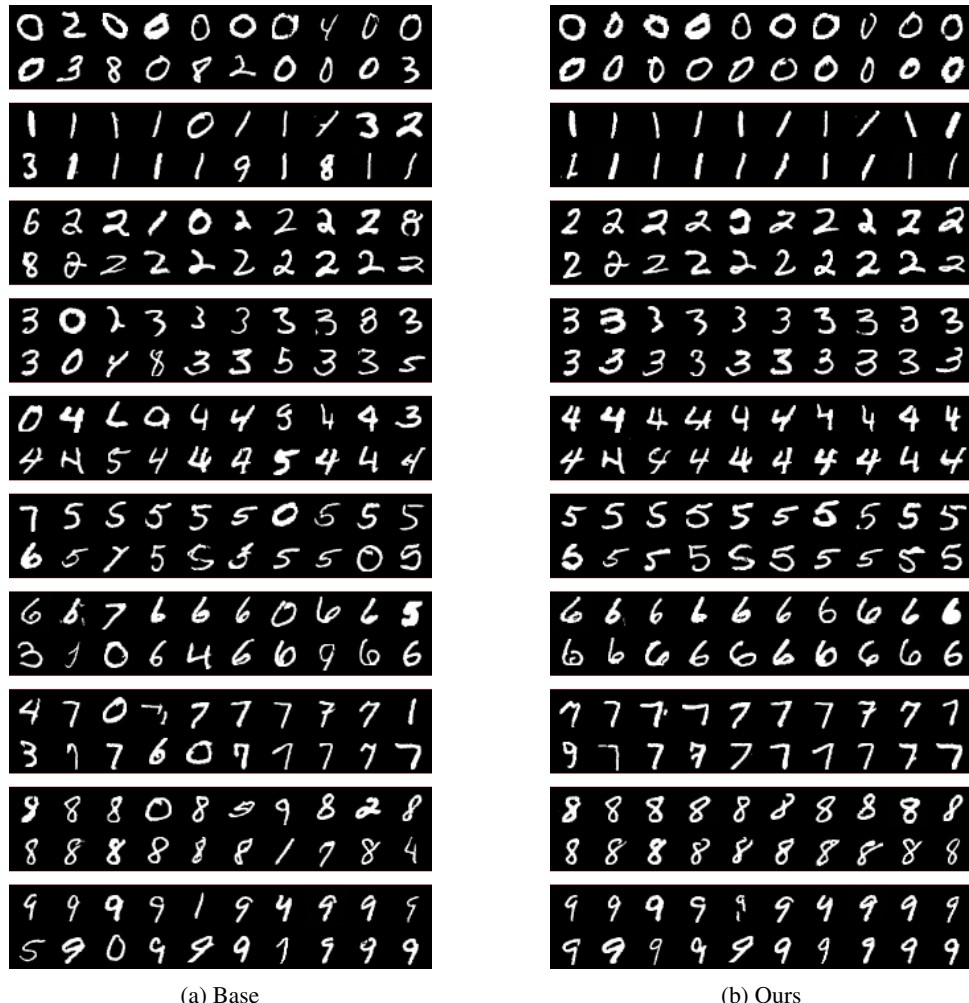

(a) Base                                    (b) Ours

Figure 16: The uncurated generated images of (a) baseline and (b) our models on the MNIST dataset with 40% symmetric noise. Each block has images of the same class. The class labels are 0, 1, 2, 3, 4, 5, 6, 7, 8, and 9, from top to bottom.

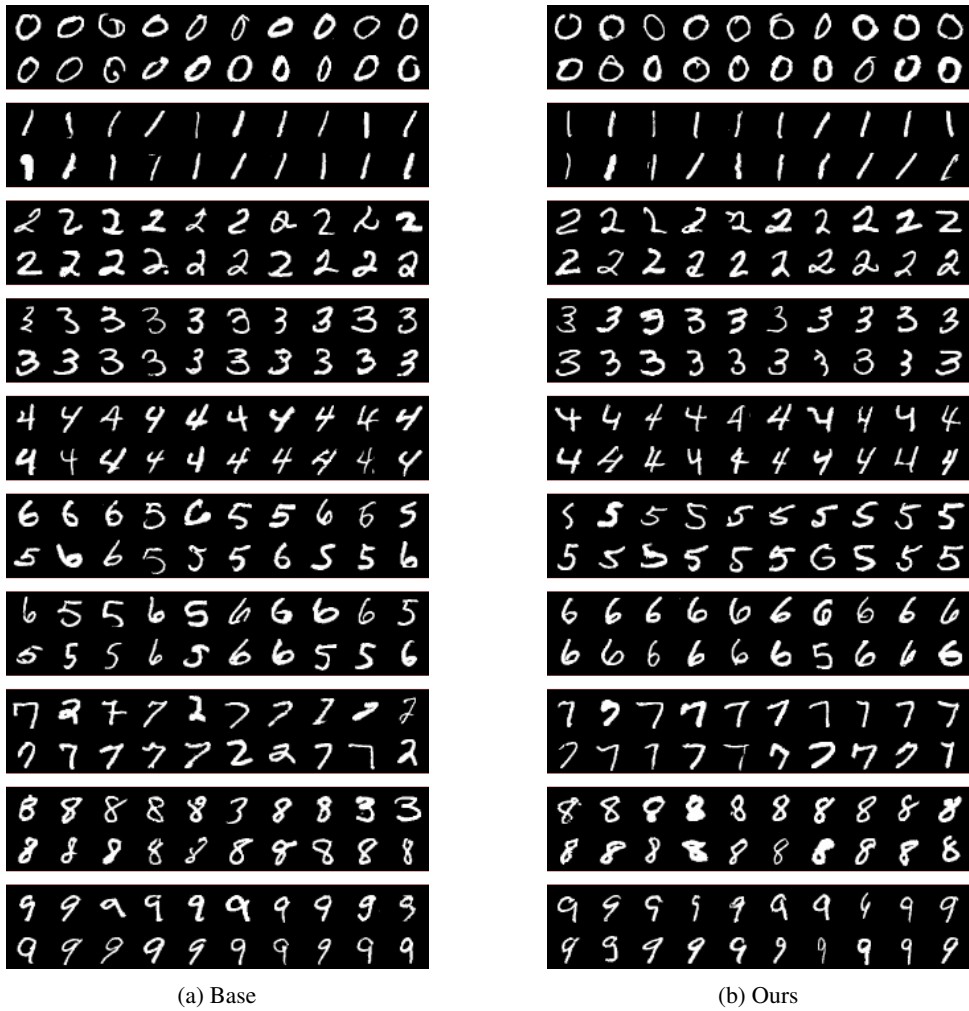

(a) Base                              (b) Ours

Figure 17: The uncurated generated images of (a) baseline and (b) our models on the MNIST dataset with 40% asymmetric noise. Each block has images of the same class. The class labels are 0, 1, 2, 3, 4, 5, 6, 7, 8, and 9, from top to bottom. The labels are flipped by $2 \to 7$, $3 \to 8$, and $5 \leftrightarrow 6$.

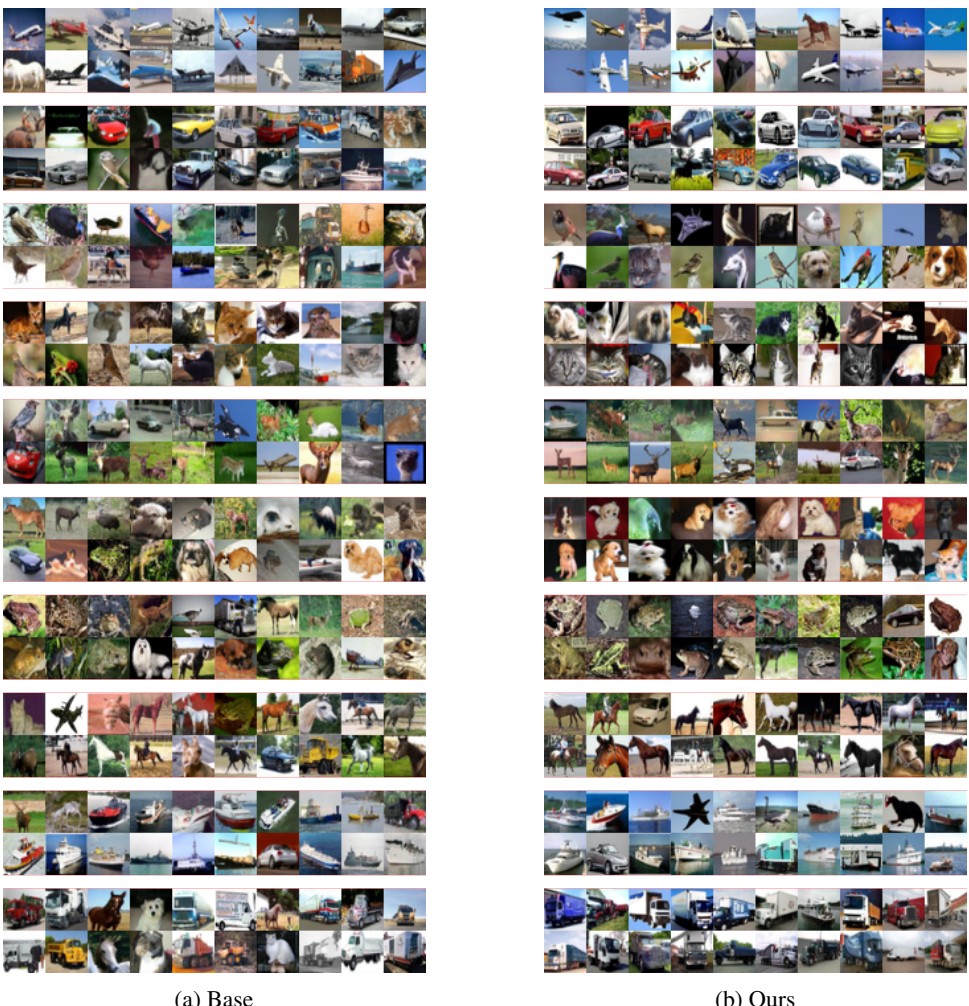

(a) Base               (b) Ours

Figure 18: The uncurated generated images of (a) baseline and (b) our models on the CIFAR-10 dataset with 40% symmetric noise. Each block has images of the same class. The class labels are `airplane`, `automobile`, `bird`, `cat`, `deer`, `dog`, `frog`, `horse`, `ship`, and `truck`, from top to bottom.

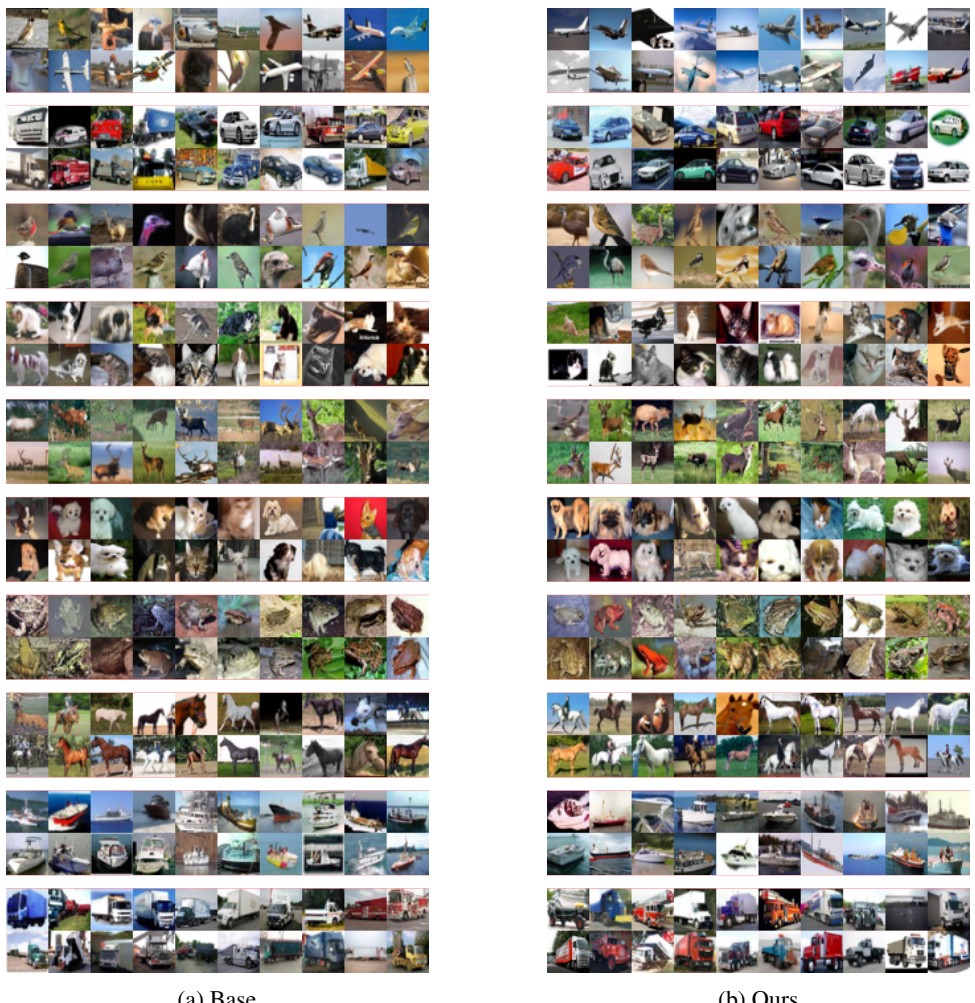

(a) Base                                        (b) Ours

Figure 19: The uncurated generated images of (a) baseline and (b) our models on the CIFAR-10 dataset with 40% asymmetric noise. Each block has images of the same class. The class labels are airplane, automobile, bird, cat, deer, dog, frog, horse, ship, and truck, from top to bottom. The labels are flipped by truck → automobile, bird → airplane, deer → horse, cat ↔ dog.

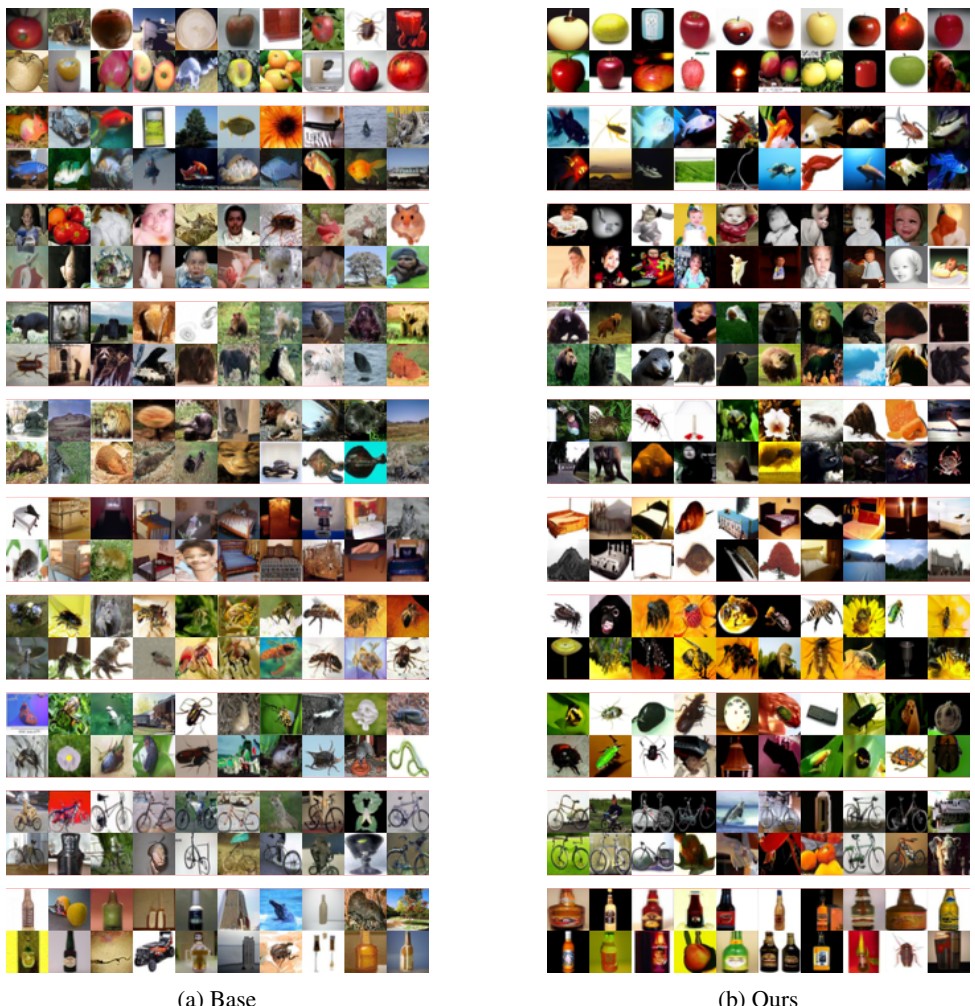

(a) Base             (b) Ours

Figure 20: The uncurated generated images of (a) baseline and (b) our models on the CIFAR-100 dataset with 40% symmetric noise. Each block has images of the same class. The class labels are `apple`, `aquarium fish`, `baby`, `bear`, `beaver`, `bed`, `bee`, `beetle`, `bicycle`, and `bottle`, from top to bottom.

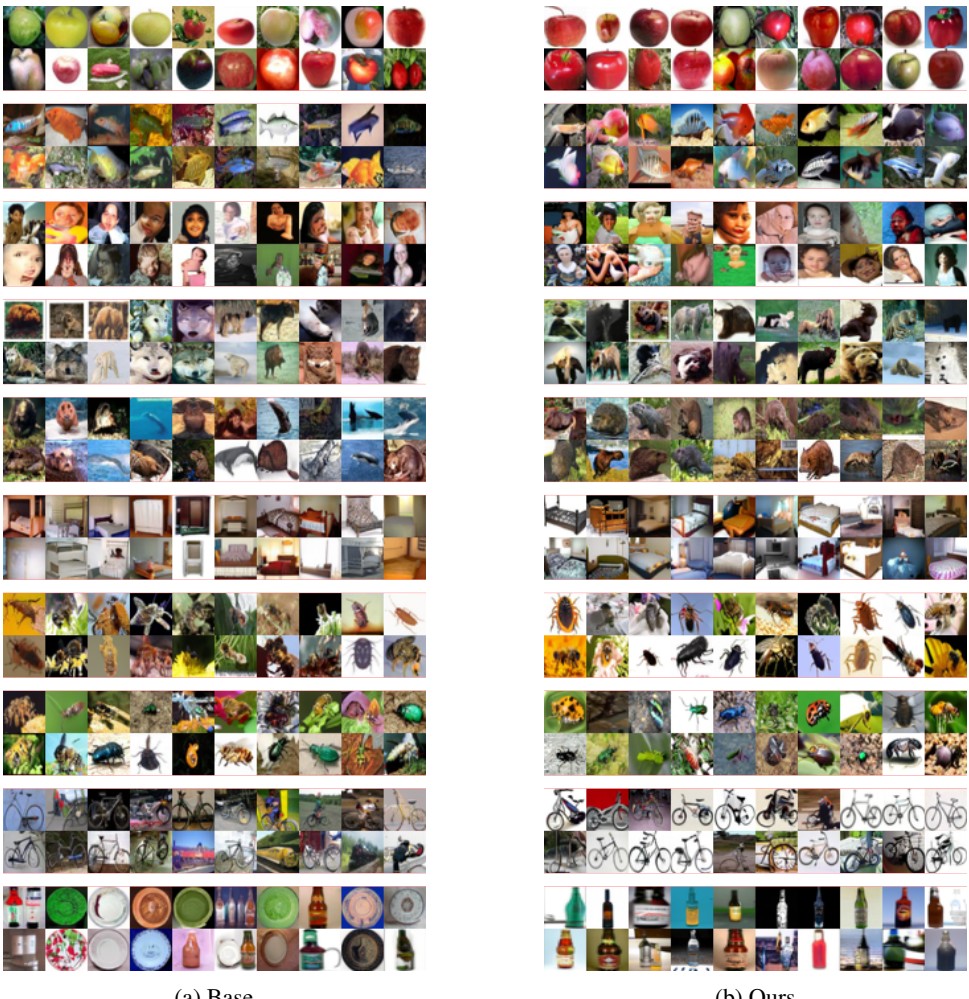

(a) Base            (b) Ours

Figure 21: The uncurated generated images of (a) baseline and (b) our models on the CIFAR-100 dataset with 40% asymmetric noise. Each block has images of the same class. The class labels are `apple`, `aquarium fish`, `baby`, `bear`, `beaver`, `bed`, `bee`, `beetle`, `bicycle`, and `bottle`, from top to bottom. For these labels, the labels are flipped to `mushroom`, `flatfish`, `boy`, `leopard`, `dolphin`, `chair`, `beetle`, `butterfly`, `bus`, and `bowl`, respectively.

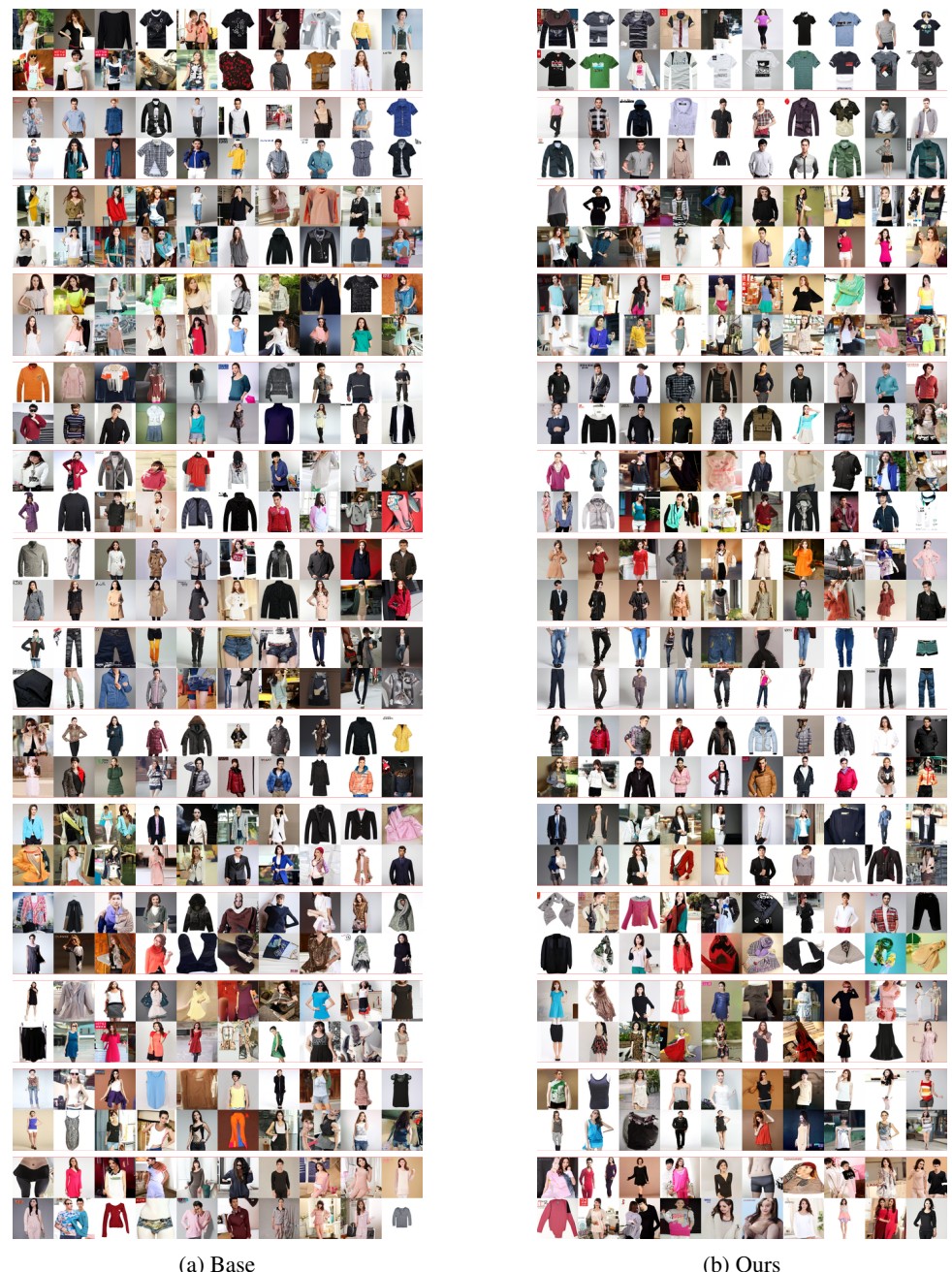

(a) Base                                                                   (b) Ours

Figure 22: The uncurated generated images of (a) baseline and (b) our models on the Clothing-1M dataset. Each block has images of the same class. The class labels are `T-shirt`, `Shirt`, `Knitwear`, `Chiffon`, `Sweater`, `Hoodie`, `Windbreaker`, `Jacket`, `Down Coat`, `Suit`, `Shawl`, `Dress`, `Vest`, and `Underwear`, from top to bottom.

