# OpenReview forum: "Label-Noise Robust Diffusion Models"
_ICLR.cc/2024/Conference — ICLR 2024 poster_

### Official Review · Reviewer_LRz9 · 2023-10-16

**Soundness:** 3 good
**Presentation:** 3 good
**Contribution:** 3 good
**Rating:** 6
**Confidence:** 3

**Summary:**

This paper proposes a novel method, namely Transition-aware weighted Denoising Score Matching (TDSM), to train conditional diffusion models with noisy labels. The TDSM objective contains a weighted sum of score networks. Additionally, it also introduces a transition-aware weight estimator to leverage a time-dependent noisy-label classifier distinctively customized to the diffusion process. Experimental results on multiple popular datasets demonstrate the effectiveness of the proposed method.

**Strengths:**

1. This paper is well-written with clear method procedure.
2. The idea is clear and effective.
3. This paper have good experimental results.

**Weaknesses:**

1. It is not clear what is the major difference from the methods that boosting the robustness of generative models on noisy labels.
2. The significance of this research topic is not clear, please explain it. Specifically, to the best of my knowledge, generative models are usually unsupervised, and thus there is only a few methods on boosting the model robustness against on noisy labels.
3. It is not clear the model performance on severe noisy labels, like 60%, 80%.
4. It is not clear whether the proposed method can boost the model classification performance?

**Questions:**

1. Why boosting the robustness of diffusion models on noisy labels is very significant?
2. Can the proposed method boost the model classification performance on noisy labels?
3. What the limitations of the proposed method and please point out the future work.

---

> ### Author Response · Authors · 2023-11-17
> **Response to Reviewer LRz9 (Part 1)**
>
> We appreciate the thorough reviews and valuable comments. We address the concerns below.
>
>
> >**Q1. [Major difference from the other methods for generative model learning with noisy label]** *(Weaknesses 1) It is not clear what is the major difference from the methods that boosting the robustness of generative models on noisy labels.*
>
> We compared our model with GAN-based generative models for the noisy label problem in the last paragraph of Section 2 and immediately below Proposition 2 in Section 3. For clarity, we present the detailed explanation of the GAN-based models and the comparison table in below and include it in the Appendix B.4 of the revised paper.
>
> |                                 | [D1, D2]                                          | Ours                                   |
> |---------------------------------|-----------------------------------------------|----------------------------------------|
> | Base model                      | GAN                                           | Diffusion model                        |
> | Objective                       | Noisy-label conditional **distribution** matching | Noisy-label conditional **score** matching |
> | Label transition information    | $p(\tilde{y}\|y)$                             | $p_t(y\|\tilde{y},x_t)$                |
> | &nbsp;&nbsp;&nbsp;&nbsp;Time-dependent label transition | X                                        | O                                 |
> | &nbsp;&nbsp;&nbsp;&nbsp;Instance-wise label transition  | X                                        | O                                 |
>
> There are a few studies for generative model learning from noisy labels. [D1, D2] proposed algorithms to overcome the difficulties of training GANs with noisy labels by matching the noisy-label conditional probability with the transition matrix $p(\tilde{y}\|y)$. On the other hand, our method focused on the robustness of diffusion models on noisy labels by matching the noisy-label conditional scores with the transition-aware weighted function $p_t(y\|\tilde{y},x_t)$.
>
> Specifically, the structure of a generator in [D1, D2] is similar to the traditional GAN, but they modify the discriminator structure by incorporating the label transition matrix. When the discriminator is fed the samples generated by the generator and the corresponding labels, the labels are corrupted by the label transition matrix to become noisy labels. Since the given dataset is also a noisy labeled dataset, training the discriminator in this way makes it a match between the noisy labeled datasets. Then, the generator can have a clean label conditional distribution.
>
> However, the diffusion model cannot completely mitigate the impact of noisy labels with the transition matrix alone because it targets the gradient of the log-likelihood, i.e., data score. We show theoretically the impact of noisy label on data score in Theorem 1. This impact could be represented by the transition-aware weighted function $w$, which is time- and instance-dependent unlike the transition matrix. We include further analysis of this function in the global response.
> ***
> [D1] Thekumparampil, K. K., Khetan, A., Lin, Z., & Oh, S. (2018). Robustness of conditional gans to noisy labels. Advances in neural information processing systems, 31.
> [D2] Kaneko, T., Ushiku, Y., & Harada, T. (2019). Label-noise robust generative adversarial networks. In Proceedings of the IEEE/CVF Conference on Computer Vision and Pattern Recognition (pp. 2467-2476).

---

> ### Author Response · Authors · 2023-11-17
> **Response to Reviewer LRz9 (Part 2)**
>
> >**Q2. [Significance of the research topic]** *(Weaknesses 2) The significance of this research topic is not clear, please explain it. Specifically, to the best of my knowledge, generative models are usually unsupervised, and thus there is only a few methods on boosting the model robustness against on noisy labels.*
>
> While generative models are typically formulated on an unsupervised basis, their practical application often involves the generation of samples tailored to the user’s specific needs [D3, D4, D5, D6]. Therefore, conditional generative models play a crucial role in the practical application of generative models. In this case, the noisy label problem is a pervasive problem in the entire field of machine learning and is particularly critical in practical applications [D7, D8, D9]. Thus, as the reviewer points out, the lack of research on generative models dealing with noisy labels is a significant problem for the application of generative models.
>
> Reviewer BRfo acknowledges that our model is the first work to consider the influence of noisy labels in diffusion models, and Reviewer Zgz7 values that our research topic targets an important problem because it is a common and practical challenge in real-world scenarios.
> ***
> [D3] Perarnau, G., van de Weijer, J., Raducanu, B., & Álvarez, J. M. (2016). Invertible Conditional GANs for image editing. NIPS Workshop on Adversarial Training.
> [D4] Das, H. P., Tran, R., Singh, J., Yue, X., Tison, G., Sangiovanni-Vincentelli, A., & Spanos, C. J. (2022, June). Conditional synthetic data generation for robust machine learning applications with limited pandemic data. In Proceedings of the AAAI Conference on Artificial Intelligence (Vol. 36, No. 11, pp. 11792-11800).
> [D5] Pinaya, W. H., Tudosiu, P. D., Dafflon, J., Da Costa, P. F., Fernandez, V., Nachev, P., ... & Cardoso, M. J. (2022, September). Brain imaging generation with latent diffusion models. In MICCAI Workshop on Deep Generative Models (pp. 117-126).
> [D6] Iyer, A., Dey, B., Dasgupta, A., Chen, W., & Chakraborty, A. (2019). A conditional generative model for predicting material microstructures from processing methods. Second Workshop on Machine Learning and the Physical Sciences in NeurIPS 2019.
> [D7] Welinder, P., Branson, S., Perona, P., & Belongie, S. (2010). The multidimensional wisdom of crowds. Advances in neural information processing systems, 23.
> [D8] Xiao, T., Xia, T., Yang, Y., Huang, C., & Wang, X. (2015). Learning from massive noisy labeled data for image classification. In Proceedings of the IEEE conference on computer vision and pattern recognition (pp. 2691-2699).
> [D9] Cheng, H., Zhu, Z., Li, X., Gong, Y., Sun, X., & Liu, Y. (2020, October). Learning with Instance-Dependent Label Noise: A Sample Sieve Approach. In International Conference on Learning Representations.
>
> ***
> ***
>
> >**Q3. [Performance on severe noisy labels]** *(Weaknesses 3) It is not clear the model performance on severe noisy labels, like 60%, 80%.*
>
> To verify the robustness of the proposed model under extreme noise, we perform the experiments on the CIFAR-10 dataset with severe symmetric noise rates of 60% and 80%. The results are presented in the table below. These results show that TDSM consistently improves performance even at severe noise rates. We include these results in Appendix F.11 of the revised version.
>
> |      | Metric                   | 60%      |            | 80%      |            |
> |------|--------------------------|----------|------------|----------|------------|
> |      |                          | DSM      | TDSM       | DSM      | TDSM       |
> | un   | FID ($\downarrow$)       | **2.17** | 2.55       | **2.22** | 2.44       |
> |      | IS ($\uparrow$)          | 9.75     | **10.09**  | 9.73     | **9.79**   |
> |      | Density ($\uparrow$)     | 102.17   | **110.78** | 103.09   | **103.69** |
> |      | Coverage ($\uparrow$)    | 81.00    | **81.65**  | 80.81    | **81.15**  |
> | cond | CW-FID ($\downarrow$)    | 48.94    | **23.36**  | 72.51    | **53.57**  |
> |      | CAS ($\uparrow$)         | 29.72    | **52.38**  | 9.93     | **24.68**  |
> |      | CW-Density ($\uparrow$)  | 58.42    | **86.58**  | 43.97    | **54.30**  |
> |      | CW-Coverage ($\uparrow$) | 61.88    | **75.08**  | 44.53    | **56.47**  |

---

> ### Author Response · Authors · 2023-11-17
> **Response to Reviewer LRz9 (Part 3)**
>
> >**Q4. [Classification Performance]** *(Weaknesses 4) It is not clear whether the proposed method can boost the model classification performance.*
>
> >*(Questions 2) Can the proposed method boost the model classification performance on noisy labels?*
>
> We provide the response to this question in the Global Response. For your convenience, we also copy and paste the response provided in the Global Response.
>
> ***
>
> **[Advantage of the proposed method]**
>
> Label recovery is an intuitive way to overcome the problem of noisy labels in a dataset. Our method does not specifically focus on robustness to noisy labels through label recovery; instead, it focuses on robustness to noisy labels from a diffusion model training perspective. Consequently, our approach can be synergistically combined with existing label recovery methods to further improve performance. The analysis of this synergy is detailed in Section 4.4.
>
> It is important to analyze how our method overcomes the noisy label robustness in diffusion models. Our analysis indicates that our proposed TDSM objective provides the diffusion model with information about the diffused label noise in the dataset. This information is represented by the transition-aware weight function $w$, depending upon the diffusion time. Therefore, we estimate the instance- and time-dependent label transition probability with the transition-aware weight function.
>
> **[Transition-aware weight function plays a different role from a typical classifier]**
>
> We want to emphasize that the transition-aware weight function $w$ plays a different role from a typical classifier. This probability represents the relationship between the noisy and clean label conditional scores (which are also time-dependent) and is an important element of the proposed training objective of a diffusion model. Therefore, we need to estimate this label transition probabilities over diffusion timesteps.
>
> Specifically, our transition-aware weight function can be reformulated as follows:
> $$
> w(x_t,\tilde{y},y,t)=\frac{p_t(Y=y|x_t)p(\tilde{Y}=\tilde{y}|Y=y)}{p_t(\tilde{Y}=\tilde{y}|x_t)}.
> $$
> As seen in the expression, the weight function is composed of the clean label classifier $p_t(Y=y|x_t)$, label transition prior $p(\tilde{Y}=\tilde{y}|Y=y)$, and the noisy label classifier  $p_t(\tilde{Y}=\tilde{y}|x_t)$. Given the triplet of perturbed data instance and its corresponding noisy label and timestep, i.e., $(x_t, \tilde{y}, t)$, each component of $w$ has the following characteristics with respect to $y$: 1) $p(Y=y|x_t)$ is maximized when $y$ is the clean label of the clean data $x_0$ of $x_t$; 2) $p(\tilde{Y}=\tilde{y}|Y=y)$ is maximized when $y$ is the noisy label $\tilde{y}$ in general. The reason for 2) is that in general, a given noisy label is sufficiently likely to be a clean label. These two trade-offs imply that the $w$ function does not behave like a clean label classifier.
>
> Furthermore, for large enough $t$, the distribution of $x_t$ converges to a label-independent prior distribution by the design of the diffusion process, so that $p_t(Y|x_t)$ converges to a uniform distribution. Therefore, for sufficiently large $t$, the $w$ function converges to a transition matrix $S$. This phenomenon can also be demonstrated in Figure 2 of the 2-D example.
>
> In summary, the transition-aware weight function needed to overcome noisy labels in a diffusion model is not represented clean label recovery information only, and this function has time-varying information.
>
> **[Experimental results of clean label accuracy]**
>
> It is possible to check the accuracy of the transition-aware weight function on the training dataset since the function provides the probability of a clean label given a noisy label and an instance. Figure 13 in the revised manuscript shows the clean label accuracy of the transition-aware weight function over diffusion timesteps. Since the noisy labels in the dataset are used as the input conditions of the baseline model for the entire diffusion timesteps, we plot the percentages of clean labels in the datasets as dots at $t=0$ and plot a dashed horizontal line for comparison.
>
> Focusing on 'Ours (VolMinNet + TDSM)', which provides the best generation performance, we found that the behavior of the $w$ function varies with diffusion timesteps, as mentioned above. In particular, as $t$ increases, the clean label accuracy converges to the clean rate of the given dataset, which is 0.6. This is because the $w$ function converges to the label transition prior $p(\tilde{Y}|Y)$. 'Base (VolMinNet + DSM)', which does not take this into account, learns the diffusion model only with corrected labels, leading to the performance difference. Therefore, while existing noisy label methods that focus on clean label recovery certainly contribute significantly to generative performance, considering the TDSM objective in diffusion model training enables additional performance improvement independent of clean label recovery.

---

> ### Author Response · Authors · 2023-11-17
> **Response to Reviewer LRz9 (Part 4)**
>
> >**Q5. [Importance of the robustness of diffusion models on noisy labels]** *(Questions 1) Why boosting the robustness of diffusion models on noisy labels is very significant?*
>
> The performance of diffusion models is highly dependent on the amount of data [D10, D11]. However, as the amount of data increases, the noisy labels of the dataset tend to increase as well. This is because the large amount of data makes curated labeling more difficult. In conditional generation, the basic requirement is to generate samples that satisfy the given conditions. Such label noise leads to the generation of samples that do not meet the conditions, making it difficult to produce the desired samples for the user.
> ***
> [D10] Nichol, A. Q., & Dhariwal, P. (2021, July). Improved denoising diffusion probabilistic models. In International Conference on Machine Learning (pp. 8162-8171). PMLR.
> [D11] Moon, T., Choi, M., Lee, G., Ha, J. W., & Lee, J. (2022, November). Fine-tuning Diffusion Models with Limited Data. In NeurIPS 2022 Workshop on Score-Based Methods.
>
> ***
> ***
> >**Q6. [Limitations and future works]** *(Questions 3) What the limitations of the proposed method and please point out the future work?*
>
> One limitation is that the theoretical analysis requires the class-dependent label noise assumption. While theoretical developments require assumptions about the noise setting, our model remains effective in real-world scenarios where these assumptions may not hold, as shown by the Clothing-1M experiment in Section 4.3.
>
> In the existing literature on noisy labels, there has been an extension from class-dependent label noise settings to instance-dependent label noise settings [D12, D13, D14, D15]. Especially in the context of learning diffusion models, there is a distinctive characteristic that, despite the class-dependent label noise assumption, the prediction of instance-dependent label transition probabilities is necessary. Therefore, investigating training methods under the current assumptions in the context of diffusion models lays the foundation for developing better methods. Similar to existing classification methods, extending our approach beyond the current assumption would be a valuable future work.
>
> ***
>
> [D12] Xia, X., Liu, T., Han, B., Wang, N., Gong, M., Liu, H., Niu, G., Tao, D., & Sugiyama, M. (2020). Part-dependent label noise: Towards instance-dependent label noise. Advances in Neural Information Processing Systems, 33, 7597-7610.
> [D13] Berthon, A., Han, B., Niu, G., Liu, T., & Sugiyama, M. (2021, July). Confidence scores make instance-dependent label-noise learning possible. In International conference on machine learning (pp. 825-836). PMLR.
> [D14] Yao, Y., Liu, T., Gong, M., Han, B., Niu, G., & Zhang, K. (2021). Instance-dependent label-noise learning under a structural causal model. Advances in Neural Information Processing Systems, 34, 4409-4420.
> [D15] Cheng, D., Liu, T., Ning, Y., Wang, N., Han, B., Niu, G., Gao, X., & Sugiyama, M. (2022). Instance-dependent label-noise learning with manifold-regularized transition matrix estimation. In Proceedings of the IEEE/CVF Conference on Computer Vision and Pattern Recognition (pp. 16630-16639).

---

> ### Author Response · Authors · 2023-11-23
>
> Dear Reviewer LRz9,
>
> We sincerely appreciate the reviewer's constructive feedback, which is invaluable in improving our work. We kindly ask the reviewer to consider our response and the revised version, where we have tried to incorporate your feedback as thoroughly as possible. As the discussion period ends in 12 hours, we are curious to know if we have adequately understood and addressed your concerns. We are always open to further discussion about our research at any time. Again, we appreciate the reviewer's efforts.
>
> Best Regards,

---

### Official Review · Reviewer_TMN7 · 2023-10-30

**Soundness:** 3 good
**Presentation:** 3 good
**Contribution:** 3 good
**Rating:** 6
**Confidence:** 4

**Summary:**

This paper highlights the challenges associated with training on extensive datasets, which often contain noise in their condition to make noisy labels. Such noise introduces the risk of condition mismatches, which can degrade the quality of the generated data. To tackle this issue, the paper presents the Transition-aware weighted Denoising Score Matching (TDSM) method. This approach is specifically designed to robustly train conditional diffusion models with noisy labels.

The TDSM framework incorporates a label-transition weight for the score networks. These weights are derived from the relationship between conditional scores for both noisy and genuine labels, and can be estimated with a pre-trained noisy-classifier. Empirical evaluations, on multiple datasets and a range of noisy label configurations, demonstrate the efficiency of the TDSM approach.

**Strengths:**

- The paper is well-written and polished, facilitating a smooth reading experience. The mathematical presentations are articulated clearly and the theoretical results are complete and sound. Additionally, the inclusion of model overviews and illustrative figures for the components simplifies the understanding of the proposed method.

- The experimental results well validate the approach. Notably, the paper comprehensively study both the effects of the conditional models and the impact of conditional generation with label guidance. Comprehensive results are provided in both the main paper and supplementary material.

- The practical side of the research is solid. The authors have been very detailed in their implementation and provided their experiment code, which ensures reproducibility.

**Weaknesses:**

- Given that the estimation of the transition-aware weight relies on a noisy-classifier, it would be advantageous for the authors to present studies evaluating how the performance of the noisy-classifier affects the model's overall performance.

- In Table 2, the authors seem to only compare with DSM with non-class-aware evaluation metrics. What's the reason for this comparison with the specific metrics?

- On "clean" datasets, TDSM demonstrates notably superior performance, suggesting the potential presence of noisy labels. It would be insightful to know the threshold or proportion of noisy labels at which a significant performance difference emerges between DSM and TDSM. Furthermore, if the datasets used to train the noisy-classifier (for estimating class transitions) contain noisy labels, would this introduce additional inaccuracies in label correction? It would be beneficial for the authors to conduct a thorough analysis of these concerns.

- In the review of diffusion models, the authors seem to only review from the score matching networks, while omit the diffusion models derivated from optmizing the ELBO.


- Previous works are not correctly reviewed or cited. For example, the reference of denoising diffusion probabilistic model (Ho et al., 2020) is classified into video generation, in the introduction, while this is a fundamental work in diffusion models, and the authors may wanted to put video diffusion models (Ho et al., 2022) there. Moreover, some prior works that tackle the uncurated label distributions are not discussed and compared in the paper.

- It would be beneficial to include a color bar in Fig 8 to interprete the meaning of the colors presented.


*Reference*:

*Ho, Jonathan, Ajay Jain, and Pieter Abbeel. "Denoising diffusion probabilistic models." Advances in neural information processing systems 33 (2020): 6840-6851.*

*Rangwani, Harsh, Konda Reddy Mopuri, and R. Venkatesh Babu. "Class balancing gan with a classifier in the loop." Uncertainty in Artificial Intelligence. PMLR, 2021.*

*Rangwani, Harsh, et al. "Improving GANs for Long-Tailed Data Through Group Spectral Regularization." European Conference on Computer Vision. Cham: Springer Nature Switzerland, 2022.*

*Ho, Jonathan et al. “Video Diffusion Models.” ArXiv abs/2204.03458 (2022).*

*Qin, Yiming, et al. "Class-Balancing Diffusion Models." Proceedings of the IEEE/CVF Conference on Computer Vision and Pattern Recognition. 2023.*

**Questions:**

Please see the Weakness section.

---

> ### Author Response · Authors · 2023-11-17
> **Response to Reviewer TMN7 (Part 1)**
>
> We appreciate the thorough reviews and valuable comments. We address the concerns below.
>
> >**Q1. [Effect of the performance of the noisy classifier]** *(Weaknesses 1) Given that the estimation of the transition-aware weight relies on a noisy-classifier, it would be advantageous for the authors to present studies evaluating how the performance of the noisy-classifier affects the model's overall performance.*
>
> We are training the diffusion models with different training levels of noisy label classifiers. In this experiment, we will analyze the performance of the noisy label classifier over time and the performance of the corresponding diffusion model. However, training the diffusion model is a time-consuming process, so we compare two classifiers in now, and we will add additional analysis before the end of the discussion period.
>
> To analyze the noisy label classifiers, we measured 1) the noisy label classification performance of the noisy label classifier over diffusion timesteps; and 2) the clean label classification performance of the $w$ function, evaluated with the classifier, over diffusion timesteps. We also compare the generation performance for the diffusion model trained with each classifier. We present these results in Figure 14 and Table 15 in Appendix F.13 of the revised manuscript.
>
> The performance of the noisy label classifier in predicting noisy labels increased with the number of epochs trained on the classifier, but its ability to predict clean labels in clean data space decreased. Meanwhile, in terms of the generation performance of the diffusion model, the less trained classifier performed better with respect to the conditional metrics, and the more trained classifier performed better with respect to the unconditional metrics. Note that both of our models outperform the baseline on most metrics.
>
> We interpret this result as follows. 1) Due to the iterative sampling procedure of the diffusion model, the noisy label classifier needs to learn enough of all the perturbed samples of the diffusion timesteps to improve the quality of the generated samples. 2) At this time, excessive training of the noisy label classifier leads to an overconfidence problem, particularly in the data space at $t=0$. This overconfidence reduces the information of the clean label and affects the information of the condition. Therefore, it is possible to improve the generation performance by training the noisy label classifier more effectively, e.g., by adjusting the temporal weight function $\lambda(t)$, and further work is needed.
>
> ***
> ***
>
> >**Q2. [class-aware evaluation metrics for Table 2]** *(Weaknesses 2) In Table 2, the authors seem to only compare with DSM with non-class-aware evaluation metrics. What's the reason for this comparison with the specific metrics?*
>
> The experiments for Table 2 in Section 4.2 applied our models to benchmark dataset with the conjecture that some annotated labels may be noisy. For the class-aware evaluation metrics, we need to have true labels, but due to the conjecture, we thought it was not appropriate to use these metrics with the annotated labels.
>
> Having said that, it is possible to evaluate the class-aware evaluation metrics with the annotated labels. The results are shown below, and we find that our model further improves the intra-class sample quality. We add this reason and include the results in the Appendix F.10 and Table 12 in the revised manuscript.
>
> |      | Metric                | MNIST     |           | CIFAR-10  |            | CIFAR-100 |           |
> |------|-----------------------|-----------|-----------|-----------|------------|-----------|-----------|
> |      |                       | Base      | Ours      | Base      | Ours       | Base      | Ours      |
> | un   | FID ($\downarrow$)    | -         | -         | 1.92      | **1.91**   | **2.51**  | 2.67      |
> |      | IS ($\uparrow$)       | -         | -         | 10.03     | **10.10**  | 12.80     | **12.85** |
> |      | Density ($\uparrow$)   | 86.20     | **88.08** | 103.08    | **104.35** | 87.98     | **90.04** |
> |      | Coverage ($\uparrow$)    | 82.90     | **83.69** | 81.90     | **82.07**  | 77.63     | **78.28** |
> | cond | CW-FID ($\downarrow$) | -         | -         | 10.23     | **10.18**  | 66.97     | **66.68** |
> |      | CAS ($\uparrow$) | **98.55** | 98.50     | **77.74** | 77.07      | **39.50** | 39.10     |
> |      | CW-Density ($\uparrow$) | 85.79     | **87.96** | 102.63    | **103.69** | 82.58     | **84.96** |
> |      | CW-Coverage ($\uparrow$)  | 82.09     | **82.98** | 81.57     | **81.88**  | 75.78     | **76.53** |

---

> ### Author Response · Authors · 2023-11-17
> **Response to Reviewer TMN7 (Part 2)**
>
> >**Q3. [Noise rate of clean datasets]** *(Weaknesses 3) On "clean" datasets, TDSM demonstrates notably superior performance, suggesting the potential presence of noisy labels. It would be insightful to know the threshold or proportion of noisy labels at which a significant performance difference emerges between DSM and TDSM.*
>
> Thank you for your valuable suggestions. To see the impact of the noise rate of TDSM on the clean dataset, we train TDSM models on the clean CIFAR-10 dataset assuming different (symmetric) noise rates and the results are shown below.
>
> | Metric             | DSM    | TDSM   |        |           |            |        |
> |--------------------|--------|--------|--------|-----------|------------|--------|
> |                    |        | 1%     | 2.5%   | 5%        | 7.5%       | 10%    |
> | FID ($\downarrow$) | 1.92   | 1.94   | 1.93   | **1.91**  | 2.02       | 2.00   |
> | IS ($\uparrow$)    | 10.03  | 10.08  | 10.04  | **10.10** | 10.04      | 10.06  |
> | Density ($\uparrow$)  | 103.08 | 103.83 | 104.07 | 104.35    | **105.44** | 105.15 |
> | Coverage ($\uparrow$)  | 81.90  | 81.79  | 81.90  | **82.07** | 82.04      | 81.95  |
>
> The percentage in the header indicates the assumed noise rate. As shown in the table, the model performs best at a noise rate of 5\%. In addition, most TDSM models show improvements compared to DSM. This suggests that the consideration of label transitions over timesteps contributes to improved performance even on clean datasets. However, it is important to note that additional research is needed to account for the actual noise labels present in the data because our experiments assume symmetric noise. We add this reason and include the results in the Appendix F.10 and Table 13 in the revised manuscript.
>
> ***
> ***
>
> >**Q4. [Impact of potential label noise in the benchmark dataset]** *(Weaknesses 3) Furthermore, if the datasets used to train the noisy-classifier (for estimating class transitions) contain noisy labels, would this introduce additional inaccuracies in label correction? It would be beneficial for the authors to conduct a thorough analysis of these concerns.*
>
> As suggested by the reviewer, the presence of noise in the annotated labels of existing benchmark datasets could potentially affect performance. To assess this influence, we apply TDSM to the CIFAR-10 dataset with 40% symmetric noise, assuming a 43% noise rate. In this case, we assume that the 60% of unaffected data in our noisy label generation process has the 5% potential noise predicted by the previous experiment, resulting in an additional 3% of total data.
>
> Due to the time-consuming training of diffusion models, we will provide the analysis before the end of the discussion period.
>
> ***
> ***
>
> >**Q5. [Review of diffusion models]** *(Weaknesses 4) In the review of diffusion models, the authors seem to only review from the score matching networks, while omit the diffusion models derivated from optmizing the ELBO.*
>
> Thank you for pointing out the problem. Our method is developed through the relationship between conditional scores; therefore, we explained the diffusion model based on the score matching model. As suggested by the reviewer, we add the discussion of the diffusion models derived from optimizing the ELBO and the explanations of the relationship between the score matching model and the diffusion model in the new Appendix B.1 in the revised version.
>
> ***
> ***
>
> >**Q6. [Previous works]** *(Weaknesses 5) Previous works are not correctly reviewed or cited. For example, the reference of denoising diffusion probabilistic model (Ho et al., 2020) is classified into video generation, in the introduction, while this is a fundamental work in diffusion models, and the authors may wanted to put video diffusion models (Ho et al., 2022) there. Moreover, some prior works that tackle the uncurated label distributions are not discussed and compared in the paper.*
>
> Thank you for pointing out the incorrect citation. We modify this citation and confirm that there are no issues with other references. We also add discussion of studies on generative models that deal with uncurated label datasets in Appendix B.4 of the revised version.
>
> ***
> ***
>
> >**Q7. [Color bar]** *(Weaknesses 6) It would be beneficial to include a color bar in Fig 8 to interprete the meaning of the colors presented.*
>
> Thank you for the suggestion. The color of the arrow means the norm of the score vector. We add a color bar and the meaning of the colors in Figure 8 of the revised version.

---

> ### Author Response · Authors · 2023-11-22
> **Updated experimental results**
>
> We update the newly obtained experimental results for **Q1** and **Q4**. A detailed discussion of each is provided below.
> ***
> For **Q1. [Effect of the performance of the noisy classifier]**, we update the generation performance for the TDSM models trained with different training levels of noisy label classifiers in Table 15 of the updated version. We color the names of the columns corresponding to the newly added results in purple. We also visualize these results in Figure 15, i.e., the change of the generation performance over the classifier training epochs.
>
> The newly added results also support our interpretation. We can see that all TDSM models outperform the baseline (DSM) model on most metrics. The TDSM model with sufficiently trained classifiers has better unconditional performance, indicating that the quality of the samples is improved by receiving a sufficiently diverse set of perturbed examples. However, this training leads to overconfidence in the noisy labels, especially in the data space at $t=0$, where the information in the clean label is reduced, resulting in a slight decrease in conditional performance.
>
> ***
> For **Q4. [Impact of potential label noise in the benchmark dataset]**, we update the results of considering potential label noise in the dataset. The table below shows the results of the TDSM model assuming a 43% noise rate on the CIFAR-10 dataset with 40% symmetric noise. In this case, we assume that the 60% of unaffected data in our noisy label generation process has the 5% potential noise predicted by the previous experiment, resulting in an additional 3% of total data.
>
> |      | Metric                   | DSM      | TDSM (40%) | TDSM (43%) |   |
> |------|--------------------------|----------|------------|------------|---|
> | un   | FID ($\downarrow$)       | **2.07** | 2.43       | 2.53       |   |
> |      | IS ($\uparrow$)          | 9.83     | **9.96**   | **9.96**   |   |
> |      | Density ($\uparrow$)     | 100.94   | 111.63     | **113.21** |   |
> |      | Coverage ($\uparrow$)    | 80.93    | 82.03      | **82.18**  |   |
> | cond | CW-FID ($\downarrow$)    | 30.45    | 15.92      | **15.76**  |   |
> |      | CAS ($\uparrow$)         | 47.21    | **62.28**      |  62.10   |   |
> |      | CW-Density ($\uparrow$)  | 73.02    | 97.80      | **100.06** |   |
> |      | CW-Coverage ($\uparrow$) | 71.63    | 78.65      | **79.11**  |   |
>
> Interestingly, we find that assuming potential label noise yields additional performance improvements for most metrics. This result further supports our conjecture that the benchmark dataset contains examples with noisy or ambiguous labels. We include the result and discussion in Appendix F.14 of the revised version, colored in purple.

---

> > ### Comment · Reviewer_TMN7 · 2023-11-23
> >
> > Thank you for your comprehensive response and the additional experimental results and clarifications provided in the latest version of your manuscript. Given the added experiment results and clarifications in the latest manuscripts, I have few concern about this paper. As a result, I am maintaining my current evaluation score and am inclined towards recommending acceptance of the paper.

---

> > > ### Author Response · Authors · 2023-11-23
> > >
> > > Thank you for taking the time to read our response, additional experimental results, and the updated manuscript. We are pleased that most concerns have been addressed. If you have any further requests or inquiries about our work, please feel free to let us know. Thank you.

---

> ### Author Response · Authors · 2023-11-23
>
> Dear Reviewer TMN7,
>
> We sincerely appreciate the reviewer's constructive feedback, which is invaluable in improving our work. We kindly ask the reviewer to consider our response and the revised version, where we have tried to incorporate your feedback as thoroughly as possible. As the discussion period ends in 12 hours, we are curious to know if we have adequately understood and addressed your concerns. We are always open to further discussion about our research at any time. Again, we appreciate the reviewer's efforts.
>
> Best Regards,

---

### Official Review · Reviewer_Zgz7 · 2023-10-31

**Soundness:** 3 good
**Presentation:** 3 good
**Contribution:** 2 fair
**Rating:** 6
**Confidence:** 4

**Summary:**

This paper introduces a new label-noise robust method for training conditional diffusion models. This is achieved by making use of an estimated transition relation from noisy labels to clean labels. Some theoretical analyses have also been proposed under the class-dependent label-noise setting. Experiments across various datasets demonstrate the effectiveness of the proposed method.

**Strengths:**

1. **Targeting an Important Problem:** The paper aims to handle noisy labels in large-scale datasets used for training diffusion models. Addressing this problem is crucial as it is a common and practical challenge in the deployment of these models in real-world scenarios.
2. **A New Approach to Noisy Labels in Diffusion Models:** The paper introduces a new methodology to address noisy labels in conditional diffusion models, a topic not extensively covered in existing literature.
3. **Clarity and Accessibility:** The introduction and overall presentation of the paper are clear and easy to follow.

**Weaknesses:**

1. **Missing Citations:** The paper could be significantly enhanced by including additional relevant literature. Learning the transition relation from noisy labels to clean labels has been previously explored under the umbrella of mixture proportion estimation. Moreover, methods for estimating the transition matrix in class-dependent settings are also very related. Acknowledging popular works could provide a richer theoretical foundation.
2. **Missing Baselines:** There are many methods for learning with noisy labels. It would be beneficial to combine existing state-of-the-art (SOTA) methods for learning with noisy labels to obtain estimated clean labels. Then, utilizing conditional diffusion models on these cleaned labels and comparing the performance with the author's method could offer a more comprehensive evaluation of the proposed method’s efficacy compared to current alternatives.
3. **Unclear Advantage:** There is a need for a more detailed explanation regarding the advantage of the proposed method over existing methods, especially in recovering clean labels. Since the performance of diffusion models in noisy label scenarios hinges significantly on the accuracy of label recovery, explaining how the method enhances this aspect compared to others would greatly benefit readers in understanding the true potential and innovation of your approach.
4. **Limited Application:** The focus on class-dependent noise settings might limit the broader applicability of the proposed method. The assumption of class-dependent noise could be strong and not verifiable in practical scenarios. It would be insightful if the paper could discuss the potential implications and limitations of this assumption, including how it might affect the generalizability of the proposed method to other noise settings or real-world applications where such assumptions may not be easily verified.

**Questions:**

1. Could the authors elaborate on how their method of learning the transition relation from noisy labels to clean labels differs from or aligns with the existing literature in mixture proportion estimation?
2. Could the authors provide further details on how their method more effectively recovers clean labels compared to existing state-of-the-art methods VolMinNet (ICML 21), InstanceGM (WACV23), and DivideMix (ICLR20)?
3. Would the authors consider adding additional experiments to demonstrate the performance when combining the existing state-of-the-art methods for learning with noisy labels with a conditional diffusion model? Additionally, could they compare the accuracy of clean label recovery with these methods?
4. Could the authors explain which specific mechanisms or features in their approach contribute to improved label recovery?
5. How do the authors verify the presence of class-dependent noise in practical applications?

---

> ### Author Response · Authors · 2023-11-17
> **Response to Reviewer Zgz7 (Part 1)**
>
> We appreciate the thorough reviews and valuable comments. We address the concerns below.
>
> >**Q1. [Missing Citations]** *(Weaknesses 1) The paper could be significantly enhanced by including additional relevant literature. Learning the transition relation from noisy labels to clean labels has been previously explored under the umbrella of mixture proportion estimation. Moreover, methods for estimating the transition matrix in class-dependent settings are also very related. Acknowledging popular works could provide a richer theoretical foundation.*
>
> We briefly discussed the methods of existing noisy label learning studies in Appendix B.2 of the original manuscript. As the reviewer pointed out, the methods of mixture proportion estimation and transition matrix estimation are highly relevant to our method. We have added more detailed descriptions of these methods in Appendix B.3 of the revised manuscript, and we have mentioned this discussion in the main paper.
>
> ***
> ***
>
> >**Q2. [Missing Baselines]** *(Weaknesses 2) There are many methods for learning with noisy labels. It would be beneficial to combine existing state-of-the-art (SOTA) methods for learning with noisy labels to obtain estimated clean labels. Then, utilizing conditional diffusion models on these cleaned labels and comparing the performance with the author's method could offer a more comprehensive evaluation of the proposed method’s efficacy compared to current alternatives.*
>
> In Section 4.4, we conducted experiments comparing our model with the baseline model proposed by the reviewer (*combining existing SOTA methods for learning with noisy labels to obtain estimated clean labels, and utilizing conditional diffusion models on these cleaned labels*). We applied the methods of the state-of-the-art models based on the transition matrix (VolMinNet, Li et al., ICML 2021 [B1]) and the latest model (DISC, Li et al., CVPR 2023 [B2]) to CIFAR datasets. The complete results are shown in Tables 10 and 11 of Appendix F.8 of the revised version. The experimental results indicate that applying our objective consistently improves the performance even with the cleaned labels. Therefore, our approach tackles the noisy label problem from a diffusion model learning perspective, providing an orthogonal direction compared to conventional noisy label methods.
> ***
> [B1] Li, X., Liu, T., Han, B., Niu, G., & Sugiyama, M. (2021, July). Provably end-to-end label-noise learning without anchor points. In International conference on machine learning (pp. 6403-6413). PMLR.
> [B2] Li, Y., Han, H., Shan, S., & Chen, X. (2023). DISC: Learning from Noisy Labels via Dynamic Instance-Specific Selection and Correction. In Proceedings of the IEEE/CVF Conference on Computer Vision and Pattern Recognition (pp. 24070-24079).

---

> ### Author Response · Authors · 2023-11-17
> **Response to Reviewer Zgz7 (Part 2)**
>
> >**Q3. [Unclear Advantage]** *(Weaknesses 3) There is a need for a more detailed explanation regarding the advantage of the proposed method over existing methods, especially in recovering clean labels. Since the performance of diffusion models in noisy label scenarios hinges significantly on the accuracy of label recovery, explaining how the method enhances this aspect compared to others would greatly benefit readers in understanding the true potential and innovation of your approach.*
>
> >*(Questions 4) Could the authors explain which specific mechanisms or features in their approach contribute to improved label recovery?*
>
> We provide the response to this question in the Global Response. For your convenience, we also copy and paste the response provided in the Global Response.
>
> ***
> **[Advantage of the proposed method]**
>
> Label recovery is an intuitive way to overcome the problem of noisy labels in a dataset. Our method does not specifically focus on robustness to noisy labels through label recovery; instead, it focuses on robustness to noisy labels from a diffusion model training perspective. Consequently, our approach can be synergistically combined with existing label recovery methods to further improve performance. The analysis of this synergy is detailed in Section 4.4.
>
> It is important to analyze how our method overcomes the noisy label robustness in diffusion models. Our analysis indicates that our proposed TDSM objective provides the diffusion model with information about the diffused label noise in the dataset. This information is represented by the transition-aware weight function $w$, depending upon the diffusion time. Therefore, we estimate the instance- and time-dependent label transition probability with the transition-aware weight function.
>
> **[Transition-aware weight function plays a different role from a typical classifier]**
>
> We want to emphasize that the transition-aware weight function $w$ plays a different role from a typical classifier. This probability represents the relationship between the noisy and clean label conditional scores (which are also time-dependent) and is an important element of the proposed training objective of a diffusion model. Therefore, we need to estimate this label transition probabilities over diffusion timesteps.
>
> Specifically, our transition-aware weight function can be reformulated as follows:
> $$
> w(x_t,\tilde{y},y,t)=\frac{p_t(Y=y|x_t)p(\tilde{Y}=\tilde{y}|Y=y)}{p_t(\tilde{Y}=\tilde{y}|x_t)}.
> $$
> As seen in the expression, the weight function is composed of the clean label classifier $p_t(Y=y|x_t)$, label transition prior $p(\tilde{Y}=\tilde{y}|Y=y)$, and the noisy label classifier  $p_t(\tilde{Y}=\tilde{y}|x_t)$. Given the triplet of perturbed data instance and its corresponding noisy label and timestep, i.e., $(x_t, \tilde{y}, t)$, each component of $w$ has the following characteristics with respect to $y$: 1) $p(Y=y|x_t)$ is maximized when $y$ is the clean label of the clean data $x_0$ of $x_t$; 2) $p(\tilde{Y}=\tilde{y}|Y=y)$ is maximized when $y$ is the noisy label $\tilde{y}$ in general. The reason for 2) is that in general, a given noisy label is sufficiently likely to be a clean label. These two trade-offs imply that the $w$ function does not behave like a clean label classifier.
>
> Furthermore, for large enough $t$, the distribution of $x_t$ converges to a label-independent prior distribution by the design of the diffusion process, so that $p_t(Y|x_t)$ converges to a uniform distribution. Therefore, for sufficiently large $t$, the $w$ function converges to a transition matrix $S$. This phenomenon can also be demonstrated in Figure 2 of the 2-D example.
>
> In summary, the transition-aware weight function needed to overcome noisy labels in a diffusion model is not represented clean label recovery information only, and this function has time-varying information.

---

> ### Author Response · Authors · 2023-11-17
> **Response to Reviewer Zgz7 (Part 3)**
>
> >**Q4. [Limited Application]** *(Weaknesses 4) The focus on class-dependent noise settings might limit the broader applicability of the proposed method. The assumption of class-dependent noise could be strong and not verifiable in practical scenarios. It would be insightful if the paper could discuss the potential implications and limitations of this assumption, including how it might affect the generalizability of the proposed method to other noise settings or real-world applications where such assumptions may not be easily verified.*
>
> In Section 4.3, we conducted experiments on the Clothing-1M dataset, which suffers from real-world label noise and does not adhere to the class-dependent noise assumption. Table 3 shows that our model even achieves superior performance on this dataset. While theoretical developments require assumptions about the noise setting, our model remains effective in real-world scenarios where these assumptions may not hold.
>
> On the other hand, in the existing literature on noisy labels, there has been an extension from class-dependent label noise settings to instance-dependent label noise settings [B3, B4, B5, B6]. Especially in the context of learning diffusion models, there is a distinctive characteristic that, despite the class-dependent label noise assumption, the prediction of instance-dependent label transition probabilities is necessary. Therefore, investigating training methods under the current assumptions in the context of diffusion models lays the foundation for developing better methods. Similar to existing classification methods, extending our approach beyond the current assumption would be a valuable future work.
> ***
> [B3] Xia, X., Liu, T., Han, B., Wang, N., Gong, M., Liu, H., Niu, G., Tao, D., & Sugiyama, M. (2020). Part-dependent label noise: Towards instance-dependent label noise. Advances in Neural Information Processing Systems, 33, 7597-7610.
> [B4] Berthon, A., Han, B., Niu, G., Liu, T., & Sugiyama, M. (2021, July). Confidence scores make instance-dependent label-noise learning possible. In International conference on machine learning (pp. 825-836). PMLR.
> [B5] Yao, Y., Liu, T., Gong, M., Han, B., Niu, G., & Zhang, K. (2021). Instance-dependent label-noise learning under a structural causal model. Advances in Neural Information Processing Systems, 34, 4409-4420.
> [B6] Cheng, D., Liu, T., Ning, Y., Wang, N., Han, B., Niu, G., Gao, X., & Sugiyama, M. (2022). Instance-dependent label-noise learning with manifold-regularized transition matrix estimation. In Proceedings of the IEEE/CVF Conference on Computer Vision and Pattern Recognition (pp. 16630-16639).
>
> ***
> ***
>
> >**Q5. [Relation with the mixture proportion estimation]** *(Questions 1) Could the authors elaborate on how their method of learning the transition relation from noisy labels to clean labels differs from or aligns with the existing literature in mixture proportion estimation?*
>
> Our estimation of transition-aware weights differs from the traditional mixture proportion methods in that our weights depend on the diffusion timestep, which requires training a **time-dependent** noisy label classifier.
>
> Additionally, it is important to note that our weight function represents the weights related to the data score relationship, and these values do not directly indicate the proportion of the distribution. Therefore, conventional methods for estimating mixture proportions based on the samples may not be directly applicable. We add this discussion in Appendix B.3 of the revised manuscript.
>
> ***
> ***
>
> >**Q6. [Clean label recovery and combining the existing methods for learning with noisy labels]** *(Questions 2) Could the authors provide further details on how their method more effectively recovers clean labels compared to existing state-of-the-art methods VolMinNet (ICML 21), InstanceGM (WACV23), and DivideMix (ICLR20)?*
>
> >*(Questions 3) Would the authors consider adding additional experiments to demonstrate the performance when combining the existing state-of-the-art methods for learning with noisy labels with a conditional diffusion model?*
>
> As we mentioned in the global response and Q3, our method does not focus on clean label recovery, which is the goal of existing methods for classification learning with noisy label such as VolMinNet, InstanceGM, and DividedMix, etc. Our method aims to overcome noisy labels that are characteristic of diffusion models. As a result, by combining our method with existing clean label recovery methods, we can further improve the robustness of the diffusion model to noisy labels. We demonstrated this in Section 4.4 of the manuscript, using the VolMinNet (Li et al., ICML 21) [B1] and DISC (Li et al., CVPR 23) [B2].

---

> ### Author Response · Authors · 2023-11-17
> **Response to Reviewer Zgz7 (Part 4)**
>
> >**Q7. [Accuracy of clean label recovery]** *(Questions 3) Additionally, could they compare the accuracy of clean label recovery with these methods?*
>
> We provide the response to this question in the Global Response. For your convenience, we also copy and paste the response provided in the Global Response.
>
> ***
> **[Experimental results of clean label accuracy]**
>
> It is possible to check the accuracy of the transition-aware weight function on the training dataset since the function provides the probability of a clean label given a noisy label and an instance. Figure 13 in the revised manuscript shows the clean label accuracy of the transition-aware weight function over diffusion timesteps. Since the noisy labels in the dataset are used as the input conditions of the baseline model for the entire diffusion timesteps, we plot the percentages of clean labels in the datasets as dots at $t=0$ and plot a dashed horizontal line for comparison.
>
> Focusing on 'Ours (VolMinNet + TDSM)', which provides the best generation performance, we found that the behavior of the $w$ function varies with diffusion timesteps, as mentioned above. In particular, as $t$ increases, the clean label accuracy converges to the clean rate of the given dataset, which is 0.6. This is because the $w$ function converges to the label transition prior $p(\tilde{Y}|Y)$. 'Base (VolMinNet + DSM)', which does not take this into account, learns the diffusion model only with corrected labels, leading to the performance difference. Therefore, while existing noisy label methods that focus on clean label recovery certainly contribute significantly to generative performance, considering the TDSM objective in diffusion model training enables additional performance improvement independent of clean label recovery.
>
> ***
> ***
>
> >**Q8. [Practical applications of class-dependent label noise]** *(Questions 5) How do the authors verify the presence of class-dependent noise in practical applications?*
>
> We believe that the class-dependent noise is still applicable in practical applications. One example is when classes are distinguished by fine-grained concepts such as breeds of dogs, species of birds, models of cars [B7, B8]. In this case, the distinction between classes is ambiguous, so that non-expert human labelers will have difficulty distinguishing between them, regardless of the data instances.
> ***
> [B7] Krause, J., Sapp, B., Howard, A., Zhou, H., Toshev, A., Duerig, T., ... & Fei-Fei, L. (2016). The unreasonable effectiveness of noisy data for fine-grained recognition. In Computer Vision–ECCV 2016: 14th European Conference, Amsterdam, The Netherlands, October 11-14, 2016, Proceedings, Part III 14 (pp. 301-320).
> [B8] Patrini, G., Rozza, A., Krishna Menon, A., Nock, R., & Qu, L. (2017). Making deep neural networks robust to label noise: A loss correction approach. In Proceedings of the IEEE conference on computer vision and pattern recognition (pp. 1944-1952).

---

> ### Author Response · Authors · 2023-11-23
>
> Dear Reviewer Zgz7,
>
> We sincerely appreciate the reviewer's constructive feedback, which is invaluable in improving our work. We kindly ask the reviewer to consider our response and the revised version, where we have tried to incorporate your feedback as thoroughly as possible. As the discussion period ends in 12 hours, we are curious to know if we have adequately understood and addressed your concerns. We are always open to further discussion about our research at any time. Again, we appreciate the reviewer's efforts.
>
> Best Regards,

---

### Official Review · Reviewer_BRfo · 2023-11-01

**Soundness:** 3 good
**Presentation:** 3 good
**Contribution:** 2 fair
**Rating:** 5
**Confidence:** 3

**Summary:**

the authors find the noisy-label conditional score can be expressed as a convex combination of the clean- label conditional scores with some coefficients
, accordingly they propose a weighted loss function to address the problem of noisy labels in class-conditional diffusion models.

**Strengths:**

1.	The paper is well organized and the proofs are detailed.
2.	This paper is the first work to consider the influence of noisy label condition to the generation performance in diffusion models

**Weaknesses:**

The meaning of this work is limited. The proposed method is tailored for diffusion models which are conditioned on class, but most existing diffusion models are conditioned on text or other modalities, and the class label also can be expressed by language. In addition, noisy label datasets are not common in diffusion model.

**Questions:**

Please refer to my comments.

---

> ### Author Response · Authors · 2023-11-17
> **Response to Reviewer BRfo**
>
> We appreciate the thorough reviews and valuable comments. We address the concerns below.
>
> >**Q. [Meaning of work]** *(Weaknesses) The meaning of this work is limited. The proposed method is tailored for diffusion models which are conditioned on class, but most existing diffusion models are conditioned on text or other modalities, and the class label also can be expressed by language. In addition, noisy label datasets are not common in diffusion model.*
>
> We appreciate your feedback. We agree with the reviewer that recent diffusion model advancements have led to successful outcomes in text-conditional models, introducing notable models like Stable Diffusion and DALL-E3. However, it is important to note that class-conditional generative models have been widely implemented across various sectors of the industry: image editing [A1], medical data [A2, A3], and manufacturing [A4]. In fields where such models are essential, utilizing diffusion models, which have strong performances in generative tasks, could lead to improved results. Nevertheless, it is crucial to acknowledge that the problem of noisy labels in datasets continues to pose a significant challenge in real-world applications [A5, A6, A7].
>
> One possible approach involves adapting publicly available text-conditional diffusion models for the purpose of class-conditional generation. However, in cases involving data privacy and label specificity, it may not be feasible to utilize the public text-conditional model. Therefore, it still becomes important to train a model from the incomplete dataset. Ongoing research on a class-conditional diffusion model that can effectively handle noisy labels from an application perspective is a significant endeavor. This research has the potential to solve a crucial problem in real-world scenarios and broaden the scope of diffusion models.
>
> We would like to emphasize that there is no previous research that specifically addresses label noise in diffusion models. Much research on noisy label classification has extended from class-level studies to text-level or other modality-level studies [A8, A9]. Our work on the class-conditional is a necessary step to solidify the theoretical foundation on the noisy-image and noisy-text conditioned generation, which we believe that those issues will be published in the future.
> ***
> [A1] Perarnau, G., van de Weijer, J., Raducanu, B., & Álvarez, J. M. (2016). Invertible Conditional GANs for image editing. NIPS Workshop on Adversarial Training.
> [A2] Das, H. P., Tran, R., Singh, J., Yue, X., Tison, G., Sangiovanni-Vincentelli, A., & Spanos, C. J. (2022, June). Conditional synthetic data generation for robust machine learning applications with limited pandemic data. In Proceedings of the AAAI Conference on Artificial Intelligence (Vol. 36, No. 11, pp. 11792-11800).
> [A3] Pinaya, W. H., Tudosiu, P. D., Dafflon, J., Da Costa, P. F., Fernandez, V., Nachev, P., ... & Cardoso, M. J. (2022, September). Brain imaging generation with latent diffusion models. In MICCAI Workshop on Deep Generative Models (pp. 117-126).
> [A4] Iyer, A., Dey, B., Dasgupta, A., Chen, W., & Chakraborty, A. (2019). A conditional generative model for predicting material microstructures from processing methods. Second Workshop on Machine Learning and the Physical Sciences in NeurIPS 2019.
> [A5] Welinder, P., Branson, S., Perona, P., & Belongie, S. (2010). The multidimensional wisdom of crowds. Advances in neural information processing systems, 23.
> [A6] Xiao, T., Xia, T., Yang, Y., Huang, C., & Wang, X. (2015). Learning from massive noisy labeled data for image classification. In Proceedings of the IEEE conference on computer vision and pattern recognition (pp. 2691-2699).
> [A7] Cheng, H., Zhu, Z., Li, X., Gong, Y., Sun, X., & Liu, Y. (2020, October). Learning with Instance-Dependent Label Noise: A Sample Sieve Approach. In International Conference on Learning Representations.
> [A8] Huang, Z., Niu, G., Liu, X., Ding, W., Xiao, X., Wu, H., & Peng, X. (2021). Learning with noisy correspondence for cross-modal matching. Advances in Neural Information Processing Systems, 34, 29406-29419.
> [A9] Yang, M., Huang, Z., Hu, P., Li, T., Lv, J., & Peng, X. (2022). Learning with twin noisy labels for visible-infrared person re-identification. In Proceedings of the IEEE/CVF conference on computer vision and pattern recognition (pp. 14308-14317).

---

> ### Author Response · Authors · 2023-11-23
>
> Dear Reviewer BRfo,
>
> We sincerely appreciate the reviewer's constructive feedback, which is invaluable in improving our work. We kindly ask the reviewer to consider our response and the revised version, where we have tried to incorporate your feedback as thoroughly as possible. As the discussion period ends in 12 hours, we are curious to know if we have adequately understood and addressed your concerns. We are always open to further discussion about our research at any time. Again, we appreciate the reviewer's efforts.
>
> Best Regards,

---

> > ### Comment · Reviewer_BRfo · 2023-11-23
> >
> > Dear Authors,
> >
> > I am still unclear that whether label noise is a critical issue of diffusion models in practice. Moreover, what is the contribution of this paper compared to other techniques for addressing the label noise?

---

> > > ### Author Response · Authors · 2023-11-23
> > >
> > > Thank you for taking the time to read our response.
> > > ***
> > > **[Label noise is a critical issue of diffusion models in practice]**
> > >
> > > First, we would like to mention the presence of noisy labels in the training datasets of diffusion models in practical situations. For example, the stable diffusion paper [1] mentions the use of the large-scale LAION dataset [2] in text-to-image generation experiments. Such large-scale datasets inevitably contain noisy labels introduced by human annotators or web crawlers during the data collection process. It has been reported that using pre-trained models on noisy labeled datasets can lead to performance degradation [3]. Therefore, we argue that label noise is also a significant problem for diffusion models, which are widely used in practical scenarios with large-scale datasets.
> > >
> > > In this context, we claim that research considering noise in text labels builds on the foundation of noise in class labels. Text labels can be viewed as a continuous extension of the discrete class label $Y$, i.e., replacing the label space $\mathcal{Y}=${$1,…,c$} with $\mathbb{R}^m$. Consequently, the label transition probability $p(Y=y|\tilde{Y}=\tilde{y})$ can be extended from a probability mass function to a probability density function. Therefore, while our current research is based on class conditions, we are the first to consider the impact of label noise in diffusion models and to propose a method specific to diffusion models, and we argue that our study will serve as a foundation for extensions such as noisy-text conditional generation.
> > > ***
> > > [1] Rombach, R., Blattmann, A., Lorenz, D., Esser, P., & Ommer, B. (2022). High-resolution image synthesis with latent diffusion models. In Proceedings of the IEEE/CVF conference on computer vision and pattern recognition (pp. 10684-10695).
> > > [2] Schuhmann, C., Vencu, R., Beaumont, R., Kaczmarczyk, R., Mullis, C., Katta, A., ... & Komatsuzaki, A. (2021). Laion-400m: Open dataset of clip-filtered 400 million image-text pairs. arXiv preprint arXiv:2111.02114.
> > > [3] Anonymous. (2023). Understanding and Mitigating the Label Noise in Pre-training on Downstream Tasks. Submitted to The Twelfth International Conference on Learning Representations. Retrieved from https://openreview.net/forum?id=TjhUtloBZU.
> > > ***
> > > ***
> > > **[Contribution compared to other techniques for addressing the label noise]**
> > >
> > > We describe our contribution by comparing it to 1) methods that address the problem of noisy labels in the classification, and 2) methods that address problem of noisy labels in generative models.
> > >
> > > **1) Methods that address problem of noisy labels in the classification**
> > >
> > > Since these methods deal with classification problems, they are only concerned with finding clean labels well. However, since our goal is to generate images that match the given labels, we need to consider the impact of noisy labels on the diffusion model. Therefore, our model aims to mitigate the problem of noisy labels in diffusion model training, besides recovering clean labels, which is the goal of existing methods. Therefore, we propose a new diffusion model training objective, TDSM, to provide the diffusion model with information about the noisy labels in the dataset.
> > >
> > > Technically, the novelty of this objective is that we propose a transition-aware weight function. This function plays a different role from the clean classifier that existing methods try to find. This function represents the relationship between the noisy and clean label conditional scores, and this function has time-varying information. We explained this in detail in the Global Response, [Transition-aware weight function plays a different role from a typical classifier].
> > >
> > > **2) Methods that address problem of noisy labels in generative models**
> > >
> > > There have been efforts in the GAN community to overcome the problem of noisy labels [4, 5]. These methods used only instance- and time-independent transition matrix. However, the diffusion model cannot completely mitigate the impact of noisy labels with the transition matrix alone because it targets the gradient of the log-likelihood, i.e., data score. We show theoretically the impact of noisy label on data score in Theorem 1. This impact could be represented by the transition-aware weighted function, which is time- and instance-dependent unlike the transition matrix.
> > >
> > > We present the detailed explanation of the GAN-based models and the comparison table in below and we included this discussion in the Appendix B.4 of the revised paper.
> > > ***
> > > [4] Thekumparampil, K. K., Khetan, A., Lin, Z., & Oh, S. (2018). Robustness of conditional gans to noisy labels. Advances in neural information processing systems, 31.
> > > [5] Kaneko, T., Ushiku, Y., & Harada, T. (2019). Label-noise robust generative adversarial networks. In Proceedings of the IEEE/CVF Conference on Computer Vision and Pattern Recognition (pp. 2467-2476).

---

### Author Response · Authors · 2023-11-17
**Global Response**

We sincerely thank for the reviewers for their constructive and valuable reviews. We provide individual responses to each reviewer’s concerns. Also, we update the manuscript to reflect the reviewer’s comments. We highlight the changes in blue. Please check our response and feel free to post additional comments.

Additionally, we respond to questions raised by multiple reviewers regarding the **clean label recovery**.
***
**[Advantage of the proposed method]**

Label recovery is an intuitive way to overcome the problem of noisy labels in a dataset. Our method does not specifically focus on robustness to noisy labels through label recovery; instead, it focuses on robustness to noisy labels from a diffusion model training perspective. Consequently, our approach can be synergistically combined with existing label recovery methods to further improve performance. The analysis of this synergy is detailed in Section 4.4.

It is important to analyze how our method overcomes the noisy label robustness in diffusion models. Our analysis indicates that our proposed TDSM objective provides the diffusion model with information about the diffused label noise in the dataset. This information is represented by the transition-aware weight function $w$, depending upon the diffusion time. Therefore, we estimate the instance- and time-dependent label transition probability with the transition-aware weight function.

**[Transition-aware weight function plays a different role from a typical classifier]**

We want to emphasize that the transition-aware weight function $w$ plays a different role from a typical classifier. This probability represents the relationship between the noisy and clean label conditional scores (which are also time-dependent) and is an important element of the proposed training objective of a diffusion model. Therefore, we need to estimate this label transition probabilities over diffusion timesteps.

Specifically, our transition-aware weight function can be reformulated as follows:
$$
w(x_t,\tilde{y},y,t)=\frac{p_t(Y=y|x_t)p(\tilde{Y}=\tilde{y}|Y=y)}{p_t(\tilde{Y}=\tilde{y}|x_t)}.
$$
As seen in the expression, the weight function is composed of the clean label classifier $p_t(Y=y|x_t)$, label transition prior $p(\tilde{Y}=\tilde{y}|Y=y)$, and the noisy label classifier  $p_t(\tilde{Y}=\tilde{y}|x_t)$. Given the triplet of perturbed data instance and its corresponding noisy label and timestep, i.e., $(x_t, \tilde{y}, t)$, each component of $w$ has the following characteristics with respect to $y$: 1) $p(Y=y|x_t)$ is maximized when $y$ is the clean label of the clean data $x_0$ of $x_t$; 2) $p(\tilde{Y}=\tilde{y}|Y=y)$ is maximized when $y$ is the noisy label $\tilde{y}$ in general. The reason for 2) is that in general, a given noisy label is sufficiently likely to be a clean label. These two trade-offs imply that the $w$ function does not behave like a clean label classifier.

Furthermore, for large enough $t$, the distribution of $x_t$ converges to a label-independent prior distribution by the design of the diffusion process, so that $p_t(Y|x_t)$ converges to a uniform distribution. Therefore, for sufficiently large $t$, the $w$ function converges to a transition matrix $S$. This phenomenon can also be demonstrated in Figure 2 of the 2-D example.

In summary, the transition-aware weight function needed to overcome noisy labels in a diffusion model is not represented clean label recovery information only, and this function has time-varying information.

**[Experimental results of clean label accuracy]**

It is possible to check the accuracy of the transition-aware weight function on the training dataset since the function provides the probability of a clean label given a noisy label and an instance. Figure 13 in the revised manuscript shows the clean label accuracy of the transition-aware weight function over diffusion timesteps. Since the noisy labels in the dataset are used as the input conditions of the baseline model for the entire diffusion timesteps, we plot the percentages of clean labels in the datasets as dots at $t=0$ and plot a dashed horizontal line for comparison.

Focusing on 'Ours (VolMinNet + TDSM)', which provides the best generation performance, we found that the behavior of the $w$ function varies with diffusion timesteps, as mentioned above. In particular, as $t$ increases, the clean label accuracy converges to the clean rate of the given dataset, which is 0.6. This is because the $w$ function converges to the label transition prior $p(\tilde{Y}|Y)$. 'Base (VolMinNet + DSM)', which does not take this into account, learns the diffusion model only with corrected labels, leading to the performance difference. Therefore, while existing noisy label methods that focus on clean label recovery certainly contribute significantly to generative performance, considering the TDSM objective in diffusion model training enables additional performance improvement independent of clean label recovery.

---

### Author Response · Authors · 2023-11-22

Dear Reviewers,

We update the manuscript to add the newly obtained experimental results:
1) Generation performance for the TDSM models trained with different training levels of noisy label classifiers for *Effect of the performance of the noisy classifier*. (Appendix F.13)
2) Generation performance of the TDSM model assuming a 43% noise rate on the CIFAR-10 dataset with 40% symmetric noise for *Impact of potential label noise in the benchmark dataset*. (Appendix F.14)

These newly added results are colored purple in the revised version.

Best regards,

---

### Meta-Review · Area_Chair_7r2S · 2023-12-04

**Metareview:**

This is a borderline paper worked on training class-conditional diffusion models using a big dataset with noisy labels. Perhaps there is no major issue --- just no reviewer became excited, because its novelty and significance are overall not very competitive. However, this paper may serve as a good starting point and inspire other researchers who will want to go along this line of research. As a result, I still recommend accepting the paper for publication at ICLR 2024.

The recommendation is based on the internal discussion. All 4 reviewers joined the discussion and expressed their opinions:
> **BRfo** I would like to keep my rating with the concerns listed as follows:\
> The noise defined by the assumption could be always inconsistent with the one in practice, making it unknown whether the proposed method works in real applications.\
> The technique used in this work, i.e., the transition matrix, is similar to the ones in the existing studies.

> **LRz9** The authors have addressed most of my concerns. I also think this paper is at the borderline. I recommend to an acceptance, but I do not hold on to my opinion if it is rejected.

> **Zgz7** I also think this is a borderline paper. Most of my concerns have been addressed by the authors. I only have one concern left: this paper focuses on class-dependent noise settings.\
> Although this assumption can limit the practical usefulness, I agree with Reviewer TMN7 that this still provides a good starting point for further exploration. Therefore, I raised my rate and am inclined to recommend an acceptance.

> **TMN7** In my opinion, this is a borderline paper.\
> I personally think the problem addressed in the paper is useful, as noisy labels are a common issue in real-world scenarios. Especially if the authors extend this work to a text-conditioned case, this could make this exploration more valuable and practical. To this end, I respectively agree with the weakness pointed out by Reviewer BRfo and Zgz7. Despite these points, the current focus on class-conditional aspects still provides a good starting point for further exploration, and this is why I prefer to encourage the authors to dig deeper into this area.\
> Prior to the rebuttal, my primary concern was the limited study on the effects of the noisy detector (classifier) and certain aspects of clarity in the manuscript's presentation. The authors have conducted extensive experiments and addressed these issues satisfactorily.\
> Overall, I am inclined to recommend an acceptance. However, I am also open to the possibility of rejection. In this case, it would be beneficial to advise the authors to extend their research to other instances of noisy-label scenarios, including textual or alternative formats.

Personally, I agree with our reviewers and also think a diffusion model conditioned on the class label is a bit strange, because it is the single-label multi-class setting. I would like my diffusion model to be conditioned on a piece of text or at least multi-labels to have more control on the generated results.

On the other hand, if I really need a class-conditional diffusion model, I may want to find a big enough dataset with clean labels to fine-tune another diffusion model trained in the usual way. Even if I have to train my model from scratch, I may want to go data-centric rather than model-centric --- there are many data/label-cleaning methods and even some commercialized data/label-cleaning products in the area of learning with noisy labels, so that I can first clean my big dataset with noisy labels and then train my model using the smaller dataset with clean labels.

Those points are definitely something the authors should discuss about in the introduction to motivate their own research --- the introduction in its current version is just too short! Please prepare a better introduction in the camera-ready version.

**Justification For Why Not Higher Score:**

Its novelty and significance are overall not very competitive.

**Justification For Why Not Lower Score:**

This paper may serve as a good starting point and inspire other researchers who will want to go along this line of research.

---

### Decision · Program_Chairs · 2024-01-16

Accept (poster)